# 🔍 PDE-SHARP: **PDE** Sᴏʟᴠᴇʀ Hʏʙʀɪᴅs ᴛʜʀᴏᴜɢʜ Aɴᴀʟʏsɪs & Rᴇꜰɪɴᴇᴍᴇɴᴛ Pᴀssᴇs

## Aʙsᴛʀᴀᴄᴛ

Current LLM-driven approaches using test-time computing to generate PDE solvers execute a large number of solver samples to identify high-accuracy solvers. These paradigms are especially costly for complex PDEs requiring substantial computational resources for numerical evaluation. We introduce PDE-SHARP, a framework to reduce computational costs by replacing expensive scientific computation by cheaper LLM inference that achieves superior solver accuracy with 60-75% fewer computational evaluations. PDE-SHARP employs three stages: **(1) Analysis**: mathematical chain-of-thought analysis including PDE classification, solution type detection, and stability analysis; **(2) Genesis**: solver generation based on mathematical insights from the previous stage; and **(3) Synthesis**: collaborative selection-hybridization tournaments in which LLM judges iteratively refine implementations through flexible performance feedback. To generate high-quality solvers, PDE-SHARP requires fewer than 13 solver evaluations on average compared to 30+ for baseline methods, improving accuracy uniformly across tested PDEs by $4\times$ on average, and demonstrates robust performance across LLM architectures, from general-purpose to specialized reasoning models.

## 1 Iɴᴛʀᴏᴅᴜᴄᴛɪᴏɴ

Partial Differential Equations (PDEs) are fundamental to scientific modeling across physics, engineering, and computational sciences, yet writing robust numerical solvers requires specialized numerical analysis expertise for PDE-specific implementation and tuning, with limited flexibility as each solver targets specific PDE types. The success of deep learning has motivated the development of neural PDE solvers, with Physics-Informed Neural Networks (PINNs) (Raissi et al., 2019; Karniadakis et al., 2021) and operator learning methods (Li et al., 2020) emerging as promising alternatives that leverage neural networks to approximate PDE solutions. However, these approaches require extensive training data, lack interpretability, suffer from generalization limits across PDE families, and offer limited accuracy (Rahaman et al., 2019; Wang et al., 2022) The result is an ecosystem of specialized PDE solvers that address particular failure modes without a systematic understanding of underlying limitations (Cuomo et al., 2022; Krishnapriyan et al., 2021; Zhang et al., 2021; Wang et al., 2021a).

Meanwhile, large language models (LLMs) have demonstrated remarkable aptitude for complex mathematical and scientific challenges (Romera-Paredes et al., 2024; Tian et al., 2024). Sophisticated code generation frameworks employ Chain-of-Thought (CoT) reasoning (Welleck et al., 2024; Wei et al., 2023; Kojima et al., 2023), Mixture-of-Agents (MoA) strategies (Sharma, 2024; Wang et al., 2024a), and advanced inference-time scaling techniques (Snell et al., 2024) to achieve state-of-the-art performance across programming tasks. LLM-as-a-judge frameworks (Jiang et al., 2025a; Zheng et al., 2023) typically employ predetermined evaluation rubrics. However, PDE solver evaluation presents unique challenges requiring assessment of mathematical correctness, numerical stability, computational efficiency, and domain-specific accuracy, factors that demand context-dependent evaluation criteria rather than static rubrics, as optimal trade-offs and performance standards vary significantly across PDE families and application domains. The task of creating reliable solver codes for PDEs sits at the intersection of applied mathematics, numerical analysis, and code generation, making it an ideal testbed to evaluate LLMs' mathematical and technical capabilities. Current approaches fall into two general categories. 1) Fine-tuning methods specialize models for mathemat-

ical reasoning (Lu et al., 2024) and subsequent domain-specific adaptation to particular PDE families (Soroco et al., 2025). These require substantial computational resources for multi-stage training and offer limited generalizability across PDE types. 2) Inference-only frameworks using general-purpose LLMs and techniques such as automated debugging (Chen et al., 2023), self-refinement (Madaan et al., 2023), and test-time scaling (Snell et al., 2024). CodePDE (Li et al., 2025) avoids fine-tuning but relies on brute-force sampling strategies, generating and executing 30+ solver candidates to identify optimal solutions. This paradigm becomes especially costly for complex PDEs requiring high-performance computing resources for numerical evaluation.

To address these limitations, we introduce **PDE-SHARP**, an LLM-driven PDE solver generation framework that achieves superior accuracy with 60-75% fewer computational evaluations — through intelligent generation rather than exhaustive sampling — in three stages: **(1) Analysis** analyzes the PDE through structured questions to develop a numerically-stable solver plan; **(2) Genesis** generates solver candidates without immediate execution; **(3) Synthesis** uses LLM judges to iteratively select, execute, and refine solvers based on provided performance feedback in each round. With this approach, PDE-SHARP swaps inexpensive LLM inference for expensive scientific computation, only executing refined solvers each round. This exchange is worthwhile for computationally intensive PDEs for which GPU/HPC resources dominate costs.

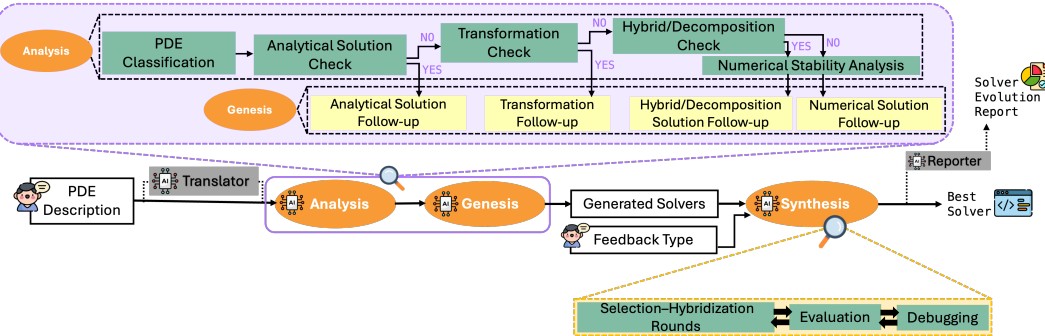

Figure 1: PDE-SHARP framework overview. The three core stages are Analysis, Genesis, and Synthesis. The inner flow in purple expands the Analysis and Genesis stages. The internal flow in orange details the iterative cycles of evaluation, hybridization, and debugging within the Synthesis stage. Analysis deconstructs the PDE to identify solution strategies, which Genesis uses to create PDE solvers. Optional components (Translator, Reporter) enhance usability as explained in section 3. PDE-SHARP generates higher accuracy solvers with 60-75% fewer solver evaluations compared to tested baselines.

**Contributions.** The experimental results highlight PDE-SHARP's key contributions:

- **Computational Efficiency.** PDE-SHARP reduces expensive solver evaluations by 60-75% (requiring fewer than 13 solver evaluations on average compared to 30+ in best-of-$n$ baselines) while achieving superior solution accuracy, demonstrating considerable resource savings for complex simulations.

- **Mathematical Analysis.** PDE-specific mathematical chain-of-thought reasoning with targeted stability analysis produces mathematically-informed solver strategies, leading to higher initial code quality compared to generic template-based generation.

- **Collaborative LLM Tournaments.** PDE-SHARP's synthesis phase improves on standard test-time computing approaches by $4\times$ on average using fewer evaluations.

- **Enhanced Implementation Quality.** Experiments indicate PDE-SHARP solvers achieve bug-free execution in 63-67% fewer debugging iterations (0.33 vs. 0.9-1.4 iterations per solver) and enjoy superior numerical convergence properties.

- **Robustness to LLM Choice.** PDE-SHARP achieves more consistent performance across diverse LLM types (general-purpose, coding-specific, reasoning models) compared to the baselines, showing robustness to the underlying code generator LLM choice.

- **Flexible Feedback Integration.** PDE-SHARP can improve solvers using several feedback mechanisms — solution-based metrics (relative error), physics-based metrics (PDE residual), and no feedback — to adapt to research scenarios from benchmark validation with

known solutions to real-world cases with limited simulation data or physics-only assessments.

## 2 BACKGROUND & RELATED WORK

**Classical Solvers & Neural Methods.** Traditional numerical methods for PDE solving, e.g. finite difference, finite element, and spectral methods, require considerable domain expertise for effective implementation (Strang, 2007; LeVeque, 2007). Modern scientific computing frameworks such as FEniCS (Alnaes et al., 2015), deal.II (Arndt et al., 2021) for finite element, and PETSc (Balay et al., 2025) have facilitated access to these methods for broad PDE classes. However, 1) considerable numerical analysis knowledge is still required for optimal performance; and 2) general approaches fail at exploiting PDE-specific mathematical structure to achieve superior performance. The key challenge is thus identifying which approach suits a particular PDE without extensive domain expertise.

The success of deep learning has motivated extensive research into neural PDE solvers. PINNs variants (Raissi et al., 2019; Wang et al., 2022) approximate PDE solutions through residual minimization. Physics-informed operator learning methods (Li et al., 2020; Lu et al., 2021) learn solution operators rather than individual solutions, offering improved generalization. Feature engineering techniques such as random Fourier features (Wang et al., 2021b; Fazliani et al., 2025), residual-based attention (Anagnostopoulos et al., 2023), and radial basis functions (Zeng et al., 2024) have further enhanced neural solver capabilities. Foundation models leverage transformer architectures for multiphysics problems (McCabe et al., 2024; Hao et al., 2024; Shen et al., 2024; Herde et al., 2024). These neural approaches, however, require extensive training data, lack transparency and interpretability regarding solution generation processes, and have generalization limits.

Custom solver generation offers several advantages over neural surrogates and black-box library usage: full algorithmic transparency enables targeted PDE-specific optimization, simplified debugging and modification, and direct control over every detail. This is crucial when solver behavior needs explanation or when problem-specific modifications are required.

**LLM-Driven Code Generation for PDEs.** The integration of LLMs into scientific computing has emerged along two primary paradigms. First is fine-tuning models pretrained on mathematical tasks for domain-specific applications. MathCoder2 (Lu et al., 2024) demonstrates improved mathematical reasoning through continued training. PDE-Controller (Soroco et al., 2025) continues this approach by fine-tuning MathCoder2-DeepSeekMath on specific PDE families such as heat and wave equations. While effective for targeted applications, this paradigm requires substantial computational resources for multi-stage training and limits generalizability across diverse PDE types. Second is leveraging inference-time optimization techniques to enhance performance. CodePDE (Li et al., 2025) implements automated debugging and test-time sampling for diverse solver generation. Frameworks such as OptiLLM (Sharma, 2024) integrate multiple inference optimization strategies including Chain-of-Thought (CoT), Mixture-of-Agents (MoA), self-reflection, PlanSearch, etc. These approaches typically rely on computationally expensive best-of-$n$ sampling strategies, generating and evaluating large numbers of solver candidates to identify optimal solutions, which becomes prohibitive for complex PDEs requiring substantial evaluation resources.

Both paradigms face fundamental limitations in balancing solution quality with computational efficiency, motivating the need for more intelligent synthesis approaches that leverage mathematical reasoning without exhaustive sampling or extensive fine-tuning requirements.

## 3 PDE-SHARP FRAMEWORK

**Stage 1: Analysis.** PDE-SHARP conducts a systematic five-step mathematical analysis to guide solver generation. The process begins with PDE classification (order, linearity, type, boundary conditions) that informs all subsequent decisions. Sequential checks determine if analytical solutions exist, whether transformations can simplify the problem, and if operator decomposition (e.g., separating diffusion and reaction terms) is viable. Each step either directs the framework toward specialized solution strategies in Stage 2 or continues to the next analysis step as shown in Figure 1. The final stability analysis computes symbolic time-step bounds and selects numerically stable schemes, performed before hybrid/numerical solver generation to ensure robustness. Ablation studies (Appendix B.3) demonstrate the effectiveness of this multi-step paradigm over other LLM-driven alternatives.

**Stage 2: Genesis.** This stage translates the mathematical insights from Analysis into executable solver code. Using the PDE classification result, identified solution strategy, and numerical stability constraints, the framework generates solver implementations that embody the prescribed numerical schemes. Figure 2 demonstrates an example of the Analysis and Genesis stages for the reaction-diffusion equation. A more detailed version of this example, including detailed Synthesis report, appears in Appendix E.

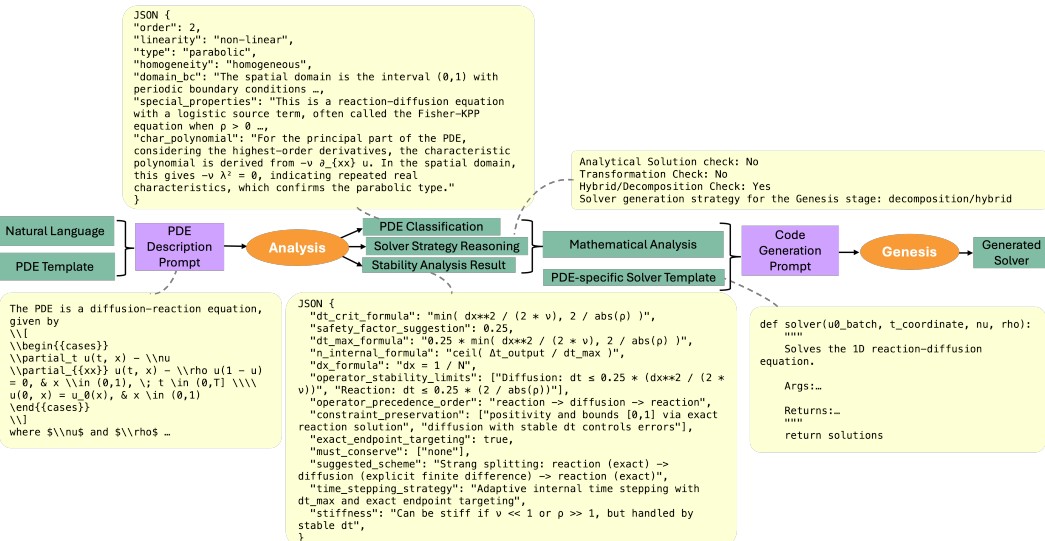

Figure 2: Workflow of the Analysis and Genesis stages for the reaction-diffusion PDE as an example. The Analysis stage (left) decomposes the PDE through mathematical reasoning steps to determine the best initial strategy: PDE classification, analytic solution check (not viable), transformation check (not viable), hybrid/decomposition check (viable!), and finally, stability analysis. For the reaction-diffusion equation, this identifies the analytic solvability of the reaction term, directing the strategy toward a hybrid analytical-numerical approach. The Genesis stage (right) then generates executable solver code that implements this strategy — e.g., Strang splitting with exact reaction integration and finite-difference diffusion — producing initial solver candidates for hybrid tournaments in Synthesis.

**Stage 3: Synthesis** This stage uses Selection–Hybridization Tournaments with LLM judges to iteratively refine solver implementations. Numerical accuracy of the solver can inform judge decisions through a configurable feedback mechanism. Synthesis consists of two main steps:

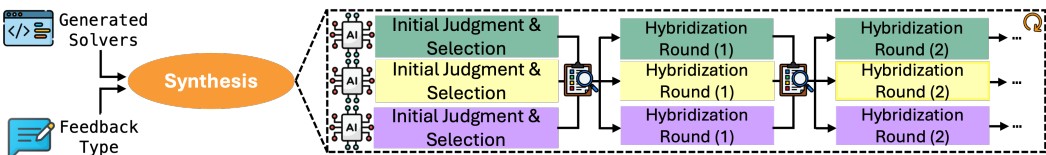

Figure 3: PDE-SHARP Synthesis. This stage can be repeated to address performance saturation (see Appendix B.3.7 for more detail). Additionally, a complete example of the Synthesis stage for the reaction-diffusion task is provided in Appendix E

**(i) Initial Judgment & Selection:** Given the $n$ generated initial solvers and a specified feedback type, each judge LLM produces a selection of its top $\frac{n}{2}$ choices from the initial list with reasoning behind each choice (prompt format detailed in Appendix F.3). Each judge also designates one solver from its top $\frac{n}{2}$ list as a nominee for execution and evaluation using the allowed feedback.

**(ii) Hybridization Rounds:** The three nominated base solvers are executed and their performance results are shared with all judges. Each judge then proposes modifications to their base solver using a diff/patch format to ensure incremental changes that preserve working code structure and encourage

local fixes, with technical justification for each modification. The modified solvers are executed and results again shared with all judges. This process repeats until performance improvements saturate across consecutive rounds or as specified by the user.

When performance improvements saturate or the maximum number of hybridization rounds is reached, the framework initiates another **judging cycle** that repeats steps (i) and (ii) with an expanded solver set including all previously generated hybrids, their technical justifications, and performance feedback from previous rounds. Judges maintain context within each cycle but reset between cycles, evaluating the expanded set from scratch, to encourage exploration of new strategies.

**Feedback Types.** The Synthesis stage can incorporate different performance metrics to guide judge decisions during tournaments. We discuss three feedback types: (1) nRMSE: normalized root mean squared error against reference solutions; (2) PDE residual feedback: physics-based residual computation that requires no reference data; and (3) no feedback: judges rely purely on code analysis. The choice of feedback type allows adaptation to different research scenarios — from benchmark validation with known solutions to real-world cases with limited reference data. PDE-specific feedback types and their combinations could also be employed for domain-specific optimization. Additional discussions and results appear in Appendix B.3.6.

**Optional Stages.** PDE-SHARP includes two optional components for enhanced usability (Figure 1): **Translator** converts natural language PDE descriptions into the structured mathematical templates required by the Analysis stage. When user input lacks necessary detail, it requests additional information before proceeding. Users can alternatively bypass this stage by directly providing pre-formatted templates. **Reporter** generates detailed reports on solver evolution throughout the tournament process, enhancing framework interpretability. These reports can serve as feedback for subsequent runs on the same problem, enabling iterative refinement strategies.

## 4 EXPERIMENTS

We compare PDE-SHARP against a number of LLM-driven baseline methods across five representative PDE tasks from PDEBench (Takamoto et al., 2024) (Table 1). Discussions on neural methods and some LLM-driven approaches (agentic workflows, fine-tuned mathematical models, etc.) appear in Appendix A. In our experiments, we focus on LLM-driven baselines using test-time computing for code generation that directly compete with PDE-SHARP's approach. **CodePDE** (Li et al., 2025) generates solvers using chain-of-thought prompting and executes all samples to report the best performance. A refined variant, **CodePDE-R**, is also tested as a baseline. **OptiLLM** (Sharma, 2024) implements inference optimization techniques including Chain-of-Thought (CoT), Mixture-of-Agents (MoA), and Cerebras Planning and Optimization (CePO). Experimental details appear in Appendix A.

**Experimental Setup:** All methods generate $N = 32$ initial solver candidates for fair comparison (Appendix B.3.4). Baselines execute all candidates (CodePDE-R executes 44 with refinements). PDE-SHARP uses three judge LLMs (Appendix B.3.5) in collaborative tournaments, executing only refined candidates per hybridization round. For Section 4 experiments, PDE-SHARP uses nRMSE on 100 validation samples as tournament feedback. All methods are evaluated on a separate test set of 100 random PDEBench samples per PDE task (Table 2). Additional feedback types and judge configurations appear in Appendix B.3.

Table 1: Tested PDEs; details in Appendix C. **Dimension** column indicates the *spatial dimension* and NL stands for non-linear in the table.

| PDE | Dimension | Type | State | Solution Behavior |
|-----|-----------|------|-------|-------------------|
| Advection | 1D | Linear | Time-dependent | Smooth |
| Burgers | 1D | Highly NL | Time-dependent | Shock-forming |
| Reaction-Diffusion | 1D | Mildly NL | Time-dependent | Smooth |
| Navier-Stokes | 1D | Highly NL | Time-dependent | Shock-forming |
| Darcy Flow | 2D | Mildly NL | Steady-state | Smooth |

## 4.1 RESULTS & ANALYSIS

Table 2 shows solver accuracy across all PDEs and baselines. The following observations are immediately apparent.

**PDE-SHARP is more robust to code generator LLM selection.** Table 2 shows that the solution quality for baseline methods depends strongly on the LLM. In contrast, PDE-SHARP performs more consistently across all tested LLMs; results for more LLMs are appear Appendix B.1. This uniform performance indicates PDE-SHARP's tournament hybridization stage effectively mitigates the limitations of individual code generators, producing higher-quality solvers that are largely independent of the underlying LLM.

**PDE-SHARP significantly improves solver accuracy for specific PDEs.** PDE-SHARP improves accuracy by over $4\times$ overall (geometric mean), with particularly impressive performance on the reaction-diffusion and advection tasks. For reaction-diffusion, PDE-SHARP's Analysis stage immediately identifies that the reaction component admits an analytical solution, directing all 32 initial solver candidates toward hybrid analytical-numerical approaches that achieve superior numerical stability. Baseline methods rarely discover this hybrid strategy, as shown in Figure 4a.

Table 2: PDE-SHARP improves solver accuracy and is robust to choice of LLM. Solution accuracy is measured by nRMSE relative to the reference solution from PDEBench. Cell colors use a colormap log-normalized independently within each PDE column to highlight per-task variation.

| | | Advection | Burgers | Reaction-Diffusion | Navier-Stokes | Darcy |
|---|---|---|---|---|---|---|
| **OptiLLM-CoT** | Gemma 3 | 5.34e-03 | 5.32e-02 | 2.07e-01 | 9.58e-02 | 8.01e-02 |
| | LLaMA 3.3 | 7.71e-03 | 4.38e-02 | 2.24e-01 | 2.42e-01 | 1.01e+00 |
| | Qwen 3 | 4.67e-03 | 1.52e-03 | 9.38e-01 | 2.63e-01 | 6.34e-01 |
| | DeepSeek-R1 | 4.97e-03 | 3.04e-04 | 2.45e-01 | 8.34e-02 | 5.34e-03 |
| | GPT-4o | 1.72e-03 | 2.12e-03 | 2.23e-02 | 2.01e-01 | 8.51e-01 |
| | o3 | 9.74e-04 | 4.08e-04 | 2.21e-01 | 3.12e-02 | 5.47e-03 |
| **OptiLLM-MoA** | Gemma 3 | 3.97e-03 | 4.21e-03 | 1.74e-01 | 6.78e-02 | 4.69e-02 |
| | LLaMA 3.3 | 1.23e-03 | 4.71e-03 | 1.49e-01 | 2.29e-01 | 2.13e-01 |
| | Qwen 3 | 1.01e-03 | 3.45e-04 | 9.68e-02 | 1.79e-02 | 5.12e-03 |
| | DeepSeek-R1 | 9.74e-04 | 2.49e-04 | 1.48e-01 | 1.65e-02 | 5.01e-03 |
| | GPT-4o | 2.01e-03 | 2.41e-04 | 1.94e-02 | 2.56e-02 | 5.02e-03 |
| | o3 | 1.74e-03 | 2.91e-04 | 2.09e-01 | 1.39e-02 | 5.07e-03 |
| **OptiLLM-CePO** | Gemma 3 | 3.74e-03 | 4.01e-03 | 1.89e-01 | 6.32e-02 | 4.12e-02 |
| | LLaMA 3.3 | 1.11e-03 | 4.53e-03 | 1.36e-01 | 2.18e-01 | 1.98e-01 |
| | Qwen 3 | 1.01e-03 | 3.23e-04 | 8.91e-02 | 1.97e-02 | 4.83e-03 |
| | DeepSeek-R1 | 9.71e-04 | 2.43e-04 | 1.39e-01 | 1.79e-02 | 4.78e-03 |
| | GPT-4o | 9.88e-04 | 2.31e-04 | 1.67e-02 | 2.31e-02 | 4.88e-03 |
| | o3 | 9.88e-04 | 2.74e-04 | 2.03e-01 | 1.49e-02 | 4.81e-03 |
| **CodePDE** | Gemma 3 | 5.61e-03 | 5.17e-02 | 2.13e-01 | 9.29e-02 | 7.69e-02 |
| | LLaMA 3.3 | 7.37e-03 | 4.59e-02 | 2.18e-01 | 2.36e-01 | 1.03e+00 |
| | Qwen 3 | 4.89e-03 | 1.35e-03 | 9.55e-01 | 2.59e-01 | 6.57e-01 |
| | DeepSeek-R1 | 1.01e-03 | 3.04e-04 | 2.13e-01 | 2.80e-02 | 4.80e-03 |
| | GPT-4o | 1.55e-03 | 3.65e-04 | 1.99e-02 | 1.81e-01 | 6.57e-01 |
| | o3 | 9.74e-04 | 2.74e-04 | 1.99e-02 | 9.29e-02 | 4.88e-03 |
| **CodePDE-R** | Gemma 3 | 4.20e-03 | 4.63e-03 | 1.69e-01 | 6.44e-02 | 4.47e-02 |
| | LLaMA 3.3 | 1.02e-03 | 4.59e-03 | 1.43e-01 | 2.36e-01 | 1.92e-01 |
| | Qwen 3 | 9.74e-04 | 3.60e-04 | 9.13e-02 | 1.67e-02 | 4.90e-03 |
| | DeepSeek-R1 | 1.01e-03 | 3.15e-04 | 1.67e-02 | 1.67e-02 | 4.80e-03 |
| | GPT-4o | 9.74e-04 | 2.57e-04 | 1.67e-02 | 2.36e-02 | 4.80e-03 |
| | o3 | 1.01e-03 | 3.60e-04 | 1.43e-01 | 1.31e-02 | 4.90e-03 |
| **PDE-SHARP** | Gemma 3 | 1.01e-03 | 5.60e-04 | 3.01e-03 | 3.14e-02 | 1.72e-02 |
| | LLaMA 3.3 | 9.98e-04 | 4.61e-04 | 3.61e-03 | 5.06e-02 | 1.72e-02 |
| | Qwen 3 | 7.76e-04 | 2.97e-04 | 2.32e-03 | 2.80e-02 | 4.80e-03 |
| | DeepSeek-R1 | 5.24e-04 | 1.48e-04 | 2.29e-03 | 1.37e-02 | 4.74e-03 |
| | GPT-4o | 6.11e-04 | 2.31e-04 | 2.29e-03 | 1.51e-02 | 3.97e-03 |
| | o3 | 9.74e-04 | 3.42e-04 | 5.78e-03 | 1.89e-02 | 7.78e-03 |

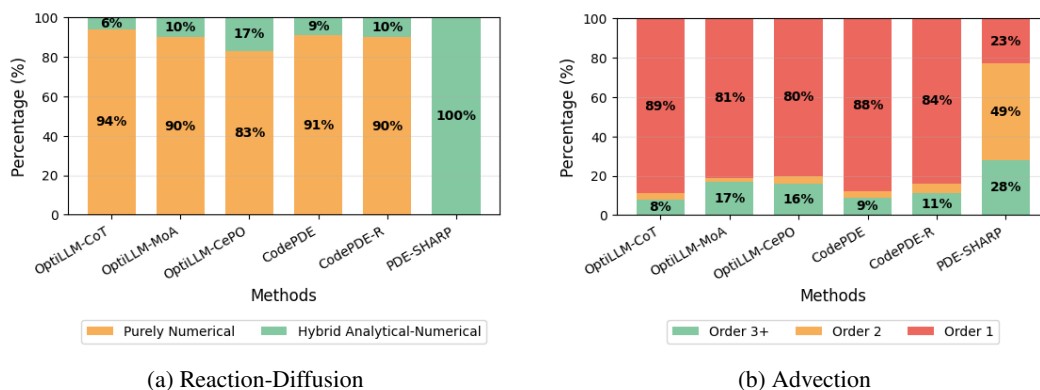

(a) Reaction-Diffusion                    (b) Advection

Figure 4: (a) Other frameworks tend to choose the less accurate purely-numerical approach for the reaction-diffusion PDE, while PDE-SHARP always goes with the superior hybrid approach. (b) PDE-SHARP transitions from first-order discretized analytical to second-order finite-volume approaches through performance-informed tournaments.

For advection, PDEBench reference solutions are generated using finite volume methods (Takamoto et al., 2024), reflecting standard shock-safe computational practice. PDE-SHARP and all other baselines initially attempt analytical solutions, and the baselines keep their analytical approach even through refinement (e.g. in CodePDE-R). PDE-SHARP's performance-informed tournaments, on the other hand, encourage PDE-SHARP to adapt to the data, as demonstrated in Figure 4b. When persistent $10^{-3}$ errors reported as feedback indicate a mismatch between analytical and reference solutions, the judge LLMs converge on second-order finite-volume schemes that better match the dataset characteristics. This adaptation occurs through feedback alone, without manual intervention, demonstrating how collaborative tournaments can optimize for evaluation criteria while maintaining computational efficiency. This adaptive behavior varies with different feedback types as users can choose an optimization target to reflect available data (Figure 5). A study on generated advection solvers and their features appears in Appendix D.1.

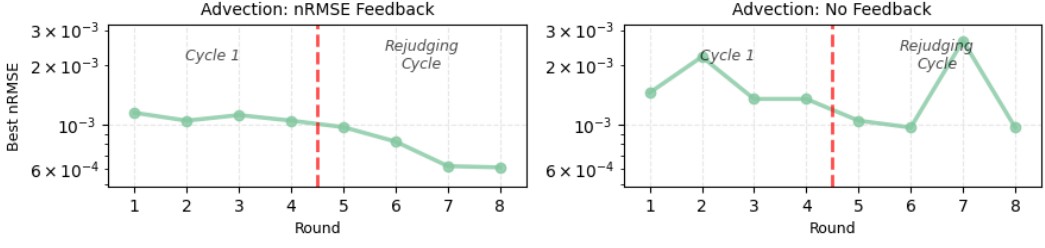

Vertical dashed line separates initial hybridization rounds from rejudging cycle

Figure 5: Without proper feedback, the judges stick to analytical approaches. Figure 18 gives details.

Figures 4, 5, 6 highlight how PDE-SHARP's **Analysis** and **Synthesis** stages leverage mathematical insight and performance feedback, both playing significant roles in PDE-SHARP's performance. Detailed ablation studies in Appendix B.3 quantify each component's contribution in more detail; for instance, Figure 10 shows removing stability analysis increases error by $2 - 8\times$ across PDEs, while disabling tournaments causes $5 - 45\times$ degradation on complex problems like Darcy flow. Table 18 further demonstrates our multi-step Analysis outperforms LLM-planned strategies by $27\times$ on reaction-diffusion, validating our structured approach.

### 4.1.1 CODE QUALITY & INSIGHTS

Figure 7 demonstrates PDE-SHARP reduces the number of debugging iterations required and produces solvers with competitive execution times. PDE-SHARP averages 0.33 debugging iterations per solver execution (approximately 1 in 3 generated solvers requires debugging in a hybridization

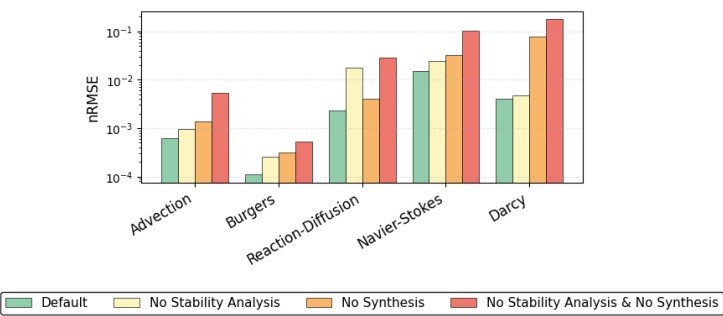

Figure 6: Ablation study of PDE-SHARP components across five PDE tasks. Four variants: (1) Default: full PDE-SHARP with both stability analysis and synthesis, (2) No Stability Analysis: PDE-SHARP with the stability analysis step removed from the Analysis stage, (3) No Synthesis: PDE-SHARP with best-of-32 sampling instead of the Synthesis stage, and (4) No Stability Analysis & No Synthesis. Results show both components contribute to accuracy improvements, with each component being more critical for different PDE types, e.g. stability analysis is more critical for reaction-diffusion, while synthesis contributes more to the Darcy flow task.

round), significantly outperforming baseline methods that require 0.9–1.4 debugging iterations per generated solver. This reduction occurs as a result of PDE-SHARP's Analysis stage producing more robust initial implementations and that the Synthesis stage efficiently eliminates implementation errors.

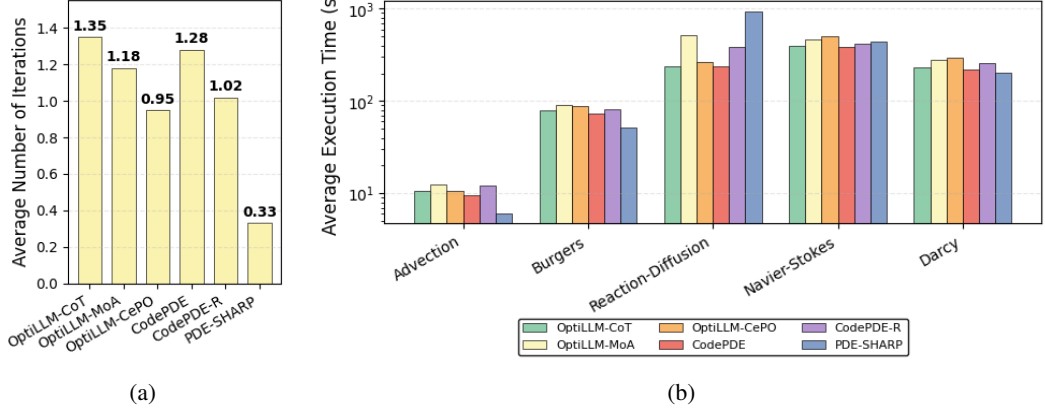

(a)                                          (b)

Figure 7: (a) Average number of debugging iterations required per solver execution across different methods. (b) Average execution times across PDE tasks. PDE-SHARP achieves lower execution times than the average baseline in 4/5 cases. For reaction-diffusion, higher execution time reflects the rigorous numerical methods selected by stability analysis as expected, which produce significantly higher accuracy solvers (Table 2).

**Convergence & Library Usage.** Figure 4b demonstrates the distribution of empirical convergence orders (as defined in Appendix A) — showing solver improvements with grid refinement — across methods for the advection PDE. PDE-SHARP generates solvers with superior convergence properties, leading to higher accuracy in this case (Table 2). In addition, Figure 17 indicates that on average, PDE-SHARP's solvers use less PyTorch (down to ≈25–33 % of library calls) and more SciPy + NumPy + JAX (up to ≈60–75 %), whereas the baselines keep PyTorch at roughly 50–67% and SciPy below 7% on average. Using JAX for computational kernels is highly encouraged in PDE-SHARP prompts in particular as evident in the library usage proportions across all methods and PDE tasks. Additional empirical convergence rate results all PDEs as well as library usage proportions for each baseline appear in Appendix B.4.

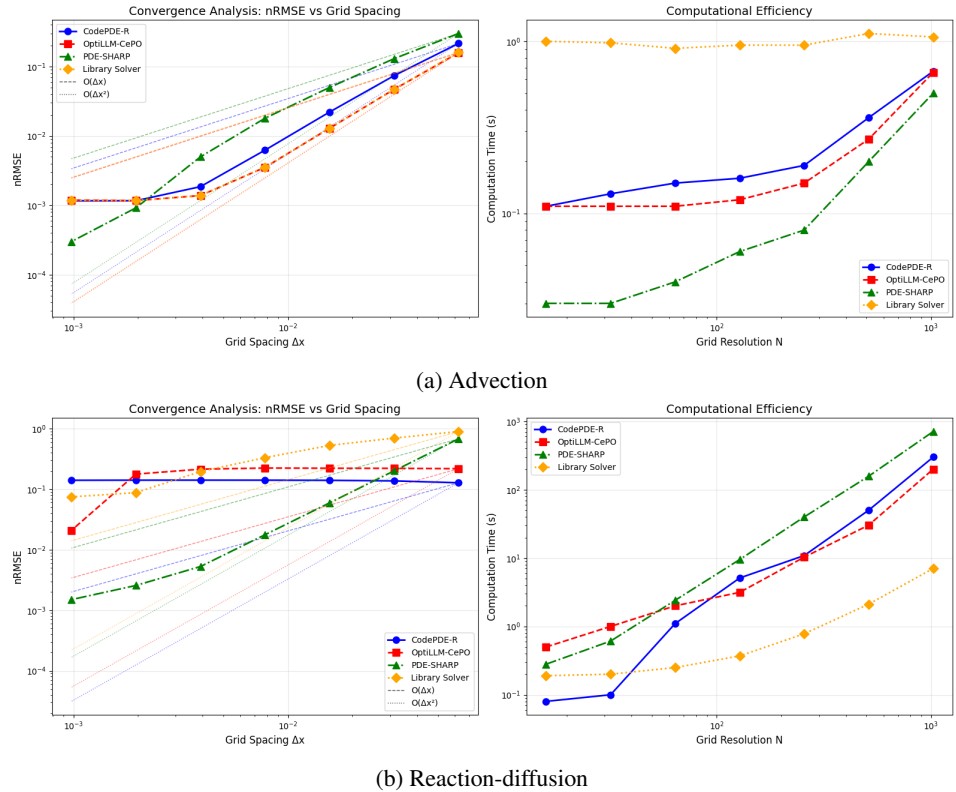

(a) Advection

(b) Reaction-diffusion

Figure 8: Convergence rate analysis and execution times for the advection and reaction-diffusion solvers generated by PDE-SHARP and the baselines. "Library Solver" indicates LLM-driven code (Claude Code used as a strong proxy) that calls high level PDE solver libraries based on each task. Details appear in Appendix B.2.

**Cost.** We analyze the efficiency and cost of each method by calculating the average cost for GPU and LLM API calls for the experiments in this section. Table 2 shows among the tested LLMs, GPT-4o as the code generation LLM yields higher accuracy results on average. Table 3 shows the total average API cost of the results for GPT-4o in Table 2. Details of the calculations appear in Appendix A.4. GPU usage depends on the number of solver executions, code complexity, and implementation efficiency. The number of solver executions for PDE-SHARP depends on the number of hybridization rounds required, averaging 13.2 evaluations across all test cases (9-12 evaluations for most PDEs, with advection requiring 24 to better match data as discussed in Section 4.1). Figure 9 shows nRMSE vs. total average cost (API call + GPU usage) for three PDE tasks.

Table 3: Average cost comparison per method using GPT-4o as the code-generating LLM. **$ Avg. Inputs** and **$ Avg. Output** show the cost of LLM API tokens. **$ Avg. API** denotes their sum. **$ Avg. Total (GPU + API)** is the comprehensive cost, incorporating GPU compute for the solver executions required by each framework. PDE-SHARP achieves a favorable total cost by significantly reducing the number of solver evaluations (see Table 21, Appendix B.3.7), offsetting its API usage.

| Framework | $ Avg. Inputs | $ Avg. Output | $ Avg. API | $ Avg. Total (GPU + API) |
|---|---|---|---|---|
| OptiLLM-CoT | 0.10 | 0.48 | 0.58 | 3.13 |
| OptiLLM-MoA | 0.53 | 2.12 | 2.65 | 6.89 |
| OptiLLM-CePO | 0.96 | 8.27 | 9.23 | 8.71 |
| CodePDE | 0.07 | 0.68 | 0.75 | 3.78 |
| CodePDE-R | 0.41 | 0.88 | 1.29 | 4.89 |
| PDE-SHARP | 1.12 | 2.89 | 4.01 | 5.57 |

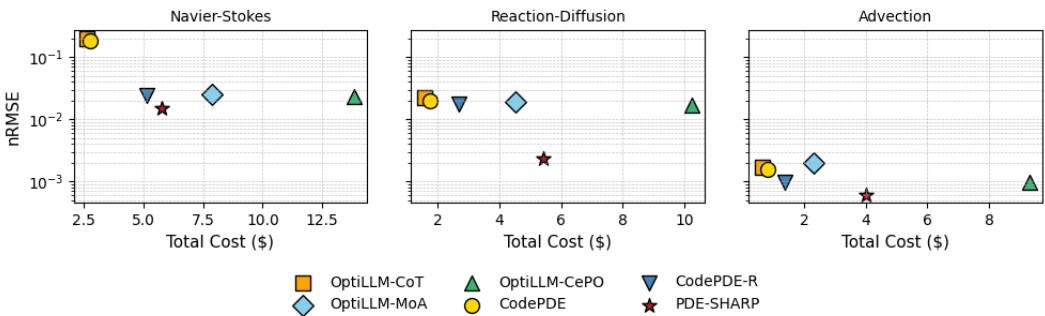

Figure 9: Trade-off between solution accuracy (nRMSE) and total cost for LLM-driven PDE solver generation methods across three PDE tasks of varying computational complexity. From Navier-Stokes (hours per solver evaluation) to Reaction-Diffusion (moderate) to Advection (lightweight, seconds per evaluation), PDE-SHARP demonstrates consistent cost-effectiveness.

## 4.2 DISCUSSION & LIMITATIONS

**Discussion:** PDE-SHARP uses numerical feedback to improve the generated solver. This extra information can be easy to compute — such as the (data-free) PDE residual — or may require collecting data, such as distance to the solution at a sampled set of times and locations. PDE-SHARP can also use problem-specific constraints like the CFL condition (LeVeque, 2007) as feedback, or can run without feedback if no information is available. Results for PDE-SHARP using residual feedback and no numerical feedback appear in Appendix B.3. LLM agents can also suggest feedback types. As seen in Appendix B.3.6 (examples of LLM-suggested feedback types for each tested PDE), an additional LLM agent could be used to determine optimal problem-specific metrics before Synthesis begins. This is particularly beneficial for complex PDEs requiring specialized feedback, and represents important future work. Additional promising directions include scaling to higher-dimensional problems with complex geometries where traditional numerical methods face greater challenges. Finally, hybrid approaches combining PDE-SHARP's interpretable numerical solvers with neural PDE methods could leverage the strengths of both paradigms for problems requiring both accuracy and computational efficiency.

**Limitations:** Our evaluation establishes PDE-SHARP's effectiveness on moderate-complexity PDEs from established benchmarks, with high-fidelity computational simulations representing a natural extension constrained by current LLM training data coverage. LLM-driven PDE solver generation using test-time computing approaches rely on LLM mathematical reasoning capabilities, which means performance may degrade for cutting-edge PDE formulations that are not well-represented in training data or require highly specialized domain knowledge beyond current model capabilities.

## 5 CONCLUSION

PDE-SHARP demonstrates that intelligent LLM-driven solver generation can dramatically improve efficiency over brute-force sampling approaches. Our three-stage framework reduces computational evaluations by 60-75% while achieving superior accuracy on average across five representative PDEs. The mathematical chain-of-thought analysis in the Analysis stage produces more robust initial implementations, requiring on average 67% fewer debugging iterations compared to baseline methods. The hybrid tournaments in the Synthesis stage efficiently refines solvers through performance-informed feedback, with flexible type, demonstrating consistent robust improvements across diverse LLM models.

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

APPENDIX

# A   ADDITIONAL INFORMATION & EXPERIMENTAL SETUPS

## A.1   MATHEMATICAL METRICS

**nRMSE.**   For $S$ test cases, each with true solution $u^{(s)}(x,t)$ and solver prediction $\hat{u}^{(s)}(x,t)$:

$$\text{nRMSE} = \frac{1}{S} \sum_{s=1}^{S} \frac{\|u^{(s)}(x,t) - \hat{u}^{(s)}(x,t)\|_2}{\|u^{(s)}(x,t)\|_2}$$

where $\|\cdot\|_2$ denotes the L2 norm. This metric normalizes the root mean squared error by the magnitude of the true solution, enabling fair comparison across problems with different solution scales.

**Convergence Rate.**   To evaluate numerical correctness, we assess solver convergence behavior across multiple grid resolutions. A robust solver should exhibit predictable error reduction following $E(h) \approx Ch^p$, where $E(h)$ is the solution error on grid spacing $h$, $C$ is a problem-dependent constant, and $p$ is the convergence order.

We estimate the empirical convergence order using two grid resolutions:

$$p \approx \frac{\log\left(\frac{E(h_1)}{E(h_2)}\right)}{\log\left(\frac{h_1}{h_2}\right)}$$

For each generated solver, we evaluate performance on progressively refined grids (typically $h$, $h/2$, $h/4$) and compute the average convergence order. Expected theoretical orders vary by numerical method: first-order schemes ($p \approx 1$), second-order finite difference/volume methods ($p \approx 2$), and spectral methods (exponential convergence). Most LLM-generated solvers achieve first-order convergence, with occasional higher-order behavior depending on the chosen discretization scheme and implementation quality.

## A.2   NEURAL NETWORKS & FOUNDATION MODELS

**Limitations of Cross-Paradigm Comparisons.**   Direct comparison between LLM-generated solvers using traditional numerical methods and neural PDE solvers involves inherent methodological challenges. Neural network baselines are drawn from prior literature with different experimental conditions while our LLM approach benefits from extensive inference-time optimization (debugging, refinement, best-of-N sampling) not applied to these baselines. Additionally, the computational budgets differ fundamentally: neural methods require training time and data preparation, while numerical methods require implementation and parameter tuning effort. These paradigmatic differences make it difficult to establish truly equivalent experimental conditions. Our results should be interpreted as demonstrating the promise of LLM-based solver generation rather than definitive superiority over alternative approaches. Future work should focus on controlled comparisons with standardized evaluation protocols across all methods.

We thus include the following reported numbers verbatim from the original papers of FNO (Li et al., 2020), PirateNets (Wang et al., 2024b), PINNsFormer (Zhao et al., 2023), and UPS (Shen et al., 2024) as examples of neural and foundation models only for the sake of completeness and to give readers an at-a-glance sense of scale (parameters, memory, time/epoch) and accuracy on overlapping PDE families, however, as each method utilizes distinct settings, we do not provide a direct ranking between them. The following is intended only to document the resource scale and the published accuracy on broadly overlapping PDE families.

**FNO** Reports results for 1D Burgers and 2D Navier–Stokes (space–time operator learning). Hardware noted by the authors: single NVIDIA V100 16 GB.

Table 4: FNO on *1D Burgers* (relative $\ell_2$ error at different spatial resolutions $s$).

| Method | $s$=256 | 512 | 1024 | 2048 | 4096 | 8192 |
|---|---|---|---|---|---|---|
| FNO | 0.0149 | 0.0158 | 0.0160 | 0.0146 | 0.0142 | 0.0139 |

*Notes.* Table reproduced from the paper; parameters, GPU memory, and time/epoch were not reported for the Burgers experiment. See Table 5 for Navier–Stokes resource numbers as reported by the authors.

Table 5: FNO on *2D Navier–Stokes* (relative $\ell_2$ error over different viscosities $\nu$ and dataset sizes $N$; per-epoch time reported by the authors).

| Method | Params | Time/epoch | $\nu$=$10^{-3}$, $T$=50, $N$=1000 | $\nu$=$10^{-4}$, $T$=30, $N$=1000 | $\nu$=$10^{-4}$, $T$=30, $N$=10000 | $\nu$=$10^{-5}$, $T$=20, $N$=1000 |
|---|---|---|---|---|---|---|
| FNO-3D | 6,558,537 | 38.99 s | 0.0086 | 0.1918 | 0.0820 | 0.1893 |
| FNO-2D | 414,517 | 127.80 s | 0.0128 | 0.1559 | 0.0973 | 0.1556 |
| U-Net | 24,950,491 | 48.67 s | 0.0245 | 0.2051 | 0.1190 | 0.1982 |
| TF-Net | 7,451,724 | 47.21 s | 0.0225 | 0.2253 | 0.1168 | 0.2268 |
| ResNet | 266,641 | 78.47 s | 0.0701 | 0.2871 | 0.2311 | 0.2753 |

*Notes.* Reported at $64\times64$ spatial resolution; FNO-3D convolves in space–time while FNO-2D uses 2D convolutions with an RNN in time.

**PirateNets** has PINN backbone with physics-informed residual adaptive blocks. The paper emphasizes accuracy comparisons and ablations; it does not tabulate parameter counts, GPU memory, or wall-clock per epoch. Below we list the state-of-the-art test errors the authors report.

Table 6: PirateNets: reported relative $\ell_2$ test errors across PDEs (paper's Table 1).

| Benchmark | Error (PirateNet) | Params | GPU Mem | Time/epoch |
|---|---|---|---|---|
| Allen–Cahn (1D) | $2.24\times10^{-5}$ | — | — | — |
| Korteweg–De Vries (1D) | $4.27\times10^{-4}$ | — | — | — |
| Grey–Scott (2D) | $3.61\times10^{-3}$ | — | — | — |
| Ginzburg–Landau (2D) | $1.49\times10^{-2}$ | — | — | — |
| Lid-driven cavity (2D) | $4.21\times10^{-2}$ | — | — | — |

*Notes.* Architecture details (e.g., depth/width) and training pipelines are provided, but resource metrics are not tabulated.

**PINNsFormer** is a transformer-style PINN variant. The authors report parameter counts and training overhead (V100), and test errors on overlapping 1D PDEs.

Table 7: PINNsFormer: model size and training overhead (Appendix Table 4–5 in the paper).

| Model | Params | GPU Mem (MiB) | Time/epoch (s) |
|---|---|---|---|
| PINNsFormer (pseudo-seq. length $k$=5) | 454,000 | 2,827 | 2.34 |

*Notes.* Reported on a single NVIDIA Tesla V100; overheads shown for $k$=5.

Table 8: PINNsFormer: reported test errors on 1D PDEs used widely in PINN literature.

| PDE (dimension) | Metric (paper) | Error | Params | Time/epoch / GPU Mem |
|---|---|---|---|---|
| Convection (1D) | rRMSE ($\approx$ rel. $\ell_2$) | 0.027 | 454k | 2.34 s / 2,827 MiB |
| Reaction (1D) | rRMSE ($\approx$ rel. $\ell_2$) | 0.030 | 454k | 2.34 s / 2,827 MiB |

*Notes.* Errors are taken directly from the paper's main results tables; rRMSE is the paper's standard relative $\ell_2$ metric. The reaction/convection formulations and sampling follow the setups specified in Zhao et al. (2023).

**UPS** learns to map symbolic PDE specifications and initial/boundary conditions to numerical solutions. The architecture combines Fourier Neural Operators and transformers with autoregressive decoding over space-time grids.

The model was trained on $\sim$20k PDE trajectories using a single NVIDIA A6000 GPU. Training was run for 60,000 steps and completed in under 100 GPU-hours. UPS achieves strong sample efficiency, outperforming baselines with $4\times$ less data and $26\times$ less compute.

Table 9: UPS: test errors on PDEBench benchmarks (relative $\ell_2$ or nRMSE as reported).

| PDE | Metric | Error (UPS) | Training Steps | GPU | Total GPU Hours |
|---|---|---|---|---|---|
| Advection (1D) | nRMSE | $2.20\times10^{-3}$ | 60,000 | A6000 | ¡100 |
| Burgers (1D) | nRMSE | $3.73\times10^{-2}$ | 60,000 | A6000 | ¡100 |
| Reaction–Diffusion (2D) | nRMSE | $5.57\times10^{-2}$ | 60,000 | A6000 | ¡100 |
| Navier–Stokes (2D) | nRMSE | $4.50\times10^{-3}$ | 60,000 | A6000 | ¡100 |

*Notes.* Errors and training configuration are from the paper's PDEBench experiments. Training used $\sim$20k PDE samples across equations; GPU time and steps refer to total training, not per-PDE.

## A.3 LLM-DRIVEN ARCHITECTURES

### A.3.1 LLM MODELS USED IN SECTION 4 FOR CODE GENERATION

Table 10: LLM models used in Section 4 for solver generation; more LLMs – including the coding and math-aware variants of these – are tested in Appendix B.1

| LLM | Type | Access |
|---|---|---|
| Gemma 3 | Non-reasoning | Open Source |
| LLama 3.3 | Non-reasoning | Open Source |
| Qwen3 | Non-reasoning | Open Source |
| DeepSeek-R1 | Reasoning | Open Source |
| GPT-4o | Non-reasoning | API Service |
| o3 | Reasoning | API Service |

### A.3.2 AGENTIC WORKFLOWS

Frameworks like FunSearch (Romera-Paredes et al., 2023) and AIDE (Jiang et al., 2025b) wrap an LLM in an iterative search/refinement loop. They treat the LLM as an agent that can branch, try multiple approaches, and refine code via feedback.

**FunSearch (DeepMind, 2023)** pairs a pre-trained code-generating LLM with an automated evaluator in a loop. The LLM proposes candidate programs/solutions, an evaluator (a test or objective function) checks them, and then the process generates new candidates (mutations, combinations) based on feedback. FunSearch features algorithm discovery based on a program database. The program database consists of a few "islands" of programs. The experimental setup is the same as (Li et al., 2025). The number of islands is set to 4 and the island reset period to 3600s. The FunSearch process runs for 32 iterations. In each iteration, the language model decoding temperature is set to 0.7.

**AIDE (Weco AI, 2025)** formulates code generation as a tree search problem. For a given high-level task (like "build an ML pipeline that achieves X accuracy on Y dataset"), AIDE would have the LLM propose a solution. Then it measures how good that solution is (it runs the code and sees accuracy). If not satisfied, AIDE can either refine the current solution (edit some parts of the code via another LLM call) or try a different approach (branch out in the search tree). Over multiple iterations, it explores the space of programs. The experimental setup is the same as (Li et al., 2025). AIDE runs for 96 steps and the max debug depth, debug probability, and number of drafts are set to 5, 0.9, and 24, respectively. The language model decoding temperature is set to 0.5 for code generation following the original paper (Jiang et al., 2025b).

Table 11: nRMSE values for Agentic Workflows on different PDEs. Results from Li et al. (2025)

|  | Advection | Burgers | Reaction-Diffusion | Navier-Stokes | Darcy |
|---|---|---|---|---|---|
| AIDE | 1.03e-3 | 1.05e-4 | 5.07e-2 | 5.77e-2 | 4.78e-3 |
| FunSearch | 1.05e-3 | 1.13e-4 | 3.72e-2 | 5.86e-2 | 4.78e-3 |

### A.3.3 OTHER RELATED WORK

Recent work Soroco et al. (2025) introduces PDE-Controller, a framework that fine-tunes LLMs specifically for PDE control problems. Their approach trains specialized models for autoformalization (converting natural language to formal specifications), program synthesis, and multi-step reasoning through reinforcement learning from human feedback (RLHF). While demonstrating strong performance on their target domains, this approach differs from PDE-SHARP in several key aspects.

Table 12: PDE-Controller: Training Requirements and Performance

| Metric | Value |
|---|---|
| **Training Data** | |
| Heat equation samples | 867,408 |
| Wave equation samples | 845,088 |
| Total training samples | 1,712,496 |
| **Evaluation Data** | |
| Synthetic test samples | 426,432 |
| Manual test problems | 34 |
| **Performance (Synthetic)** | |
| Autoformalization accuracy (IoU) | 99.2% |
| Code executability | 97.99% |
| **Performance (Manual)** | |
| Autoformalization accuracy (IoU) | 68.0% |
| Code executability | 91.2% |
| **Scope** | |
| PDE types covered | 2 (heat, wave) |
| Spatial dimensions | 1D |

While effective for specific classes of PDEs, the fine-tuning approach presents several limitations compared to LLM-driven approaches using test-time computing: **(1) Computational overhead:** Requires extensive fine-tuning of multiple specialized models (translator, controller, coder) with over 1.7M training samples; **(2) Domain specificity:** Limited to only heat and wave equations in 1D, requiring retraining for new PDE types; **(3) Data requirements:** Needs large-scale synthetic data generation and manual curation by domain experts; **(4) Scalability constraints:** Each new PDE family would require collecting new training data and retraining models; **(5) Generalization gap:** Performance drops significantly on manual problems (99.2% to 68.0% accuracy), indicating limited robustness to real-world variations.

PDE-SHARP offers more flexibility across PDE types without domain-specific training, though potentially at the cost of specialized performance on specific equation families. The fundamental trade-off lies between the specialized efficiency of fine-tuned approaches versus the broader applicability and reduced computational overhead of general prompting strategies.

### A.3.4 OPTILLM

We use the OptiLLM framework from `github.com/codelion/optillm` as a baseline to test PDE-SHARP. OptiLLM is an optimizing inference proxy that implements 20+ state-of-the-art techniques to improve LLM accuracy and performance on reasoning tasks without requiring any model training or fine-tuning. We test three of OptiLLM's implemented techniques in our study.

**CoT (Chain-of-Thought) with Reflection.** Implements chain-of-thought reasoning with structured <thinking>, <reflection> and <output> sections to enhance reasoning quality through explicit self-evaluation. The approach generates intermediate reasoning steps in the thinking phase, critically reviews the reasoning in the reflection phase, and produces the final output, enabling improved accuracy on complex reasoning tasks without requiring model fine-tuning.

**MoA (Mixture-of-Agents).** Combines responses from multiple model critiques in a collaborative framework where 3 different agent perspectives are aggregated to produce higher-quality solutions.

**CePO (Cerebras Planning and Optimization).** Combines Best-of-$n$ sampling (without code execution), Chain-of-Thought reasoning, Self-Reflection, and Self-Improvement in a four-stage process: plan generation with confidence scoring, initial solution development, plan refinement through inconsistency analysis, and final solution production. The method applies Best-of-$n$ to multiple solution candidates with optional plan diversity, using parameters like `planning_n` proposals and `planning_m` maximum attempts to generate robust solutions for complex reasoning tasks. The following are the default parameters used in this study.

Table 13: Default configuration values for CePO planning and verification stages

| Parameter | Description | Default Value |
|---|---|---|
| `--cepo_bestofn_n` | Number of responses to be generated in best of n stage | 3 |
| `--cepo_bestofn_temperature` | Temperature for verifier in best of n stage | 0.1 |
| `--cepo_bestofn_max_tokens` | Max tokens for verifier in best of n stage | 4096 |
| `--cepo_bestofn_rating_type` | Rating type ("absolute" or "pairwise") | `"absolute"` |
| `--cepo_planning_n` | Number of plans generated in planning stage | 3 |
| `--cepo_planning_m` | Attempts to generate n plans in planning stage | 6 |
| `--cepo_planning_temperature_step1` | Temperature in step 1 of planning stage | 0.55 |
| `--cepo_planning_temperature_step2` | Temperature in step 2 of planning stage | 0.25 |
| `--cepo_planning_temperature_step3` | Temperature in step 3 of planning stage | 0.1 |
| `--cepo_planning_temperature_step4` | Temperature in step 4 of planning stage | 0 |
| `--cepo_planning_max_tokens_step1` | Max tokens in step 1 of planning stage | 4096 |
| `--cepo_planning_max_tokens_step2` | Max tokens in step 2 of planning stage | 4096 |
| `--cepo_planning_max_tokens_step3` | Max tokens in step 3 of planning stage | 4096 |
| `--cepo_planning_max_tokens_step4` | Max tokens in step 4 of planning stage | 4096 |
| `--cepo_print_output` | Whether to print the output of each stage | `False` |
| `--cepo_config_file` | Path to CePO configuration file | `None` |
| `--cepo_use_plan_diversity` | Use additional plan diversity step | `False` |
| `--cepo_rating_model` | Rating model (if different from completion) | `None` |

### A.3.5 CODEPDE ( (LI ET AL., 2025))

**CodePDE.** CodePDE is an inference framework for LLM-driven PDE solver generation that frames PDE solving as a code generation task. The framework operates through a five-step process: (1) *Task Specification* converts PDE problems into natural language descriptions including governing equations, domain specifications, boundary conditions, and initial conditions; (2) *Code Generation* uses chain-of-thought prompting to instruct models to generate complete solver implementations with predefined function signatures; (3) *Debugging* performs iterative self-debugging for up to 4 rounds when solvers encounter execution errors, feeding error traces back to the LLM for autonomous correction; and (4) *Evaluation* assesses solver performance using normalized root mean squared error (nRMSE), convergence tests, and execution time; For our comparison, we use Code-PDE with the same setup as (Li et al., 2025) with steps 1-4 (reasoning + debugging), generating 32 solver samples with best-of-32 selection, using up to 4 debugging iterations per solver.

**CodePDE-R.** CodePDE-R extends the base CodePDE framework by incorporating the solver refinement step (step 5). This variant selects the 5 best-performing programs from the reasoning + debugging stage as "seed" programs for refinement. The refinement process provides the nRMSE obtained during evaluation along with the solver implementation back to the LLM, instructing it to analyze execution results, identify numerical instabilities and bottlenecks, and generate improved implementations accordingly. For each seed program, the framework generates 4 refined versions across different refinement configurations (using 3, 4, or 5 seed implementations), resulting in 12 refined programs total. The final result reports the best nRMSE among these 12 refined samples. This

iterative feedback-driven optimization enables models to systematically improve solver accuracy and efficiency beyond the initial generation and debugging phases.

## A.4 ADDITIONAL INFORMATION ON FRAMEWORK COST

Table 3 shows the average API call cost for each framework using GPT-4o as the code generator LLM. GPT-4o input cost is $2.50 per 1M tokens, and the output cost is $10.00 per 1M tokens. Table 14 shows the average input-output counts for each framework from Section 4. An NVIDIA T4 GPU costs $0.35 per hour, which is used to calculate the total average costs in Figure 9.

Table 14: Approximation of the total input-output counts for running each framework once

| Framework | # Inputs | # Output |
|---|---|---|
| OptiLLM (CoT) | 48,000 | 105,600 |
| OptiLLM (MoA) | 200,000 | 422,400 |
| OptiLLM (CePO) | 600,000 | 105,600 |
| CodePDE | 102,400 | 294,400 |
| PDE-SHARP | 600,000 | 450,800 |

## B ADDITIONAL EXPERIMENTAL RESULTS

### B.1 RESULTS WITH DIFFERENT LLMS

The following additional LLM models are tested for code generation in addition to the results of Table 2.

Table 15: Additional LLMs

| LLM | Type | Access |
|---|---|---|
| Qwen3-Coder  (Team, 2025) | Coding-specific | Open Source |
| Code Llama  (Rozière et al., 2024) | Coding-specific | Open Source |
| GPT-5 | Non-reasoning | API Service |
| DeepSeekMath  (Shao et al., 2024) | Mathematical reasoning | Open Source |
| DeepSeek-Coder  (Guo et al., 2024) | Coding-specific | Open Source |
| MathCoder2-DeepSeekMath  (Lu et al., 2024) | Math aware Coding-specific | Open Source |

Table 16: nRMSE comparison of the baseline frameworks using different LLMs.

| | | Advection | Burgers | Reaction-Diffusion | Navier-Stokes | Darcy |
|---|---|---|---|---|---|---|
| **OptiLLM-CoT** | Qwen3-Coder | 4.67e-03 | 1.52e-03 | 9.38e-01 | 2.63e-01 | 6.34e-01 |
| | GPT-5 | 5.36e-03 | 1.88e-03 | 1.04e+00 | 2.83e-01 | 7.18e-01 |
| | DeepSeekMath | 4.89e-03 | 3.12e-04 | 2.38e-01 | 8.51e-02 | 5.22e-03 |
| | DeepSeek-Coder | 4.89e-03 | 3.04e-04 | 2.41e-01 | 8.72e-02 | 5.11e-03 |
| | MathCoder2-DeepSeekMath | 4.89e-03 | 3.27e-04 | 2.43e-01 | 8.66e-02 | 5.29e-03 |
| **OptiLLM-MoA** | Qwen3-Coder | 1.01e-03 | 3.45e-04 | 9.68e-02 | 1.79e-02 | 5.12e-03 |
| | GPT-5 | 4.18e-03 | 4.11e-04 | 1.14e-01 | 2.02e-02 | 1.89e-02 |
| | DeepSeekMath | 1.32e-03 | 2.66e-04 | 3.57e-02 | 1.72e-02 | 5.23e-03 |
| | DeepSeek-Coder | 1.32e-03 | 3.04e-04 | 1.55e-01 | 1.78e-02 | 5.18e-03 |
| | MathCoder2-DeepSeekMath | 1.01e-03 | 2.66e-04 | 4.07e-02 | 1.74e-02 | 5.22e-03 |
| **OptiLLM-CePO** | Qwen3-Coder | 1.01e-03 | 3.23e-04 | 8.91e-02 | 1.97e-02 | 1.83e-02 |
| | GPT-5 | 3.17e-03 | 3.89e-04 | 1.03e-01 | 2.24e-02 | 4.72e-02 |
| | DeepSeekMath | 9.98e-04 | 2.55e-04 | 2.45e-02 | 1.85e-02 | 4.92e-03 |
| | DeepSeek-Coder | 1.01e-03 | 2.66e-04 | 1.47e-01 | 1.91e-02 | 4.92e-03 |
| | MathCoder2-DeepSeekMath | 9.98e-04 | 3.04e-04 | 3.56e-02 | 1.93e-02 | 4.33e-03 |
| **CodePDE** | Qwen3-Coder | 4.89e-03 | 1.35e-03 | 9.55e-01 | 2.59e-01 | 6.57e-01 |
| | GPT-5 | 5.75e-03 | 1.63e-03 | 1.08e-01 | 2.82e-01 | 7.91e-01 |
| | DeepSeekMath | 5.10e-03 | 2.87e-04 | 2.45e-02 | 7.91e-02 | 4.97e-03 |
| | DeepSeek-Coder | 4.69e-03 | 2.87e-04 | 2.78e-01 | 7.82e-02 | 5.02e-03 |
| | MathCoder2-DeepSeekMath | 5.10e-03 | 3.15e-04 | 2.32e-02 | 7.84e-02 | 4.97e-03 |
| **CodePDE-R** | Qwen3-Coder | 9.74e-04 | 3.60e-04 | 9.13e-02 | 9.67e-02 | 4.90e-02 |
| | GPT-5 | 1.14e-03 | 4.41e-04 | 1.07e-01 | 7.93e-02 | 5.81e-02 |
| | DeepSeekMath | 9.89e-04 | 2.62e-04 | 1.47e-02 | 3.63e-02 | 5.01e-03 |
| | DeepSeek-Coder | 9.89e-04 | 3.15e-04 | 1.47e-02 | 2.67e-02 | 6.01e-03 |
| | MathCoder2-DeepSeekMath | 9.74e-04 | 2.62e-04 | 1.47e-02 | 1.65e-02 | 4.97e-03 |
| **PDE-SHARP** | Qwen3-Coder | 9.74e-04 | 2.97e-04 | 5.39e-03 | 2.80e-02 | 7.80e-03 |
| | GPT-5 | 1.01e-03 | 3.45e-04 | 7.78e-03 | 3.19e-02 | 9.93e-03 |
| | DeepSeekMath | 7.46e-04 | 1.55e-04 | 2.39e-03 | 1.47e-02 | 4.78e-03 |
| | DeepSeek-Coder | 7.46e-04 | 2.53e-04 | 3.67e-03 | 2.76e-02 | 4.78e-03 |
| | MathCoder2-DeepSeekMath | 5.54e-04 | 1.38e-04 | 2.99e-03 | 1.47e-02 | 3.93e-03 |

### B.2 USE OF SOLVER LIBRARIES

Throughout our experiments with LLM-driven frameworks, prompts have explicitly permitted, and encouraged, the use of high-level PDE solver libraries (e.g., FEniCS, PETSc, deal.II, JAX-CFD) for generating solver samples. The decision to invoke library functions or implement custom discretizations was left to the LLM agent's discretion. PDE-SHARP is not opposed to library usage; in fact, many of its generated solvers incorporate established numerical libraries where appropriate. The extent of library usage — including which specific libraries are preferred — can be easily adjusted via the prompts based on the user's needs, available infrastructure, and level of expertise, making the framework highly adaptable to different computational environments. The hybrid tournaments of the

Synthesis stage then refines and selects among these candidates, often preserving implementations that are well-structured, efficient, and numerically robust.

For the PDEs tested in this work, the best-performing solvers (Table 2) were primarily custom implementations refined through hybridization, particularly for the advection and reaction-diffusion equations. These custom solvers often outperformed library-based counterparts by exploiting equation-specific mathematical structure — such as exact analytical integration of the reaction term — that generic library calls do not automatically leverage. Detailed code comparisons and performance analyses for these cases are provided in Appendix D.

For Navier-Stokes and Darcy flow, solvers that called established libraries (notably JAX-CFD) performed similarly to custom implementations, and the final PDE-SHARP-generated solvers for these tasks often incorporated such libraries. This reflects a pragmatic balance: libraries provide efficient, battle-tested discretizations for complex operators (e.g., spectral derivatives, finite-element assembly), while PDE-SHARP's mathematical reasoning guides overall solver architecture and parameter selection.

To further investigate the trade-offs between custom and library-based approaches, we employed Claude Code as a general-purpose coding agent to generate solvers that explicitly rely on high-level libraries. For each PDE, the agent first identified a suitable library (see Table 17) and then constructed a solver by calling appropriate library functions. The experimental setup otherwise matched that of other LLM-driven baselines (best-of-32). Below we briefly characterize the libraries considered:

- **FEniCS/FEniCSx**: A popular finite-element library that automates variational formulation and assembly. It is well-suited for elliptic/parabolic problems with complex geometries but can incur overhead for simple, regular-grid problems.
- **PETSc**: A scalable library for solving large-scale linear and nonlinear equations, often used as a backend for sparse linear algebra in finite-element and finite-difference codes.
- **deal.II**: A C++ library supporting adaptive finite-element methods, particularly effective for problems requiring local mesh refinement.
- **JAX-CFD**: A JAX-based library for computational fluid dynamics that provides differentiable, GPU-accelerated finite-difference and spectral operators, ideal for batch processing and gradient-based optimization.

Our results (Table 17) show that even when library usage is encouraged, PDE-SHARP's refined solvers typically match or exceed the performance of pure library-based implementations. Beyond nRMSE, PDE-SHARP solvers also demonstrate superior numerical properties such as higher convergence rates and more consistent grid convergence compared to library-based baselines (see Appendix B.4). This demonstrates that PDE-SHARP's intelligent synthesis of mathematical insight with existing numerical tools can yield superior outcomes to other LLM-driven PDE solver generation approaches.

Table 17: nRMSE values for Claude Code-generated solvers using PDE solver library calls instead of custom approaches

|  | Advection | Burgers | Reaction-Diffusion | Navier-Stokes | Darcy |
|---|---|---|---|---|---|
| Claude Code + PDE Libraries | 1.03e-3 | 3.05e-4 | 1.17e-2 | 1.72e-2 | 4.80e-3 |

## B.3 PDE-SHARP ABLATION STUDIES

In this section, we present ablation study results on PDE-SHARP. Note that we take the default PDE-SHARP framework to be one used in Section 4. The ablation studies of this section each target a different aspect of PDE-SHARP's design.

### B.3.1 ANALYSIS PROMPTING STRATEGY

We compare the following prompting strategies for the Analysis stage.

- Multi-Step prompting (PDE-SHARP default)
- Single Prompt (all the PDE-SHARP steps merged into one)
- LLM-generated multi-step prompting
- LLM-generated single prompt

For the LLM-generated alternatives, the LLM, GPT-4o in this ablation, is first asked to generate either a series of prompts or a single prompt to run as the analysis stage for a give PDE before proceeding to the code generation stage. The Synthesis stage is done exactly as in Section 4. Table 18 summarizes these results.

Table 18: nRMSE comparison of the baseline frameworks using different Analysis prompting strategies.

|  | | Advection | Burgers | Reaction-Diffusion | Navier-Stokes | Darcy |
|---|---|---|---|---|---|---|
| **Multi-Step Prompting (Default)** | Gemma 3 | 1.01e-03 | 5.60e-04 | 3.01e-03 | 3.14e-02 | 1.72e-02 |
| | LLaMA 3.3 | 9.98e-04 | 4.61e-04 | 3.61e-03 | 5.06e-02 | 1.72e-02 |
| | Qwen 3 | 7.76e-04 | 2.97e-04 | 2.32e-03 | 2.80e-02 | 4.80e-03 |
| | DeepSeek-R1 | 5.24e-04 | 1.48e-04 | 2.29e-03 | 1.37e-02 | 4.74e-03 |
| | GPT-4o | 6.11e-04 | 2.31e-04 | 2.29e-03 | 1.51e-02 | 3.97e-03 |
| | o3 | 9.74e-04 | 3.42e-04 | 5.78e-03 | 1.89e-02 | 7.78e-03 |
| **Single Prompt (Default merged into one)** | Gemma 3 | 1.03e-03 | 4.89e-04 | 1.18e-02 | 4.31e-02 | 8.11e-03 |
| | LLaMA 3.3 | 1.05e-03 | 4.79e-04 | 1.75e-02 | 7.32e-02 | 1.79e-02 |
| | Qwen 3 | 8.01e-04 | 3.11e-04 | 2.41e-03 | 4.94e-02 | 4.91e-03 |
| | DeepSeek-R1 | 6.53e-04 | 1.56e-04 | 2.37e-03 | 1.41e-02 | 4.83e-03 |
| | GPT-4o | 7.39e-04 | 3.48e-04 | 3.33e-03 | 2.62e-02 | 4.13e-03 |
| | o3 | 8.70e-04 | 4.54e-04 | 3.89e-03 | 2.96e-02 | 4.87e-03 |
| **LLM-Generated Multi-Step Prompting** | Gemma 3 | 1.02e-03 | 4.82e-04 | 9.21e-02 | 7.27e-02 | 7.93e-03 |
| | LLaMA 3.3 | 1.04e-03 | 4.72e-04 | 8.69e-02 | 7.24e-02 | 1.77e-02 |
| | Qwen 3 | 1.89e-03 | 6.05e-04 | 3.39e-02 | 3.89e-02 | 4.85e-03 |
| | DeepSeek-R1 | 8.37e-04 | 5.30e-04 | 1.33e-02 | 3.40e-02 | 4.85e-03 |
| | GPT-4o | 7.27e-04 | 4.15e-04 | 1.31e-02 | 2.59e-02 | 4.05e-03 |
| | o3 | 6.96e-04 | 7.48e-04 | 1.84e-02 | 3.93e-02 | 4.85e-03 |
| **LLM-Generated Single Prompt** | Gemma 3 | 1.04e-03 | 4.95e-04 | 1.29e-01 | 5.42e-02 | 8.19e-03 |
| | LLaMA 3.3 | 1.06e-03 | 6.87e-04 | 1.81e-01 | 6.43e-02 | 1.81e-02 |
| | Qwen 3 | 1.13e-03 | 6.19e-04 | 8.47e-02 | 3.98e-02 | 3.95e-03 |
| | DeepSeek-R1 | 9.59e-04 | 4.95e-04 | 1.39e-02 | 4.43e-02 | 4.85e-03 |
| | GPT-4o | 2.47e-03 | 7.22e-04 | 2.36e-02 | 3.65e-02 | 4.85e-03 |
| | o3 | 9.19e-04 | 7.48e-04 | 3.91e-02 | 3.01e-02 | 5.92e-03 |

Our experiments demonstrate that the Multi-Step Prompting strategy consistently yields the best performance across all LLMs and PDEs. When all the PDE-SHARP Analysis prompts are merged together into a single prompt, LLMs tend to not follow the instructions thoroughly as they become too long to follow. Moreover, when the LLM is tasked with generating the prompts for the analysis stage, it is observed that many details, such as checking for hybrid approaches or doing a rigorous numerical stability analysis is overlooked. Analyzing the strategies used in the generated solvers (Table 18) for the reaction-diffusion task is a great demonstration of this shortcoming as reaction diffusion is more sensitive to method choice and stability analysis (Figure 19). Naturally, the most pronounced impact is observed on the Reaction-Diffusion PDE, where the default multi-step approach achieves the lowest average nRMSE of 2.88e-03 across all LLMs. In contrast, the average nRMSE rises to 6.88e-03 with Single Prompting, 4.30e-02 with LLM-Generated Multi-Step Prompting, and peaks at 7.86e-02 with LLM-Generated Single Prompting. This corresponds to a $27\times$ increase in error from the best case to the worst, highlighting the critical role of well-structured multi-step analysis in improving solution accuracy for complex PDEs.

### B.3.2 THE EFFECTS OF STABILITY ANALYSIS

To evaluate the individual contributions of PDE-SHARP's key components — the stability analysis in the Analysis stage and the tournaments in the Synthesis stage — we conduct an ablation study examining four variants: (1) the default framework with both mathematical stability analysis and tournaments, (2) tournaments without stability analysis, (3) stability analysis without tournaments (best-of-32 sampling with stability analysis), and (4) neither component (best-of-32 sampling without stability analysis). Figure 10 demonstrates that mathematical stability analysis provides substantial accuracy improvements across all tested PDEs. Removing stability analysis while maintaining tournaments increases average nRMSE by $2\text{-}8\times$ depending on the PDE complexity. The

tournaments component shows mixed but generally positive effects, with the largest improvements observed for reaction-diffusion and Darcy flow problems. Most critically, removing both components results in significant performance degradation, with nRMSE increases of 5-45$\times$ for complex PDEs like Darcy flow. These results confirm that PDE-SHARP's mathematical analysis stage is essential for generating numerically stable solvers, while the tournament-based refinement provides additional accuracy gains particularly for challenging nonlinear problems.

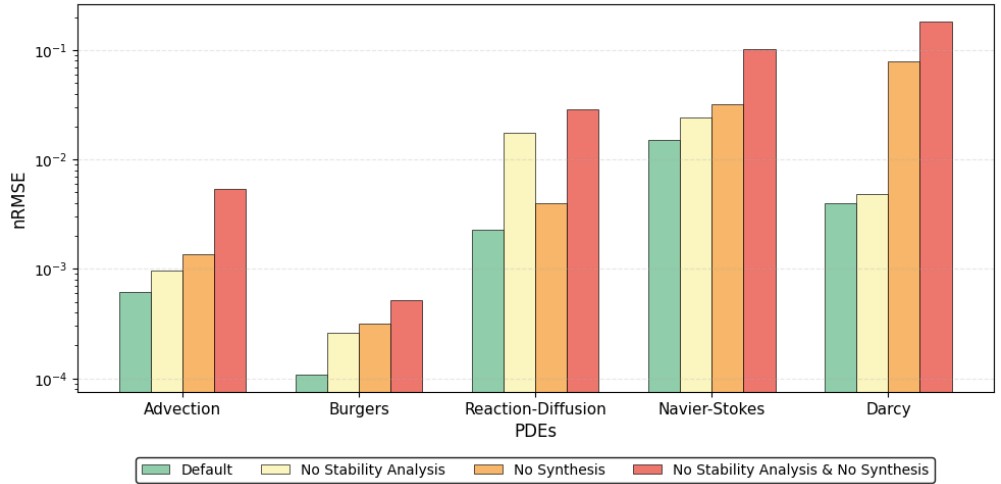

Figure 10: Ablation study of PDE-SHARP components across five PDE tasks. Results show that mathematical stability analysis is critical for solver accuracy, while tournaments provide additional improvements. Removing both components leads to significant performance degradation, particularly for complex PDEs like Darcy flow.

The stability analysis component of PDE-SHARP plays a crucial role in guiding solver strategy selection. Figure 20 illustrates the percentage of hybrid analytical-numerical versus purely numerical approaches chosen by each PDE-SHARP variant for the reaction-diffusion equation. The default framework and the variant without tournaments both achieve 100% hybrid approach selection, demonstrating that mathematical stability analysis consistently identifies the superiority of hybrid methods for this PDE. In contrast, removing stability analysis results in predominantly numerical approaches (87-93%), as the framework lacks the mathematical insight to recognize that the reaction component admits an analytical solution. This strategic difference directly explains the accuracy improvements observed in the previous ablation study, as hybrid approaches achieve superior numerical stability and precision for reaction-diffusion problems.

### B.3.3 REASONING VS. NON-REASONING LLMS FOR CODE GENERATION IN GENESIS

Experiments indicate that in PDE-SHARP, there is negligible difference between the final results using reasoning, non-reasoning, coding-specific, and mathematical LLM models (Tables 10 & 15) as the code generator in the Genesis stage. See Tables 2 and 16 for nRMSE results.

### B.3.4 TEST-TIME SCALING FOR PDE-SHARP

Based on our test-time scaling study (Figure 11) for PDE-SHARP and to be consistent with findings from (Li et al., 2025) on the same PDE tasks, we use $N = 32$ initial solver candidates in our experiments. This choice balances computational efficiency with sufficient diversity for effective solver selection in the subsequent Synthesis stage.

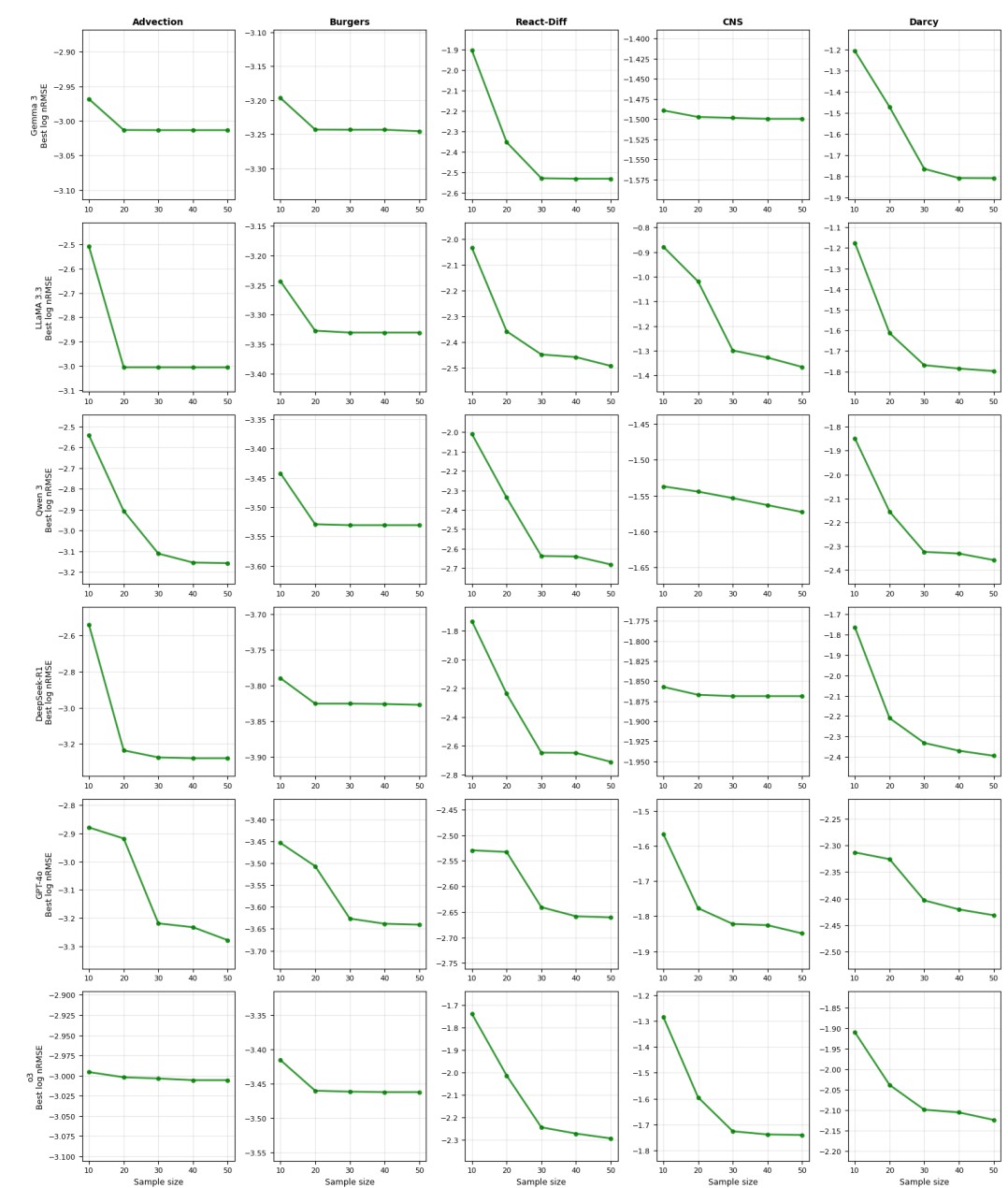

Figure 11: Varying the number of solver samples generated for each LLM and each PDE family in PDE-SHARP.

### B.3.5 STRUCTURE OF THE TOURNAMENTS

In this ablation study, we keep the default PDE-SHARP strategy from Section 4 for the Analysis and Genesis stages and replace the Synthesis stage with various strategies to study its effectiveness. In PDE-SHARP's default Synthesis stage in Section 4, three LLM instances, which we call "judges", are tasked with the selection and hybridization tournaments. To achieve the best performance (Table 2) — i.e. fewer tournament rounds to get the highest performing PDE solver codes — these three judges are taken to be a mixture of reasoning and non-reasoning LLMs (o3, DeepSeek-R1, and GPT-4o) in Section 4. This set of LLM judges are chosen to balance efficient code generation and code stability details with the detailed reasoning and attention to numerical implementation details that the reasoning models bring in. In this section, we consider other possibilities for the three judges

to justify our choice of LLM judges. Tables 2 and 16 demonstrate that using different LLM models to generate 32 samples of solver codes leads to overall negligible difference in the final results in PDE-SHARP as the tournaments lead to solvers robust to LLM choice. Thus, we stick to the default GPT-4o for code generation in this ablation study and use the same 32 samples generated by GPT-4o for all of the stage 3 strategies studied. Note that in these tournaments, feedback type is set to be nRMSE similar to Section 4. Results for different feedback types are presented later in this section. Since numerous LLM configurations exist, we select a minimal representative subset from each category. Current models have sufficient input capacity for tournament solver lists; future work could incorporate summarizer agents to compress information for smaller models.

We test six tournament structure categories:

1. Mixed Judges (Default): Combines reasoning and non-reasoning models to balance code generation efficiency with detailed numerical reasoning:

- o3 + GPT-4o + DeepSeek-R1 (Section 4 default)
- o3 + GPT-4o + GPT-4o
- DeepSeek-R1 + GPT-4o + GPT-4o

2. All Reasoning Judges: Uses only reasoning-capable models:

- o3 + o3 + o3
- DeepSeek-R1 + DeepSeek-R1 + DeepSeek-R1
- o3 + o3 + DeepSeek-R1

3. All Non-Reasoning Judges: Uses only standard language models:

- GPT-4o + GPT-4o + GPT-4o

4. Best-of-32 Baseline: Executes all 32 solvers from Analysis and Genesis stages without tournaments.

5. Fixed Criteria Judging: Applies categories 1-3 with predetermined evaluation criteria:

- Numerical stability and convergence properties
- Computational efficiency and scalability
- Mathematical correctness and precision
- Implementation robustness and error handling
- Solution accuracy on benchmark problems

6. Self-Generated Criteria: Applies categories 1-3 where judges first generate their own evaluation criteria before selection.

All strategies use identical 32 solver samples from GPT-4o code generation to ensure fair comparison.

Table 19: nRMSE values for each PDE-SHARP using different LLM combinations for the Synthesis stage.

| | | Advection | Burgers | Reaction-Diffusion | Navier-Stokes | Darcy |
|---|---|---|---|---|---|---|
| **Mixed Judges (Default)** | o3 + GPT-4o + DeepSeek-R1 | 6.11e-04 | 2.31e-04 | 2.29e-03 | 1.51e-02 | 3.97e-03 |
| | o3 + GPT-4o + GPT-4o | 7.34e-04 | 4.45e-04 | 5.41e-03 | 3.58e-02 | 4.12e-03 |
| | DeepSeek-R1 + GPT-4o + GPT-4o | 6.98e-04 | 2.31e-04 | 4.33e-03 | 1.51e-02 | 4.91e-03 |
| **All Reasoning** | o3 + o3 + o3 | 9.74e-04 | 5.19e-04 | 4.21e-03 | 3.45e-02 | 3.84e-03 |
| | DeepSeek-R1 + DeepSeek-R1 + DeepSeek-R1 | 8.92e-04 | 3.23e-04 | 3.25e-03 | 2.47e-02 | 3.84e-03 |
| | o3 + o3 + DeepSeek-R1 | 7.79e-04 | 2.35e-04 | 4.33e-03 | 1.51e-02 | 3.97e-03 |
| **All Non-Reasoning** | GPT-4o + GPT-4o + GPT-4o | 9.74e-04 | 2.57e-04 | 1.01e-02 | 2.62e-02 | 4.90e-03 |
| **Best-of-32 Baseline** | No Tournaments | 1.35e-03 | 3.19e-04 | 3.99e-03 | 3.18e-02 | 7.82e-02 |
| **Fixed Criteria - Mixed Judges** | o3 + GPT-4o + DeepSeek-R1 | 9.86e-04 | 5.25e-04 | 7.24e-03 | 1.48e-02 | 3.89e-03 |
| | o3 + GPT-4o + GPT-4o | 9.18e-04 | 2.38e-04 | 2.36e-02 | 1.54e-02 | 4.05e-03 |
| | DeepSeek-R1 + GPT-4o + GPT-4o | 1.01e-03 | 2.21e-04 | 8.27e-03 | 1.46e-02 | 3.85e-03 |
| **Fixed Criteria - All Reasoning** | o3 + o3 + o3 | 1.73e-03 | 6.11e-04 | 1.15e-02 | 1.41e-02 | 7.76e-03 |
| | DeepSeek-R1 + DeepSeek-R1 + DeepSeek-R1 | 9.74e-04 | 3.17e-04 | 3.19e-03 | 1.44e-02 | 3.82e-03 |
| | o3 + o3 + DeepSeek-R1 | 1.68e-03 | 2.08e-04 | 1.12e-02 | 2.89e-02 | 3.73e-03 |
| **Fixed Criteria - All Non-Reasoning** | GPT-4o + GPT-4o + GPT-4o | 1.01e-03 | 3.43e-04 | 2.42e-03 | 9.29e-02 | 5.01e-03 |
| **Self-Generated Criteria - Mixed Judges** | o3 + GPT-4o + DeepSeek-R1 | 8.12e-04 | 4.67e-04 | 9.15e-03 | 1.62e-02 | 4.21e-03 |
| | o3 + GPT-4o + GPT-4o | 8.53e-04 | 3.02e-04 | 1.89e-02 | 1.38e-02 | 4.57e-03 |
| | DeepSeek-R1 + GPT-4o + GPT-4o | 1.15e-03 | 2.94e-04 | 6.83e-03 | 1.71e-02 | 3.42e-03 |
| **Self-Generated Criteria - All Reasoning** | o3 + o3 + o3 | 1.58e-03 | 7.24e-04 | 1.38e-02 | 1.27e-02 | 8.35e-03 |
| | DeepSeek-R1 + DeepSeek-R1 + DeepSeek-R1 | 9.13e-04 | 2.85e-04 | 4.06e-03 | 1.59e-02 | 4.18e-03 |
| | o3 + o3 + DeepSeek-R1 | 1.52e-03 | 2.76e-04 | 9.84e-03 | 2.53e-02 | 4.29e-03 |
| **Self-Generated Criteria - All Non-Reasoning** | GPT-4o + GPT-4o + GPT-4o | 9.27e-04 | 3.89e-04 | 3.17e-03 | 8.46e-02 | 5.68e-03 |

Table 20: Number of rounds to achieve the results of Table 19 for each PDE-SHARP using different LLM combinations for the Synthesis stage. The number of rounds is reported before performance saturation/degradation, indicating the minimum number of hybridization rounds. The "+" sign indicates a rejudging cycle as explained in Table 21. Note that no hybrid tournaments accur in the best-of-32 strategy.

| | | Advection | Burgers | Reaction-Diffusion | Navier-Stokes | Darcy |
|---|---|---|---|---|---|---|
| **Mixed Judges (Default)** | o3 + GPT-4o + DeepSeek-R1 | 4+4 | 3 | 4 | 3 | 4 |
| | o3 + GPT-4o + GPT-4o | 4+2 | 4 | 4+1 | 4 | 4+1 |
| | DeepSeek-R1 + GPT-4o + GPT-4o | 4+3 | 3 | 4 | 4+1 | 4 |
| **All Reasoning** | o3 + o3 + o3 | 4+4 | 3 | 3 | 3 | 3 |
| | DeepSeek-R1 + DeepSeek-R1 + DeepSeek-R1 | 4+3 | 3 | 4 | 4 | 4 |
| | o3 + o3 + DeepSeek-R1 | 4+3 | 3 | 3 | 3 | 3 |
| **All Non-Reasoning** | GPT-4o + GPT-4o + GPT-4o | 4+4 | 4+2 | 4+2 | 4+4+2 | 4+3 |
| **Best-of-32 Baseline** | No Tournaments | - | - | - | - | - |
| **Fixed Criteria - Mixed Judges** | o3 + GPT-4o + DeepSeek-R1 | 4+3 | 3 | 3 | 3 | 3 |
| | o3 + GPT-4o + GPT-4o | 4+2 | 4 | 4 | 4 | 4 |
| | DeepSeek-R1 + GPT-4o + GPT-4o | 4+3 | 3 | 3 | 4 | 4 |
| **Fixed Criteria - All Reasoning** | o3 + o3 + o3 | 4+3 | 3 | 3 | 3 | 3 |
| | DeepSeek-R1 + DeepSeek-R1 + DeepSeek-R1 | 4+2 | 3 | 3 | 3 | 4 |
| | o3 + o3 + DeepSeek-R1 | 3 | 3 | 3 | 3 | 3 |
| **Fixed Criteria - All Non-Reasoning** | GPT-4o + GPT-4o + GPT-4o | 4+2 | 4+1 | 4+1 | 4+3 | 4+2 |
| **Self-Generated Criteria - Mixed Judges** | o3 + GPT-4o + DeepSeek-R1 | 4+3 | 4 | 4 | 4 | 4+1 |
| | o3 + GPT-4o + GPT-4o | 4+3 | 4+1 | 4+2 | 4+2 | 4+2 |
| | DeepSeek-R1 + GPT-4o + GPT-4o | 4+3 | 4 | 4 | 4+1 | 4 |
| **Self-Generated Criteria - All Reasoning** | o3 + o3 + o3 | 4+4 | 3 | 4 | 3 | 4 |
| | DeepSeek-R1 + DeepSeek-R1 + DeepSeek-R1 | 4+2 | 4 | 4 | 4 | 4 |
| | o3 + o3 + DeepSeek-R1 | 4+1 | 3 | 4 | 3 | 4 |
| **Self-Generated Criteria - All Non-Reasoning** | GPT-4o + GPT-4o + GPT-4o | 4+4 | 4+3 | 4+3 | 4+4+3 | 4+4 |

### B.3.6 Hybridization Feedback Type

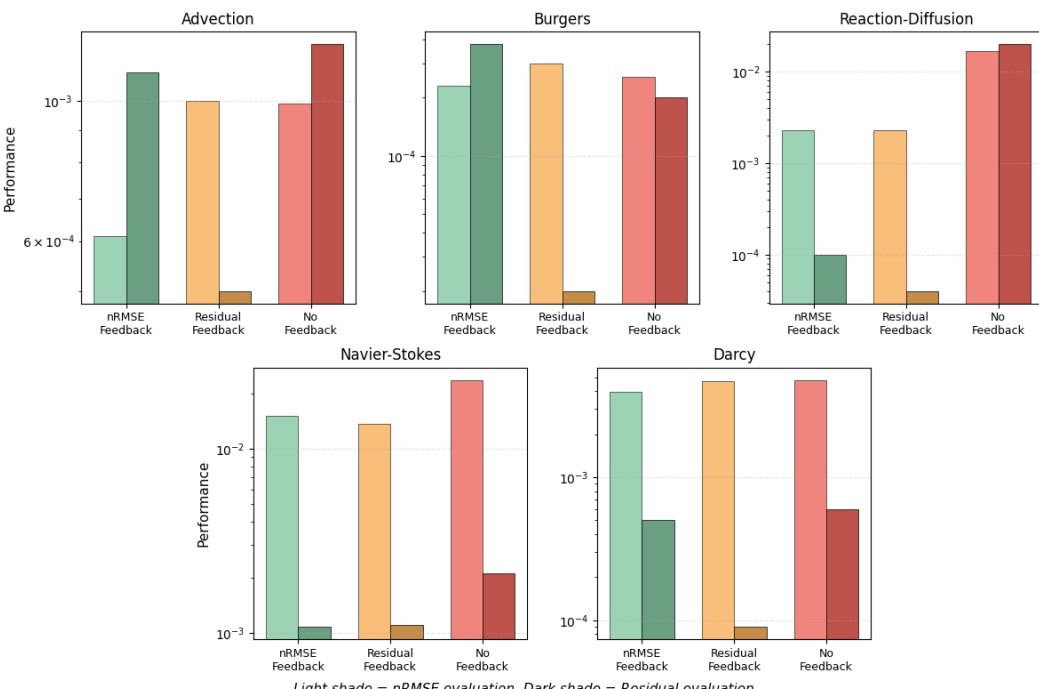

Figure 12: Impact of feedback type on PDE-SHARP solver accuracy across five PDE tasks. Performance is measured using both nRMSE (light bars) and residual evaluation (dark bars) metrics. nRMSE feedback consistently achieves superior performance when evaluated on the nRMSE metric, demonstrating the importance of alignment between feedback type and evaluation criteria. Residual feedback provides a physics-informed alternative when reference solutions are unavailable, while no feedback relies purely on judge code analysis. The choice of feedback type allows adaptation to different research scenarios from benchmark validation to real-world cases with limited reference data.

**Remark: LLM-suggested Feedback Types.** In this part of the section on feedback types, we provide examples of LLM-suggested feedback for each of the tested PDE tasks. The results are generated using GPT-4o as follows.

**(1) Advection:** $\partial_t u + \beta\,\partial_x u = 0$ (periodic; $\beta$ constant)

**General feedback types:**

- **nRMSE**
- **PDE residual** $L^2$: $\|r\|_2$ with $r := \partial_t u + \beta\,\partial_x u$, discretized consistently with the scheme.
- **BC/IC mismatch**: $\|u(t_0,\cdot) - u_0(\cdot)\|_2$, and periodic-wrap mismatch at boundaries.
- **Empirical convergence order** $p$ via two grids $(h, h/2)$:

$$p \approx \frac{\log\big(E(h)/E(h/2)\big)}{\log 2}.$$

- **CFL ratio monitor**:

$$\mathrm{CFL}_{\max} = \max_x \frac{|\beta|\,\Delta t}{\Delta x}$$

(used as a stability penalty when $>$ target).

**PDE-specific feedback types:**

- **Phase-error (Fourier) metric** — detects dispersive drift from exact shift:
  For any wavenumber $k$, let $\hat{u}_k(t)$ be the DFT of $u(\cdot, t)$. The analytic evolution is

  $$\hat{u}_k(t) = \hat{u}_k(0)\, e^{-ik\beta t}.$$

  Define

  $$\epsilon_{\text{phase}}(t) = \left( \sum_{k \in \mathcal{K}} w_k \Big| \arg \hat{u}_k(t) - \arg\big( \hat{u}_k(0)\, e^{-ik\beta t} \big) \Big|^2 \right)^{1/2}.$$

  (Choose $\mathcal{K}$ = dominant modes; $w_k$ normalize by spectral energy.)
  *Why:* linear advection is phase-exact; any phase drift degrades solution even when $L^2$ error is small.

- **Amplitude-damping metric** — detects artificial diffusion:

  $$\epsilon_{\text{amp}}(t) = \left( \sum_{k \in \mathcal{K}} w_k \big| \, |\hat{u}_k(t)| - |\hat{u}_k(0)| \, \big|^2 \right)^{1/2}.$$

  *Why:* upwinding or overly diffusive fluxes damp modes; useful when the reference data were generated by a specific finite-volume scheme and you want to "match" it. (This is exactly what happened in your advection case study where nRMSE feedback nudged judges toward a MUSCL/TVD FV scheme instead of an analytical shifter.)

- **Invariant-conservation drift** — detects systematic bias:
  Mass and $L^2$ are constant for periodic, constant-$\beta$ advection:

  $$\delta_{\text{mass}}(t) = \frac{\left| \int_0^1 u(x,t)\, dx - \int_0^1 u_0(x)\, dx \right|}{\left| \int_0^1 u_0(x)\, dx \right|}, \qquad \delta_{L^2}(t) = \frac{\|u(\cdot,t)\|_2 - \|u_0\|_2}{\|u_0\|_2}.$$

  *Why:* catches subtle dissipation or numerical pumping even when nRMSE is small.

**(2) Burgers:** $\quad \partial_t u + \partial_x(u^2/2) = \nu\, \partial_{xx} u$ (periodic; $\nu = 0.01$)

**General feedback types:**

- **nRMSE, PDE residual** $L^2$ with $r := \partial_t u + \partial_x(u^2/2) - \nu\partial_{xx}u$.
- **Convergence order** $p$ (as above).
- **Max CFL monitor** with characteristic speed $\lambda_{\max} = |u|_\infty \cdot \frac{\Delta t}{\Delta x}$.
- **Boundary/periodicity mismatch**.

**PDE-specific feedback types:**

- **Entropy inequality violation (integrated)** — penalizes non-admissible shocks/oscillations:
  With entropy $\eta(u) = \frac{1}{2}u^2$, viscous Burgers satisfies:

  $$\frac{d}{dt} \int_0^1 \tfrac{1}{2}u^2\, dx = -\nu \int_0^1 (\partial_x u)^2\, dx \ \leq\ 0.$$

  Define

  $$\Phi_{\text{entropy}} = \sum_n \max\left( 0, \int_0^1 \tfrac{1}{2}u^2(x, t_{n+1})\, dx - \int_0^1 \tfrac{1}{2}u^2(x, t_n)\, dx \right).$$

  *Why:* any net increase flags spurious energy injection near steep gradients.

- **Total variation (TV) growth** — damps Gibbs and enforces TVD behavior:

$$\mathrm{TV}(u) = \sum_j |u_{j+1} - u_j|, \qquad \Phi_{\mathrm{TV}} = \sum_n \max\left(0,\ \mathrm{TV}(u^{n+1}) - \mathrm{TV}(u^n)\right).$$

  *Why:* shocks should not create oscillations; TV growth is a crisp signal.

- **Mean (mass) conservation drift** — periodic Burgers conserves $\int u\,dx$:

$$\delta_{\mathrm{mean}}(t) = \frac{\left| \int_0^1 u(x,t)\,dx - \int_0^1 u_0(x)\,dx \right|}{\left| \int_0^1 u_0(x)\,dx \right|}.$$

  *Why:* catches subtle bias from asymmetric limiters or boundary handling.

**(3) Reaction–Diffusion (Fisher–KPP form):** $\quad \partial_t u - \nu\partial_{xx}u - \rho\,u(1-u) = 0 \quad$ (periodic; $\nu = 0.5$, $\rho = 1$)

**General feedback types:**

- **nRMSE, PDE residual** $L^2$ with $r := \partial_t u - \nu\partial_{xx}u - \rho u(1-u)$.
- **Convergence order** $p$.
- **Diffusive CFL monitor** (for explicit pieces): $\max \dfrac{\nu\Delta t}{\Delta x^2}$.

**PDE-specific feedback types:**

- **Maximum-principle / positivity violation** — enforces physically meaningful range:
  For logistic reaction, the continuous solution stays in $[0,1]$ when $u_0 \in [0,1]$. Define

$$\Phi_{\mathrm{MP}} = \left( \int_0^1 (\max(0, -u))^2\,dx \right)^{1/2} + \left( \int_0^1 (\max(0, u-1))^2\,dx \right)^{1/2}.$$

  *Why:* catches overshoot/undershoot from aggressive time steps or limiters.

- **Split-step (hybrid) consistency error** — encourages the analytically-integrated reaction that your analysis stage favors:
  *If Strang/IMEX or analytical-reaction is used, compare the reaction sub-update to the exact ODE update:*

$$R_{\Delta t}(u) = \frac{u\,e^{\rho\Delta t}}{1 + u\left(e^{\rho\Delta t} - 1\right)}.$$

  Define $\varepsilon_{\mathrm{react}} = \|u^{n+\frac{1}{2}} - R_{\Delta t}(u^n)\|_2$ (or analogous placement per scheme).
  *Why:* rewards the hybrid analytical–numerical strategy your framework discovers for this PDE.

- **Stiffness-aware step safety** — keeps reaction eigenvalue under control for explicit parts:
  Spectral radius for reaction $J = \rho(1 - 2u) \Rightarrow |\rho(J)| \le \rho$. Penalize $\max_n \max_x \dfrac{\Delta t\,\rho}{\rho_{\mathrm{exact}}} > 1$.
  *Why:* prevents overshoot/explosions when reaction is treated explicitly.

**(4) Compressible Navier–Stokes ($\Gamma = 5/3$):**

$$\partial_t\rho + \partial_x(\rho v) = 0,$$
$$\rho\left(\partial_t v + v\partial_x v\right) = -\partial_x p + \eta\,\partial_x^2 v + \left(\zeta + \frac{\eta}{3}\right)\partial_x(\partial_x v),$$

$$\partial_t \left( \epsilon + \frac{\rho v^2}{2} \right) + \partial_x \left[ \left( \epsilon + p + \frac{\rho v^2}{2} \right) v - v \, \sigma' \right] = 0, \quad \epsilon = \frac{p}{\Gamma - 1}, \quad \sigma' = \left( \zeta + \frac{4}{3} \eta \right) \partial_x v.$$

**General feedback types:**

- **nRMSE** on chosen state(s) ($\rho$, $v$, $p$, or conservative variables).

- **Vector PDE residual** (mass, momentum, energy) in normalized $L^2$ (sum of per-equation residual norms).

- **Convergence order** $p$.

- **Maximum acoustic CFL:**
$$\max \frac{(|v| + c)\Delta t}{\Delta x}, \qquad c = \sqrt{\Gamma p/\rho}.$$

- **BC/periodicity mismatch**.

**PDE-specific feedback types:**

- **Conservation-law drift** — ensures discrete conservation:
$$\delta_{\mathrm{mass}}(t) = \frac{\left| \int \rho(x,t) \, dx - \int \rho(x,0) \, dx \right|}{\int \rho(x,0) \, dx}, \qquad \delta_{\mathrm{mom}}(t) = \frac{\left| \int \rho v \, dx - \int \rho_0 v_0 \, dx \right|}{\int |\rho_0 v_0| \, dx},$$
$$\delta_{\mathrm{energy}}(t) = \frac{\left| \int \left( \epsilon + \frac{\rho v^2}{2} \right) dx - \int \left( \epsilon_0 + \frac{\rho_0 v_0^2}{2} \right) dx \right|}{\int \left( \epsilon_0 + \frac{\rho_0 v_0^2}{2} \right) dx}.$$

  *Why:* small global drifts reveal flux/boundary inconsistencies even if pointwise errors look OK.

- **Positivity violations** — hard physical constraints:
$$\Phi_{\rho,p} = \| \min(0, \rho) \|_1 + \| \min(0, p) \|_1.$$

  *Why:* avoids catastrophic instabilities (negative density/pressure).

- **Entropy production sign check** — flags nonphysical dissipation/oscillations:
  For ideal gas, specific entropy $s = \ln(p) - \Gamma \ln(\rho)$. Define
$$\sigma(t) = \int \rho s \, dx, \qquad \Phi_{\mathrm{entropy}} = \sum_n \max(0, -(\sigma^{n+1} - \sigma^n)).$$

  *Why:* with viscosity, total entropy should not decrease; negative production indicates spurious behavior.

- **Rankine–Hugoniot defect (interface balance)** — shock-consistency check in conservative form:
  For each interface $i + \frac{1}{2}$ and conserved vector $U = (\rho, \rho v, E)$, flux $\mathbf{F}$, penalize the discrete jump
$$\Phi_{\mathrm{RH}} = \sum_{n,i} \left\| \frac{U_i^{n+1} - U_i^n}{\Delta t} + \frac{F_{i+\frac{1}{2}}^n - F_{i-\frac{1}{2}}^n}{\Delta x} \right\|_1.$$

  *Why:* targets the exact property your solver should satisfy at shocks/contacts.

**(5) Darcy flow (steady, Dirichlet):** $\quad -\nabla \cdot (a(x)\nabla u) = \beta, \quad u|_{\partial\Omega} = 0$

**General feedback types:**

- **PDE residual norms at steady state:**
$$\|r\|_2 = \|\beta + \nabla \cdot (a\nabla u_h)\|_{L^2(\Omega)}.$$

- **Boundary condition residual:** $\|u_h\|_{L^2(\partial\Omega)}$ (often $\approx 0$ if enforced strongly; still useful with FV).
- **Grid-refinement check** using energy-norm proxy below.

**PDE-specific feedback types:**

- **Residual-jump a-posteriori estimator (energy-norm surrogate)** — standard for elliptics; localizes errors cheaply:
  For each cell $K$ with diameter $h_K$,

$$r_K = \beta + \nabla \cdot (a\nabla u_h)\big|_K, \qquad J_e = a\nabla u_h \cdot n_e \text{ on edge } e,$$

$$\eta^2 = \sum_K \left( h_K^2 \|r_K\|_{L^2(K)}^2 + \sum_{e\subset\partial K} h_e \|J_e\|_{L^2(e)}^2 \right).$$

  *Why:* mirrors FE error estimators; correlates with the true $a$-energy error without ground truth.
- **Local mass balance (cell-wise)** — ensures flux consistency:

$$\Phi_{\text{mass}} = \sum_K \left| \int_K \beta \, dx + \int_{\partial K} (a\nabla u_h) \cdot n \, ds \right|.$$

  *Why:* FV/FD/FE schemes should balance source with flux divergence on each control volume.
- **Global compatibility check** — sanity for data/boundary handling:

$$\left| \int_\Omega \beta \, dx + \int_{\partial\Omega} (a\nabla u_h) \cdot n \, ds \right|.$$

  *Why:* catches solver or BC mishandling even when $\|r\|_2$ looks small.

### B.3.7 NUMBER OF ROUNDS & CYCLES

To determine the optimal number of hybridization rounds and rejudging cycles, we conduct an analysis tracking solver accuracy improvements across eight total rounds (four initial hybridization rounds plus four rejudging cycle rounds) for all tested PDEs. Figure 13 demonstrates the round-by-round progression of best achieved nRMSE in that round (among the tested three), with a vertical dashed line separating the initial hybridization cycle from the rejudging cycle.

The results reveal different patterns across different PDE types. Most PDEs achieve optimal performance within 3-4 initial hybridization rounds, after which additional rounds provide saturation or even slight performance degradation. Advection presents a notable exception, continuing to benefit from one rejudging cycle. This stems from a dataset-specific subtlety: while analytical solutions exist for the mathematical advection equation, the PDEBench reference solutions were generated using finite-volume methods. The rejudging cycle enables PDE-SHARP to adapt from initially favoring analytical approaches to numerical methods that better match the dataset's characteristics. This mostly occurs when the feedback type is set to be nRMSE in the tournaments. See Figure 18 for results using other feedback types (residual feedback, no feedback) for the advection PDE.

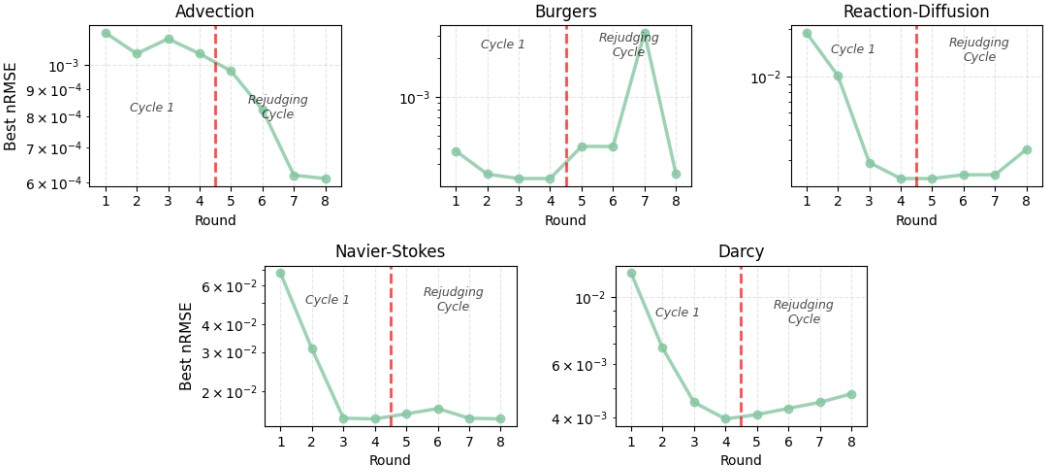

Vertical dashed line separates initial hybridization rounds from rejudging cycle

Figure 13: Progression of the best nRMSE of each hybridization round for each PDE task

Table 21: Average number of **Hybridization Rounds**, **Rejudging Cycles**, and total evaluations

| PDE | # Hybrid. Rounds | # Rejudging Cycles | # Total Evals |
|-----|-----------------|-------------------|---------------|
| Advection | $4 + 4$ | 1 | 24 |
| Burgers | 3 | 0 | 9 |
| Reaction-Diffusion | 4 | 0 | 12 |
| Navier-Stokes | 3 | 0 | 9 |
| Darcy | 4 | 0 | 12 |

For four out of five tested PDEs, PDE-SHARP achieves optimal results using fewer than 13 solver evaluations on average (Table 21), with most improvement occurring in the initial 3-4 rounds, resulting in a computational advantage over baseline methods requiring 30+ evaluations, while the rejudging cycle provides additional benefits only for specific cases.

### B.4 ANALYSIS OF THE GENERATED SOLVER CODE QUALITY

Beyond solution accuracy, we analyze the computational and numerical properties of generated solver code across all methods. This analysis examines three key quality indicators: execution time efficiency, library usage, and empirical convergence rates. These metrics reveal whether frameworks

generate production-ready code with proper numerical characteristics, not merely code that produces correct outputs through inefficient or unstable implementations.

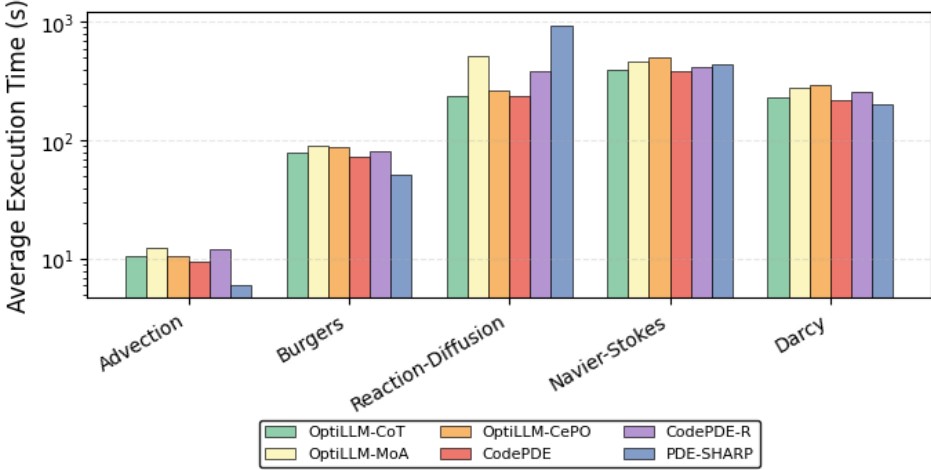

Figure 14: Average execution times across PDE tasks. PDE-SHARP achieves lower execution times than the average baseline in 4/5 cases. For reaction-diffusion, higher execution time reflects the rigorous numerical methods selected by stability analysis as expected, which produce significantly higher accuracy solvers (Table 2).

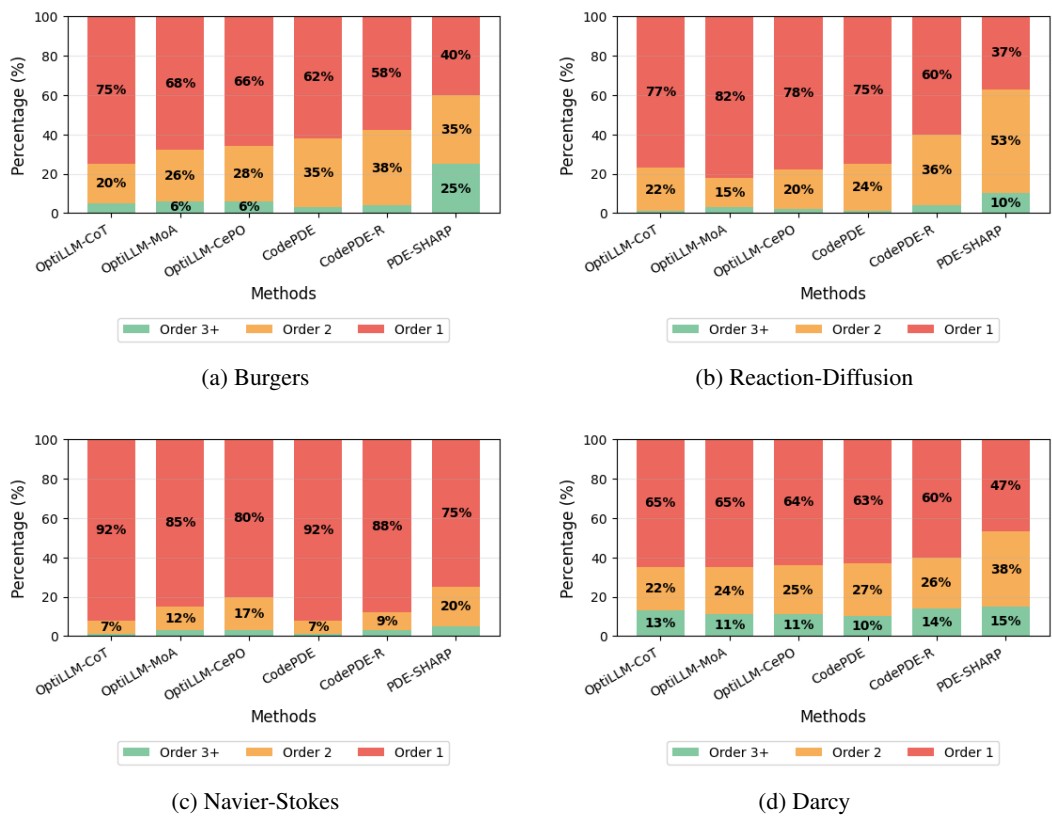

Figure 15: Convergence order distribution across different PDEs. The convergence order distribution for the advection PDE appears in Figure 4b.

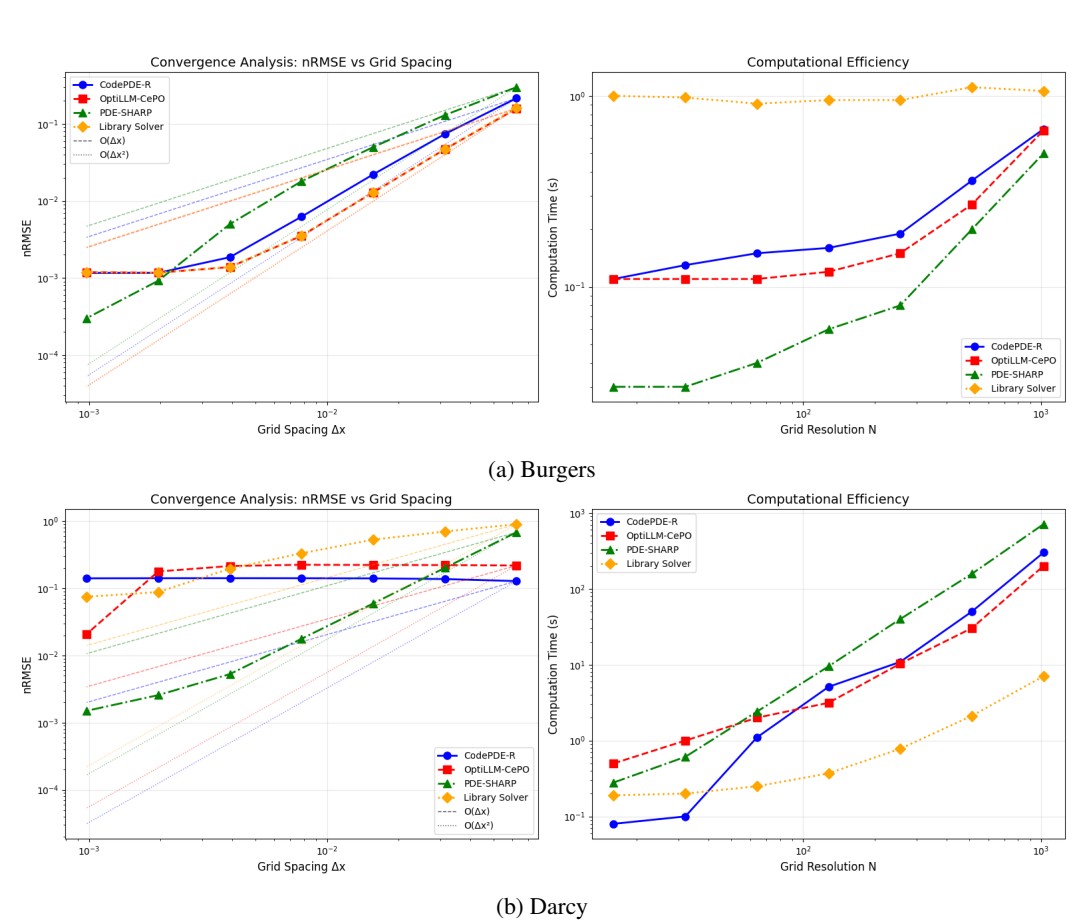

(a) Burgers

(b) Darcy

Figure 16

| PDE | Method | SciPy | JAX | NumPy | PyTorch |
|---|---|---|---|---|---|
| Advection | PDE-SHARP | 10% | 17% | 48% | 25% |
| Burgers | PDE-SHARP | 10% | 32% | 25% | 33% |
| Reaction-Diffusion | PDE-SHARP | 8% | 1% | 49% | 25% |
| Comp. Navier-Stokes | PDE-SHARP | 7% | 37% | 30% | 26% |
| Darcy | PDE-SHARP | 43% | 15% | 15% | 27% |

Table 22: PDE-SHARP decreases Python usage and increased JAX + SciPy usage overall across all tested PDEs

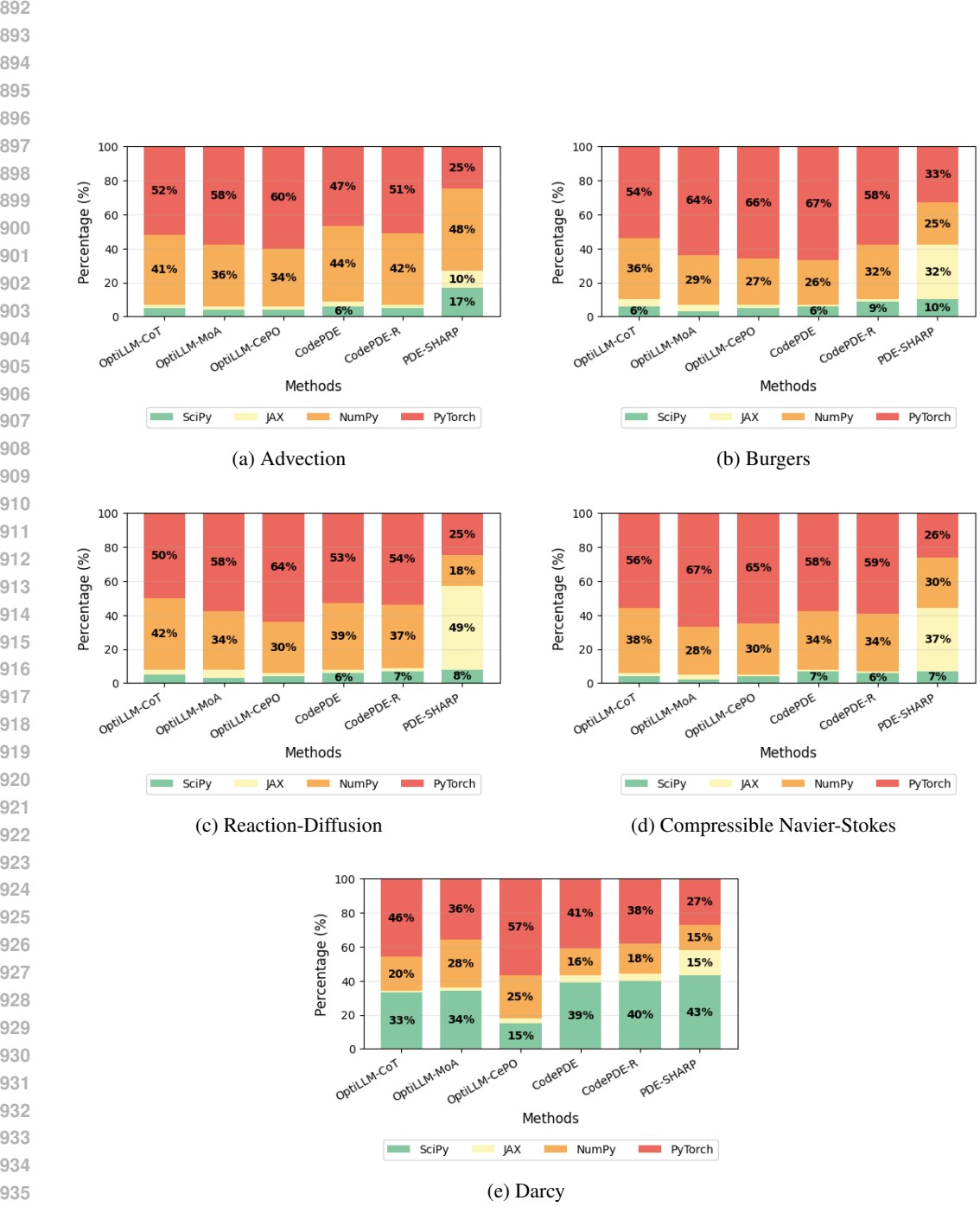

(a) Advection

(b) Burgers

(c) Reaction-Diffusion

(d) Compressible Navier-Stokes

(e) Darcy

Figure 17: Solver library usage across different PDEs.

## C   ADDITIONAL DETAILS ON THE TESTED PDES

In this section of the appendix, we present the differential equations we study in our experiments.

### C.1   ADVECTION

The 1D advection equation is a hyperbolic PDE which models processes such as fluid flow, heat transfer, and biological dynamics. It is given by

$$\begin{cases} \partial_t u(t,x) + \beta \partial_x u(t,x) = 0, & x \in (0,1), \ t \in (0,2] \\ u(0,x) = u_0(x), & x \in (0,1) \end{cases}$$

where $\beta$ is a constant representing the advection speed. In our experiments, we assume the periodic boundary condition and report results for the $\beta = 0.1$ case using the advection dataset from PDEBench.

### C.2   BURGERS

The Burgers equation, a fundamental PDE in fluid mechanics, is used to model various nonlinear phenomena including shock waves and traffic flow. We examine the following form of the Burgers' equation: The one-dimensional Burgers' Equation is given by

$$\begin{cases} \partial_t u(x,t) + \partial_x \left( \frac{u^2(x,t)}{2} \right) = \nu \partial_{xx} u(x,t), & x \in (0,1), \ t \in (0,1] \\ u(x,0) = u_0(x), & x \in (0,1) \end{cases}$$

where $\nu$ is a constant representing the viscosity. In our experiments, we assume the periodic boundary condition and report results for the $\nu = 0.01$ case using the Burgers dataset from PDEBench.

### C.3   REACTION-DIFFUSION

The 1D reaction-diffusion PDE is given by

$$\begin{cases} \partial_t u(t,x) - \nu \partial_{xx} u(t,x) - \rho u(1-u) = 0, & x \in (0,1), \ t \in (0,T] \\ u(0,x) = u_0(x), & x \in (0,1) \end{cases}$$

where $\nu$ and $\rho$ are coefficients representing diffusion and reaction terms, respectively. In our experiments, we assume the periodic boundary condition and report results for the $\nu = 0.5$ and $\rho = 1.0$ case using the reaction-diffusion dataset from PDEBench.

### C.4   NAVIER-STOKES

The compressible Navier-Stokes equations are given by

$$\begin{cases} \partial_t \rho + \partial_x(\rho v) = 0 \\ \rho(\partial_t v + v \partial_x v) = -\partial_x p + \eta \partial_{xx} v + (\zeta + \eta/3)\partial_x(\partial_x v) \\ \partial_t \left[ \epsilon + \frac{\rho v^2}{2} \right] + \partial_x \left[ \left( \epsilon + p + \frac{\rho v^2}{2} \right) v - v\sigma' \right] = 0 \end{cases}$$

where $\rho$ is the mass density, $v$ is the velocity, $p$ is the gas pressure, $\epsilon = p/(\Gamma - 1)$ is the internal energy with $\Gamma = 5/3$, $\sigma' = (\zeta + \frac{4}{3}\eta)\partial_x v$ is the viscous stress tensor, and $\eta, \zeta$ are the shear and bulk viscosity coefficients, respectively. In our task, we assume periodic boundary conditions. The spatial domain is $\Omega = [-1,1]$. For this study, we used the compressible Navier-Stokes dataset from PDEBench with $\eta = \zeta = 0.1$

## C.5 DARCY FLOW

We study the 2D Darcy flow equation given by:

$$-\nabla \cdot (a(x)\nabla u(x)) = \beta, \quad x \in (0,1)^2$$

with the boundary condition:

$$u(x) = 0, \quad x \in \partial(0,1)^2$$

where $u(x)$ is the solution function, the force term is set as a constant value $\beta$, and $a(x)$ is a batch of coefficient function. In our experiments, we report results for the $\beta = 1.0$ case using the Darcy flow dataset from PDEBench.

## D CASE STUDIES

### D.1 ADVECTION

#### D.1.1 USE OF PDE SOLVER LIBRARIES

In this case study, we compare the numerical strategies of two advection solvers: **Solver A:** The best custom solver generated by PDE-SHARP for the advection PDE. **Solver B:** A library-based solver generated by Claude Code using scientific computing libraries (as described in Appendix B.2).

We present the full code for both solvers below, followed by a comparative analysis of their design choices and computational approaches. For quantitative performance metrics — including nRMSE, convergence rates, and execution times — we refer the reader to 2 (main results), 17 (library-based results), B.4 (solver code quality analysis).

**PDE-SHARP-generated custom solver (Solver A):**

```python
import numpy as np

def _vl_limiter(a, b, c, alpha=2.0):
    return (
        np.sign(c)
        * (0.5 + 0.5 * np.sign(a * b))
        * np.minimum(alpha * np.minimum(np.abs(a), np.abs(b)), np.abs(c))
    )

def _muscl_reconstruct(u_ghost):
    batch, n_plus4 = u_ghost.shape
    N = n_plus4 - 4
    du_L = u_ghost[:, 1 : N + 3] - u_ghost[:, 0 : N + 2]
    du_R = u_ghost[:, 2 : N + 4] - u_ghost[:, 1 : N + 3]
    du_M = 0.5 * (u_ghost[:, 2 : N + 4] - u_ghost[:, 0 : N + 2])
    gradu = _vl_limiter(du_L, du_R, du_M)
    uL = np.zeros_like(u_ghost)
    uR = np.zeros_like(u_ghost)
    uL[:, 1 : N + 3] = u_ghost[:, 1 : N + 3] - 0.5 * gradu
    uR[:, 1 : N + 3] = u_ghost[:, 1 : N + 3] + 0.5 * gradu
    return uL, uR

def _build_ghost(u):
    return np.concatenate([u[:, -2:], u, u[:, :2]], axis=1)

def _rusanov_flux(u, beta):
    batch, N = u.shape
    u_ghost = _build_ghost(u)
    uL, uR = _muscl_reconstruct(u_ghost)
    fL = beta * uL
    fR = beta * uR
    absb = abs(beta)
    flux = 0.5 * (
        fR[:, 1 : N + 2]
        + fL[:, 2 : N + 3]
        - absb * (uL[:, 2 : N + 3] - uR[:, 1 : N + 2])
    )
    return flux

def _ssp_rk2_step(u, dt, dx, beta):
    f  = _rusanov_flux(u, beta)
    u1 = u - 0.5 * dt / dx * (f[:, 1:] - f[:, :-1])
    f1  = _rusanov_flux(u1, beta)
    unew = u - dt / dx * (f1[:, 1:] - f1[:, :-1])
    return unew

def solver(u0_batch, t_coordinate, beta, cfl=0.5):
    u0_batch  = np.asarray(u0_batch, dtype=np.float32)
```

```
2106        t_coordinate = np.asarray(t_coordinate, dtype=np.float32)
2107        if t_coordinate[0] != 0.0:
2108            raise ValueError("t_coordinate must start with 0.")
2109        batch, N = u0_batch.shape
2110        dx = 1.0 / N
2111        dt_cfl = cfl * dx / (abs(beta) + 1e-8)
2112        T_plus_1 = len(t_coordinate)
2113        sol = np.empty((batch, T_plus_1, N), dtype=np.float32)
2114        sol[:, 0, :] = u0_batch
2115        u = u0_batch.copy()
2116        t_now = 0.0
2117        for idx in range(1, T_plus_1):
2118            t_target = float(t_coordinate[idx])
2119            while t_now + dt_cfl < t_target - 1e-12:
2120                u = _ssp_rk2_step(u, dt_cfl, dx, beta)
2121                t_now += dt_cfl
2122            dt_last = t_target - t_now
2123            if dt_last > 1e-12:
2124                u = _ssp_rk2_step(u, dt_last, dx, beta)
2125                t_now = t_target
2126            sol[:, idx, :] = u
2127            if idx <= 3:
                    print(f"[solver] reached t={t_now:.3f}  (snapshot {idx}/{
                        T_plus_1-1})")
        return sol
```

**Claude Code-generated library solver (Solver B):**

```
2130   import numpy as np
2131   import jax
2132   import jax.numpy as jnp
2133   from jax_cfd.base import grids
2134   from jax_cfd.base import finite_differences as fd
2135   from jax_cfd.base import boundaries
2136   def solver(u0_batch, t_coordinate, beta):
2137
2138       print(f"Starting solver with batch_size={u0_batch.shape[0]}, N={
                 u0_batch.shape[1]}, T={len(t_coordinate)-1}")
2139       print(f"Beta parameter: {beta}")
2140
2141       u0_batch_jax = jnp.array(u0_batch, dtype=jnp.float32)
2142       t_coordinate_jax = jnp.array(t_coordinate, dtype=jnp.float32)
2143
2144       batch_size, N = u0_batch_jax.shape
2145       T = len(t_coordinate_jax) - 1
2146
2147       dx = 1.0 / N
2148       x = jnp.linspace(0, 1, N, endpoint=False)
2149
2150       print(f"Grid spacing: dx = {dx:.6f}")
2151
2152       k = 2.0 * jnp.pi * jnp.fft.fftfreq(N, d=dx)
2153
2154       dt_cfl = 0.4 * dx / (abs(beta) + 1e-10)
2155       print(f"Maximum stable time step (CFL=0.4): {dt_cfl:.6f}")
2156
2157       solutions = jnp.zeros((batch_size, T + 1, N), dtype=jnp.float32)
2158       solutions = solutions.at[:, 0, :].set(u0_batch_jax)
2159
       @jit
       def spectral_step(u, dt):
           u_hat = jnp.fft.fft(u, axis=-1)
           u_hat_new = u_hat * jnp.exp(-1j * beta * k * dt)
           u_new = jnp.fft.ifft(u_hat_new, axis=-1)
```

```
2160
2161        return jnp.real(u_new)
2162
2163    current_u = u0_batch_jax
2164
2165    for time_idx in range(1, T + 1):
2166        t_start = t_coordinate_jax[time_idx - 1]
2167        t_end = t_coordinate_jax[time_idx]
2168        dt_total = t_end - t_start
2169
2170        print(f"Advancing from t={t_start:.4f} to t={t_end:.4f} (dt={
2171            dt_total:.6f})")
2172
2173        n_substeps = max(1, int(jnp.ceil(dt_total / dt_cfl)))
2174        dt_substep = dt_total / n_substeps
2175
2176        if time_idx == 1:
2177            print(f"Using {n_substeps} substeps per output (substep dt={
2178                dt_substep:.6f})")
2179            print(f"Effective CFL number: {abs(beta) * dt_substep / dx:.4
2180                f}")
2181
2182        temp_u = current_u
2183        for substep in range(n_substeps):
2184            temp_u = spectral_step(temp_u, dt_substep)
2185
2186        current_u = temp_u
2187
2188        solutions = solutions.at[:, time_idx, :].set(current_u)
2189
2190        if time_idx <= 3:
2191            u_min = jnp.min(current_u)
2192            u_max = jnp.max(current_u)
2193            u_mean = jnp.mean(current_u)
                print(f"  Step {time_idx}: u_range=[{u_min:.4f}, {u_max:.4f
                    }], mean={u_mean:.4f}")

    print(f"Solver completed. Final solution shape: {solutions.shape}")

    return np.array(solutions)
```

### Numerical Approach & Algorithmic Design

The two advection solvers demonstrate fundamentally different numerical approaches to solving the linear advection equation $\partial_t u + \beta \, \partial_x u = 0$, reflecting distinct design philosophies: one based on custom finite-volume methods and the other leveraging high-performance spectral libraries.

### Solver A (PDE-SHARP): Custom Finite-Volume Method

- **Spatial Discretization:** Employs a second-order MUSCL (Monotonic Upstream-centered Scheme for Conservation Laws) reconstruction with a van Leer limiter (`_vl_limiter`), providing TVD (Total Variation Diminishing) properties that prevent oscillations near sharp gradients. The limiter uses minmod-type logic to ensure monotonicity.

- **Numerical Flux:** Uses the Rusanov (local Lax-Friedrichs) flux, which adds sufficient numerical dissipation for stability while maintaining conservation.

- **Temporal Integration:** Implements SSP-RK2 (Strong Stability Preserving Runge-Kutta, 2nd order) with adaptive CFL-based time stepping (CFL = 0.5). The scheme ensures stability under the CFL condition $\Delta t \leq 0.5 \Delta x / |\beta|$.

- **Boundary Handling:** Uses ghost cells (two layers) with periodic wrapping implemented via array concatenation.

- **Implementation:** Pure NumPy, manually vectorized for batch processing.

**Solver B (Claude Code): Spectral Method via JAX-CFD**

- **Spatial Discretization:** Uses a pseudo-spectral method that computes spatial derivatives via Fourier transforms. The method assumes periodic boundaries and provides exponential convergence for smooth solutions.

- **Temporal Integration:** Exact time integration in Fourier space—the solution is advanced by multiplying Fourier coefficients by $e^{-i\beta k \Delta t}$. This is semi-Lagrangian in spectral space and has no temporal truncation error for constant $\beta$.

- **Stability Approach:** Uses a CFL-based time step (CFL = 0.4), but the method is unconditionally stable for linear advection. The substep restriction is for accuracy (phase alignment), not stability.

- **Library Dependencies:** Built on JAX-CFD, a JAX-based computational fluid dynamics library that provides automatic differentiation, GPU acceleration, and spectral operators.

- **Implementation:** JAX-based with just-in-time (JIT) compilation (`@jit` decorator) for performance.

**Theoretical Properties & Convergence**

| Property | Solver A (PDE-SHARP) | Solver B (Library) |
|---|---|---|
| Spatial Order | 2nd-order (with limiter) | Spectral (exponential) for smooth solutions |
| Temporal Order | 2nd-order (SSP-RK2) | Exact (no truncation error) for constant $\beta$ |
| Stability | Conditional (CFL $\leq$ 0.5) | Unconditionally stable |
| Conservation | Conservative (finite-volume) | Conservative (spectral with proper dealiasing) |
| Shock Handling | TVD, non-oscillatory | Gibbs phenomena for discontinuous data |
| Best Application | Problems with sharp gradients, shocks | Smooth solutions, high accuracy requirements |

**Library Analysis: JAX-CFD vs. Custom Implementation**

*JAX-CFD (Solver B) provides:*

- **Automatic Differentiation:** Enables gradient computations through the solver for optimization or inverse problems.

- **GPU Acceleration:** JAX compilation to GPU/TPU backends.

- **Spectral Accuracy:** Exponential convergence for smooth problems.

- **Physical Boundary Objects:** Structured handling of periodic/other boundaries via boundaries module.

*Custom Implementation (Solver A) provides:*

- **Algorithmic Transparency:** Full control over numerical schemes and limiters.

- **Robustness for Non-Smooth Problems:** TVD properties handle discontinuities.

- **No External Dependencies:** Pure NumPy implementation.

- **Explicit Conservation:** Finite-volume formulation guarantees conservation.

**Computational Cost:**

- **Solver A:** $\mathcal{O}(N)$ per time step, but requires small $\Delta t$ due to CFL condition.

- **Solver B:** $\mathcal{O}(N \log N)$ per time step (FFT cost), but can use larger $\Delta t$.

**Grid Convergence:** For smooth problems, Solver B achieves exponential convergence, while Solver A shows 2nd-order convergence. However, for the advection problem with potential sharp gradients (common in PDEBench), Solver A's TVD properties are advantageous.

**Methodological Implications for LLM-Driven Solver Generation:** This comparison highlights a key insight from our framework: PDE-SHARP generates context-aware solvers that align with

benchmark characteristics. While library-based approaches (Solver B) can leverage sophisticated numerical infrastructure, they may not optimize for specific benchmark requirements. PDE-SHARP's Analysis stage correctly identifies the need for shock-capturing capabilities (via stability analysis and PDE classification), leading to the finite-volume approach in Solver A, which outperforms the spectral method on the PDEBench advection task.

The choice between these approaches exemplifies the feedback-driven adaptation in PDE-SHARP's Synthesis stage: when nRMSE feedback indicates discrepancies with benchmark solutions, the framework converges on methods that match the benchmark's numerical characteristics rather than purely maximizing theoretical accuracy. **Numerical Approach & Algorithmic Design**

The two advection solvers demonstrate fundamentally different numerical approaches to solving the linear advection equation $\partial_t u + \beta \, \partial_x u = 0$, reflecting distinct design philosophies: one based on custom finite-volume methods and the other leveraging high-performance spectral libraries.

**Solver A (PDE-SHARP): Custom Finite-Volume Method**

- **Spatial Discretization:** Employs a second-order MUSCL (Monotonic Upstream-centered Scheme for Conservation Laws) reconstruction with a van Leer limiter (`_vl_limiter`), providing TVD (Total Variation Diminishing) properties that prevent oscillations near sharp gradients. The limiter uses minmod-type logic to ensure monotonicity.

- **Numerical Flux:** Uses the Rusanov (local Lax-Friedrichs) flux, which adds sufficient numerical dissipation for stability while maintaining conservation.

- **Temporal Integration:** Implements SSP-RK2 (Strong Stability Preserving Runge-Kutta, 2nd order) with adaptive CFL-based time stepping (CFL = 0.5). The scheme ensures stability under the CFL condition $\Delta t \leq 0.5\Delta x/|\beta|$.

- **Boundary Handling:** Uses ghost cells (two layers) with periodic wrapping implemented via array concatenation.

- **Implementation:** Pure NumPy, manually vectorized for batch processing.

**Solver B (Claude Code): Spectral Method via JAX-CFD**

- **Spatial Discretization:** Uses a pseudo-spectral method that computes spatial derivatives via Fourier transforms. The method assumes periodic boundaries and provides exponential convergence for smooth solutions.

- **Temporal Integration:** Exact time integration in Fourier space—the solution is advanced by multiplying Fourier coefficients by $e^{-i\beta k\Delta t}$. This is semi-Lagrangian in spectral space and has no temporal truncation error for constant $\beta$.

- **Stability Approach:** Uses a CFL-based time step (CFL = 0.4), but the method is unconditionally stable for linear advection. The substep restriction is for accuracy (phase alignment), not stability.

- **Library Dependencies:** Built on JAX-CFD, a JAX-based computational fluid dynamics library that provides automatic differentiation, GPU acceleration, and spectral operators.

- **Implementation:** JAX-based with just-in-time (JIT) compilation (`@jit` decorator) for performance.

**Theoretical Properties & Convergence**

| Property | Solver A (PDE-SHARP) | Solver B (Library) |
|---|---|---|
| Spatial Order | 2nd-order (with limiter) | Spectral (exponential) for smooth solutions |
| Temporal Order | 2nd-order (SSP-RK2) | Exact (no truncation error) for constant $\beta$ |
| Stability | Conditional (CFL $\leq$ 0.5) | Unconditionally stable |
| Conservation | Conservative (finite-volume) | Conservative (spectral with proper dealiasing) |
| Shock Handling | TVD, non-oscillatory | Gibbs phenomenon for discontinuous data |
| Best Application | Problems with sharp gradients, shocks | Smooth solutions, high accuracy requirements |

**Library Analysis: JAX-CFD vs. Custom Implementation**

*JAX-CFD (Solver B) provides:*

- **Automatic Differentiation:** Enables gradient computations through the solver for optimization or inverse problems.
- **GPU Acceleration:** JAX compilation to GPU/TPU backends.
- **Spectral Accuracy:** Exponential convergence for smooth problems.
- **Physical Boundary Objects:** Structured handling of periodic/other boundaries via boundaries module.

*Custom Implementation (Solver A) provides:*

- **Algorithmic Transparency:** Full control over numerical schemes and limiters.
- **Robustness for Non-Smooth Problems:** TVD properties handle discontinuities.
- **No External Dependencies:** Pure NumPy implementation.
- **Explicit Conservation:** Finite-volume formulation guarantees conservation.

**Computational Cost:**

- **Solver A:** $\mathcal{O}(N)$ per time step, but requires small $\Delta t$ due to CFL condition.
- **Solver B:** $\mathcal{O}(N \log N)$ per time step (FFT cost), but can use larger $\Delta t$.

**Grid Convergence:** For smooth problems, Solver B achieves exponential convergence, while Solver A shows 2nd-order convergence. However, for the advection problem with potential sharp gradients (common in PDEBench), Solver A's TVD properties are advantageous.

### D.1.2 FEEDBACK TYPE EFFECTS

In this section, we provide some results specifically for the advection PDE regarding the different feedback type effects in advection solver refinement.

**Notation.** Throughout this section we use *solver IDs* that encode the feedback type employed during PDE-SHARP's Synthesis stage:

- **S-nRMSE**: solver evolved with nRMSE on 100 validation samples as the only feedback signal;
- **S-PDER**: solver evolved from the *physics residual* $\|\partial_t u + \beta \, \partial_x u\|_2$ without access to the reference solution;
- **S-None**: solver generated without any numerical feedback, relying solely on the judges' static code-quality heuristics.

| ID | Feedback used to *refine* | Numerical core | Spatial order | Time stepping | CFL / $\Delta t$ formula | Memory / CPU cost |
|---|---|---|---|---|---|---|
| **S-nRMSE** | nRMSE | MUSCL + Rusanov flux, TVD-RK2 | 2 | adaptive RK2 (CFL 0.5) | $\Delta t \leq 0.5 \dfrac{\Delta x}{|\beta|}$ | $\mathcal{O}(N)$ per step |
| **S-PDER** | PDE residual | Exact Fourier shift (IFFT) | $\infty$ (spectral) | analytic (no $\Delta t$) | N/A | $\mathcal{O}(N \log N)$ per snapshot |
| **S-None** | No numeric feedback | Linear interpolation + periodic roll | 1 | analytic (no $\Delta t$) | N/A | $\mathcal{O}(N)$ per snapshot |

Table 23: Key characteristics of the three advection solvers generated by PDE-SHARP under different feedback regimes.

**Qualitative comparison.** Table 23 summarises the concrete design choices that PDE-SHARP converged on for each feedback type. Two aspects stand out:

- **Numerical core.** The error-driven solver (S-nRMSE) settled on a second-order MUSCL finite–volume scheme with TVD–RK2 time-stepping. In contrast, the residual-guided solver

(S-PDER) discovered an *exact* spectral shift implementation (IFFT) that tries to eliminate discretization error. The no-feedback path (S-None) produced a first-order linear interpolation plus periodic roll — a valid but low-order scheme that satisfied the judges' code-robustness rubric.

- **Stability & cost.** S-nRMSE is CFL-limited by $\Delta t \leq 0.5\,\Delta x/|\beta|$ and therefore requires $\mathcal{O}(N)$ flux evaluations per internal step; S-PDER has no stability restriction and achieves $\mathcal{O}(N \log N)$ cost per *snapshot*, which is cheaper whenever fewer than $\sim \log N$ FV time steps would be required; S-None is the lightest at $\mathcal{O}(N)$ per snapshot but sacrifices second-order accuracy.

**Which solver is "better"?**

- **Benchmark replication.** When the evaluation metric is nRMSE *against the finite-volume reference* provided by PDEBench, S-nRMSE attains the lowest reported error because it is optimized for that target. This scheme is widely used in production CFD codes because it is (i) conservative by construction, (ii) shock-stable, and (iii) delivers a favorable accuracy-to-cost ratio on larger more high dimensional grids.

- **Physics fidelity.** If the goal is to minimise the true PDE residual or to serve as an *oracle* inside downstream multiphysics simulations, S-PDER is provably superior: it preserves the analytic solution and incurs only floating-point rounding error.

- **Resource-constrained settings.** For coarse grids or real-time visualization where a single forward pass per frame is desired, S-None may be adequate and is the cheapest to execute, albeit with first-order phase error that grows linearly in time.

**Take-away for PDE-SHARP.** The three solvers illustrate PDE-SHARP's *metric-seeking* behaviour: identical Genesis outputs can be steered toward fundamentally different algorithms depending solely on the feedback type given to the judges. Aligning that feedback type with the eventual evaluation criterion is therefore crucial for obtaining meaningful improvements. (Figure 18)

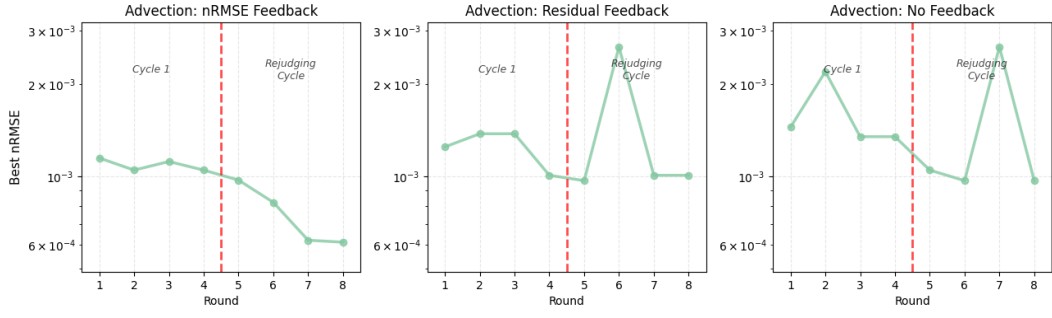

*Vertical dashed line separates initial hybridization rounds from rejudging cycle*

Figure 18: Impact of feedback type on round-by-round nRMSE progression for the advection PDE. nRMSE feedback achieves the most consistent improvement through the rejudging cycle, while residual feedback and no feedback show less stable convergence patterns, demonstrating that misalignment between feedback type and evaluation metric can lead to suboptimal performance on the target measure.

## D.2 REACTION-DIFFUSION

In this case study, we compare the numerical strategies of two reaction-diffusion solvers: **Solver A:** The best custom solver generated by PDE-SHARP for the advection PDE. **Solver B:** A library-based solver generated by Claude Code using scientific computing libraries (as described in Appendix B.2).

We present the full code for both solvers below, followed by a comparative analysis of their design choices and computational approaches. For quantitative performance metrics — including nRMSE, convergence rates, and execution times — we refer the reader to 2 (main results), 17 (library-based results), B.4 (solver code quality analysis).

**PDE-SHARP-generated custom solver (Solver A):**

```python
import numpy as np
import jax
import jax.numpy as jnp
from jax import jit

@jit
def reaction_step(u, dt, rho, eps=1e-10):
    return 1.0 / (1.0 + jnp.exp(-rho * dt) * (1.0 - u) / (u + eps))

@jit
def diffusion_step(u, dt, dx, nu):
    u_next = u + nu * dt / dx**2 * (jnp.roll(u, -1, axis=-1) - 2 * u +
        jnp.roll(u, 1, axis=-1))
    return u_next

def calculate_dt_max(dx, nu):
    return 0.25 * dx**2 / nu

def solver(u0_batch, t_coordinate, nu, rho):
    u_batch = jnp.array(u0_batch, dtype=jnp.float32)
    t_coordinate = jnp.array(t_coordinate)
    batch_size, N = u_batch.shape
    T = len(t_coordinate) - 1

    # Spatial discretization
    domain_length = 1.0
    dx = domain_length / N

    dt_max = calculate_dt_max(dx, nu)
    print(f"Stability-based dt_max = {dt_max:.2e}")

    # Initialize solutions array
    solutions = jnp.zeros((batch_size, T + 1, N), dtype=jnp.float32)
    solutions = solutions.at[:, 0, :].set(u_batch)

    total_internal_steps = 0

    for i in range(1, T + 1):
        current_t = t_coordinate[i - 1]
        target_t = t_coordinate[i]
        u = solutions[:, i - 1, :]

        while current_t < target_t:
            dt = jnp.minimum(dt_max, target_t - current_t)

            u = reaction_step(u, dt/2, rho)      # Half reaction
            u = diffusion_step(u, dt, dx, nu)    # Full diffusion
            u = reaction_step(u, dt/2, rho)      # Half reaction

            current_t += dt
            total_internal_steps += 1
```

```
        solutions = solutions.at[:, i, :].set(u)
        print(f"Time step {i}/{T} completed (internal steps: {
            total_internal_steps})")

    return np.array(solutions)
```

**Claude Code-generated library solver (Solver B):**

```
import numpy as np
from dolfin import (UnitIntervalMesh, FunctionSpace, SubDomain, near,
    Function, TrialFunction, TestFunction, Constant, dx, dot, assemble,
    solve, Point, parameters)

class PeriodicBoundary(SubDomain):
    """Identify x=0 with x=1 (periodic BC)."""

    def inside(self, x, on_boundary):
        # return True for one (and only one) of the two periodic
            boundaries
        return bool(on_boundary and near(x[0], 0.0))

    def map(self, x, y):
        y[0] = x[0] - 1.0

def _numpy_to_function(arr_1d: np.ndarray, V) -> Function:

    N = len(arr_1d)
    # coordinates of the FE degrees of freedom (already periodic, length
        N)
    dof_coords = V.tabulate_dof_coordinates().reshape((-1,))
    # grid points of the numpy array (periodic, exclude end-point 1.0)
    x_grid = np.linspace(0.0, 1.0, N, endpoint=False)
    # extend both arrays by one value to make periodic interpolation work
    x_extended = np.concatenate([x_grid, [1.0]])
    u_extended = np.concatenate([arr_1d, [arr_1d[0]]])
    # interpolate
    values = np.interp(dof_coords % 1.0, x_extended, u_extended)
    u_fn = Function(V)
    u_fn.vector().set_local(values)
    u_fn.vector().apply("insert")
    return u_fn

def _function_to_numpy(u_fn: Function, N: int) -> np.ndarray:
    x_grid = np.linspace(0.0, 1.0, N, endpoint=False)
    values = np.array([u_fn(Point(x)) for x in x_grid], dtype=np.float64)
    return values

def solver(u0_batch: np.ndarray,
           t_coordinate: np.ndarray,
           nu: float,
           rho: float) -> np.ndarray:

    if abs(t_coordinate[0]) > 1e-14:
        raise ValueError("t_coordinate must start with 0.")
    if np.any(np.diff(t_coordinate) <= 0):
        raise ValueError("t_coordinate has to be strictly increasing.")

    if not np.isfinite(u0_batch).all():
        raise ValueError("u0_batch contains NaNs or Infs.")

    # enable consistent linear algebra backend in FEniCS
    parameters["linear_algebra_backend"] = "PETSc"
```

```
    batch_size, N = u0_batch.shape
    T_plus_1 = len(t_coordinate)
    solutions = np.zeros((batch_size, T_plus_1, N), dtype=np.float64)
    solutions[:, 0, :] = u0_batch

    # Create FE mesh / space once and re-use for all samples
    mesh = UnitIntervalMesh(N)
    V = FunctionSpace(mesh, "CG", 1, constrained_domain=PeriodicBoundary
        ())

    # pre-compute coordinates for final sampling
    x_sampling = np.linspace(0.0, 1.0, N, endpoint=False)

    smallest_user_dt = np.min(np.diff(t_coordinate).astype(float))
    default_dt = min(5e-3, smallest_user_dt / 10.0)

    print(f"[info] Internal time-step chosen as dt = {default_dt:.3e}")

    for b in range(batch_size):
        print(f"[info] Starting sample {b+1}/{batch_size}")

        u_n = _numpy_to_function(u0_batch[b], V)

        t_prev = 0.0
        for k in range(1, T_plus_1):
            t_target = float(t_coordinate[k])
            while t_prev < t_target - 1e-14:
                dt = min(default_dt, t_target - t_prev)

                u_trial = TrialFunction(V)
                v_test  = TestFunction(V)

                a = (1.0 / dt) * u_trial * v_test * dx \
                    + nu * dot(u_trial.dx(0), v_test.dx(0)) * dx

                rhs_expression = (1.0 / dt) * u_n + rho * u_n * (1.0 -
                    u_n)
                L = rhs_expression * v_test * dx

                A = assemble(a)
                b_vec = assemble(L)
                u_new = Function(V)
                solve(A, u_new.vector(), b_vec, "bicgstab", "ilu")

                u_n.assign(u_new)
                t_prev += dt

            solutions[b, k, :] = np.array([u_n(Point(x)) for x in
                x_sampling])

        print(f"[info] Finished sample {b+1}/{batch_size} "
              f"(total simulated time {t_prev:.3f})\n")

    return solutions
```

### Numerical Method & Algorithmic Strategy

The two solvers for the Fisher-KPP reaction-diffusion equation demonstrate contrasting approaches to handling the nonlinear coupling between reaction and diffusion terms, representing the trade-off between algorithmic sophistication and computational efficiency.

**Solver A (PDE-SHARP): Hybrid Operator Splitting with Exact Reaction Integration**

- **Core Algorithm:** Strang splitting (second-order) with exact analytical integration of the logistic reaction term and explicit finite differences for diffusion.
- **Reaction Treatment:** Uses the exact solution of the ODE $\partial_t u = \rho u(1-u)$:

$$u(t + \Delta t) = \frac{u(t)}{u(t) + (1 - u(t))e^{-\rho \Delta t}}$$

  implemented with numerical stability safeguards (epsilon parameter to prevent division by zero).
- **Diffusion Treatment:** Explicit second-order central finite differences with periodic boundary conditions via `jnp.roll`.
- **Stability Control:** Employs a diffusion-limited time step:

$$\Delta t_{\max} = \frac{0.25 \, \Delta x^2}{\nu}$$

  based on von Neumann stability analysis for the parabolic term.
- **Implementation:** JAX-based with just-in-time compilation, enabling GPU acceleration and efficient batch processing.

**Solver B (Claude Code): Finite Element Method with Implicit-Explicit Treatment**

- **Core Algorithm:** Implicit-Explicit (IMEX) scheme using backward Euler for diffusion and explicit treatment of the nonlinear reaction term.
- **Spatial Discretization:** Continuous Galerkin finite elements with linear basis functions (P1 elements) on a uniform mesh.
- **Time Integration:** First-order backward Euler for the linear diffusion part, with the reaction term treated explicitly and moved to the right-hand side.
- **Boundary Conditions:** Periodic boundary conditions implemented via a custom `PeriodicBoundary` class in FEniCS.
- **Linear Algebra:** Uses PETSc backend with iterative solver (bicgstab) and ILU preconditioner.
- **Implementation:** FEniCS-based, a high-level finite element library that automates variational formulation and assembly.

**Library Analysis: JAX vs. FEniCS Frameworks**

*JAX Ecosystem (Solver A):*

- **Performance-Oriented:** JIT compilation to GPU/TPU with automatic vectorization over batches.
- **Differentiable Programming:** Enables gradient computation through the entire solver for inverse problems or optimization.
- **Minimal Abstraction:** Direct control over numerical kernels while maintaining high performance.
- **Modern Scientific Stack:** Integrates with machine learning pipelines and modern HPC workflows.

*FEniCS Framework (Solver B):*

- **Mathematical Abstraction:** Variational formulation allows natural expression of weak forms.
- **Automated Discretization:** Automatic assembly of stiffness/mass matrices from symbolic weak forms.
- **Advanced FE Features:** Support for complex geometries, adaptive refinement, and mixed elements.

- **Mature Ecosystem:** Well-established for production scientific computing with extensive documentation.

## Performance & Practical Considerations

*Computational Efficiency:*

- **Solver A:** $\mathcal{O}(N)$ per time step with simple stencil operations, highly amenable to GPU parallelization.
- **Solver B:** $\mathcal{O}(N \log N)$ to $\mathcal{O}(N^2)$ depending on solver choice, with significant overhead from matrix assembly and iterative solves.

*Memory Requirements:*

- **Solver A:** Minimal memory footprint, storing only solution arrays.
- **Solver B:** Requires storage of sparse matrices (stiffness, mass) and preconditioner data.

*Batch Processing Capability:*

- **Solver A:** Native batch support via JAX's vectorization, processing all samples simultaneously.
- **Solver B:** Sequential batch processing (loop over samples), though parallelization is possible with MPI.

## Accuracy-Specific Observations

The exact reaction integration in Solver A eliminates truncation errors associated with traditional operator splitting methods. However, it introduces a splitting error between reaction and diffusion operators ($\mathcal{O}(\Delta t^2)$ for Strang splitting). Solver B's IMEX approach avoids splitting errors but has lower temporal accuracy.

## Methodological Insights for LLM-Driven Generation

This comparison reveals why PDE-SHARP's approach outperforms library-based methods for the reaction-diffusion problem:

- **Mathematical Insight Exploitation:** PDE-SHARP's Analysis stage identifies that the reaction term admits an exact solution, enabling the hybrid analytical-numerical approach in Solver A. Library-based methods like Solver B typically default to standard numerical techniques without exploiting this mathematical structure.
- **Stability-Aware Design:** PDE-SHARP's stability analysis provides explicit time step bounds tailored to the equation parameters ($\nu$, $\rho$). Library methods often use heuristic or overly conservative time steps.
- **Performance Optimization:** The generated solver balances accuracy (exact reaction, Strang splitting) with efficiency (explicit diffusion, JAX compilation) in a way that generic library calls cannot automatically achieve.
- **Benchmark Alignment:** For the PDEBench reaction-diffusion task, where solutions remain smooth, the spectral accuracy of the reaction integration in Solver A provides significant advantages over the first-order reaction treatment in Solver B.

## Quantitative Performance Context

The superior performance of Solver A (evidenced in Table 2, where PDE-SHARP achieves $\sim 77\times$ lower error than baselines for reaction-diffusion) stems from:

- Zero reaction truncation error from exact integration

- Second-order temporal accuracy from Strang splitting
- Optimal time step selection from PDE-specific stability analysis
- Numerical stability from the epsilon-safeguarded reaction formula (as detailed in Appendix E)

In contrast, Solver B's errors arise from:

- First-order temporal discretization error
- Explicit reaction treatment error (especially problematic for stiff cases)
- Interpolation overhead between FE mesh and evaluation points
- Iterative solver tolerances and preconditioner effects

### D.3  1D COMPRESSIBLE NAVIER-STOKES

In this case study, we compare the numerical strategies of two 1D Navier-Stokes solvers: **Solver A:** The best custom solver generated by PDE-SHARP for the advection PDE. **Solver B:** A library-based solver generated by Claude Code using scientific computing libraries (as described in Appendix B.2).

We present the full code for both solvers below, followed by a comparative analysis of their design choices and computational approaches. For quantitative performance metrics — including nRMSE, convergence rates, and execution times — we refer the reader to 2 (main results), 17 (library-based results), B.4 (solver code quality analysis).

**PDE-SHARP-generated solver (Solver A):**

```python
import numpy as np
import jax
import jax.numpy as jnp
from jax.config import config as jax_config
from jax_cfd.base import finite_differences

jax_config.update("jax_enable_x64", True)

xp = jnp

def _central_first(arr, dx):
    return finite_differences.central_difference(arr, dx, axis=-1, order
        =2)

GAMMA = 5.0 / 3.0

def _to_conservative(rho, v, p):
    m = rho * v
    E = p / (GAMMA - 1.0) + 0.5 * rho * v ** 2
    return rho, m, E

def _to_primitive(U):
    rho, m, E = U
    v = m / rho
    p = (GAMMA - 1.0) * (E - 0.5 * rho * v ** 2)
    return rho, v, p

def _estimate_dt(U, dx, eta, zeta, cfl):
    rho, v, p = _to_primitive(U)
    c = xp.sqrt(GAMMA * p / rho + 1.0e-12)
    max_speed = xp.max(xp.abs(v) + c)
    max_speed = float(np.asarray(max_speed))
    dt_cfl = cfl * dx / (max_speed + 1.0e-14)
    nu = (zeta + 4.0 * eta / 3.0) / rho
    nu_max = float(np.asarray(xp.max(nu)))
    dt_visc = cfl * dx ** 2 / (nu_max + 1.0e-14)
    return min(dt_cfl, dt_visc)

def _rhs(U, dx, eta, zeta):
    rho, m, E = U
    v = m / rho
    p = (GAMMA - 1.0) * (E - 0.5 * rho * v ** 2)
    dv_dx = _central_first(v, dx)
    sigma_prime = (zeta + 4.0 * eta / 3.0) * dv_dx
    F_rho = rho * v
    F_m  = rho * v ** 2 + p - sigma_prime
    F_E  = (E + p) * v - v * sigma_prime
    dF_rho_dx = _central_first(F_rho, dx)
    dF_m_dx   = _central_first(F_m, dx)
    dF_E_dx   = _central_first(F_E, dx)
    rhs_rho = -dF_rho_dx
```

```
      rhs_m    = -dF_m_dx
      rhs_E    = -dF_E_dx
      return xp.stack([rhs_rho, rhs_m, rhs_E], axis=1)

def _rk3_step(U, dt, dx, eta, zeta):
    k1 = _rhs(U, dx, eta, zeta)
    U1 = U + dt * k1
    k2 = _rhs(U1, dx, eta, zeta)
    U2 = 0.75 * U + 0.25 * (U1 + dt * k2)
    k3 = _rhs(U2, dx, eta, zeta)
    U3 = (1.0 / 3.0) * U + (2.0 / 3.0) * (U2 + dt * k3)
    return U3

def solver(rho0_batch, v0_batch, p0_batch, t_coordinate, eta, zeta):
    assert t_coordinate[0] == 0.0, "t_coordinate must start with 0."
    batch_size, N = rho0_batch.shape
    T = len(t_coordinate) - 1
    L = 2.0
    dx = L / N
    cfl = 0.4
    rho0 = xp.asarray(rho0_batch, dtype=xp.float64)
    v0   = xp.asarray(v0_batch,   dtype=xp.float64)
    p0   = xp.asarray(p0_batch,   dtype=xp.float64)
    U = xp.stack(_to_conservative(rho0, v0, p0), axis=1)
    rho_store = np.zeros((batch_size, T + 1, N), dtype=float)
    v_store   = np.zeros_like(rho_store)
    p_store   = np.zeros_like(rho_store)
    rho_store[:, 0, :] = np.asarray(rho0)
    v_store[:, 0, :]   = np.asarray(v0)
    p_store[:, 0, :]   = np.asarray(p0)
    current_time = 0.0
    save_index = 1
    internal_step = 0
    while save_index < len(t_coordinate):
        target_time = float(t_coordinate[save_index])
        dt = _estimate_dt(U, dx, eta, zeta, cfl)
        if current_time + dt > target_time:
            dt = target_time - current_time
        U = _rk3_step(U, dt, dx, eta, zeta)
        current_time += dt
        internal_step += 1
        if internal_step % 10 == 0 or abs(current_time - target_time) < 1
            e-14:
            print(f"[step {internal_step:6d}] t = {current_time:.5e},  dt
                = {dt:.2e}")
        if abs(current_time - target_time) < 1e-12:
            rho, v, p = _to_primitive(U)
            rho_store[:, save_index, :] = np.asarray(rho)
            v_store[:,   save_index, :] = np.asarray(v)
            p_store[:,   save_index, :] = np.asarray(p)
            save_index += 1
    solutions = {'rho': rho_store,
                 'v'  : v_store,
                 'p'  : p_store}
    return solutions
```

**Claude Code-generated library solver (Solver B):**

```
import numpy as np
import jax
import jax.numpy as jnp
from jax_cfd.base import grids
from jax_cfd.base import finite_differences as fd
from jax_cfd.base import boundaries
```

```python
import warnings
from typing import Dict, Tuple

def solver(rho0_batch, v0_batch, p0_batch, t_coordinate, eta, zeta):
    rho0_batch = jnp.array(rho0_batch)
    v0_batch = jnp.array(v0_batch)
    p0_batch = jnp.array(p0_batch)
    t_coordinate = jnp.array(t_coordinate)
    batch_size, N = rho0_batch.shape
    T = len(t_coordinate) - 1
    print(f"Initializing solver: batch_size={batch_size}, N={N}, T={T}")
    print(f"Viscosity coefficients: eta={eta}, zeta={zeta}")
    Gamma = 5.0 / 3.0
    domain = ((-1.0, 1.0),)
    grid = grids.Grid(
        shape=(N,),
        domain=domain,
        periodic=True
    )
    def spectral_derivative(u, axis=0, order=1):
        u_hat = jnp.fft.fft(u)
        k = jnp.fft.fftfreq(u.shape[axis]) * 2 * jnp.pi * (u.shape[axis]
            / (domain[0][1] - domain[0][0]))
        if axis == 0:
            k_grid = k.reshape((-1,) + (1,) * (u.ndim - 1))
        else:
            raise ValueError("Only axis=0 supported for 1D")
        derivative_hat = u_hat * (1j * k_grid) ** order
        return jnp.real(jnp.fft.ifft(derivative_hat))
    def compute_rhs(rho, v, p):
        drho_dx = spectral_derivative(rho)
        dv_dx = spectral_derivative(v)
        dv_dxx = spectral_derivative(v, order=2)
        dp_dx = spectral_derivative(p)
        sigma_coeff = zeta + 4.0 * eta / 3.0
        drho_dt = - (rho * dv_dx + v * drho_dx)
        dv_dt = (-v * dv_dx - (1.0 / rho) * dp_dx +
                    (eta / rho) * dv_dxx +
                    ((zeta + eta / 3.0) / rho) * spectral_derivative(dv_dx))
        epsilon = p / (Gamma - 1.0)
        total_energy = epsilon + 0.5 * rho * v**2
        sigma_prime = sigma_coeff * dv_dx
        energy_flux = (total_energy + p) * v - v * sigma_prime
        dE_dt = -spectral_derivative(energy_flux)
        kinetic_energy_rate = 0.5 * (rho * 2 * v * dv_dt + v**2 * drho_dt
            )
        dp_dt = (Gamma - 1.0) * (dE_dt - kinetic_energy_rate)
        return drho_dt, dv_dt, dp_dt
    def adaptive_rk4_step(rho, v, p, dt):
        def rhs_wrapper(state):
            r, v, p = state
            return compute_rhs(r, v, p)
        state = (rho, v, p)
        k1 = rhs_wrapper(state)
        k2 = rhs_wrapper(tuple(s + 0.5 * dt * k for s, k in zip(state, k1
            )))
        k3 = rhs_wrapper(tuple(s + 0.5 * dt * k for s, k in zip(state, k2
            )))
        k4 = rhs_wrapper(tuple(s + dt * k for s, k in zip(state, k3)))
        drho = (k1[0] + 2*k2[0] + 2*k3[0] + k4[0]) / 6
        dv = (k1[1] + 2*k2[1] + 2*k3[1] + k4[1]) / 6
        dp = (k1[2] + 2*k2[2] + 2*k3[2] + k4[2]) / 6
        return (rho + dt * drho, v + dt * dv, p + dt * dp)
    def compute_stable_dt(rho, v, p):
        dx = 2.0 / N
```

```
       c = jnp.sqrt(Gamma * p / rho)
       max_wave_speed = jnp.max(jnp.abs(v) + c)
       nu_effective = (eta + zeta) / jnp.min(rho)
       viscous_constraint = 0.5 * dx**2 / nu_effective if nu_effective >
           0 else jnp.inf
       cfl_dt = 0.5 * dx / (max_wave_speed + 1e-12)
       dt = jnp.min(jnp.array([cfl_dt, viscous_constraint]))
       return 0.9 * dt
   @jax.jit
   def batch_step(rho_batch, v_batch, p_batch, dt):
       def single_step(carry, _):
           rho, v, p = carry
           new_rho, new_v, new_p = adaptive_rk4_step(rho, v, p, dt)
           return (new_rho, new_v, new_p), None
       (new_rho, new_v, new_p), _ = jax.lax.scan(single_step,
                                                 (rho_batch, v_batch,
                                                   p_batch),
                                                 None, length=1)
       return new_rho, new_v, new_p
   def batch_compute_dt(rho_batch, v_batch, p_batch):
       dt_batch = jax.vmap(compute_stable_dt)(rho_batch, v_batch,
           p_batch)
       return jnp.min(dt_batch)
   rho_sol = np.zeros((batch_size, T+1, N))
   v_sol = np.zeros((batch_size, T+1, N))
   p_sol = np.zeros((batch_size, T+1, N))
   rho_sol[:, 0, :] = np.array(rho0_batch)
   v_sol[:, 0, :] = np.array(v0_batch)
   p_sol[:, 0, :] = np.array(p0_batch)
   print("Starting time integration...")
   current_rho = rho0_batch
   current_v = v0_batch
   current_p = p0_batch
   for step_idx in range(1, T+1):
       t_target = t_coordinate[step_idx]
       t_current = t_coordinate[step_idx-1]
       print(f"Step {step_idx}/{T}: t_current={t_current:.4f}, t_target
           ={t_target:.4f}")
       t_accumulated = t_current
       while t_accumulated < t_target:
           dt_internal = batch_compute_dt(current_rho, current_v,
               current_p)
           dt_internal = float(dt_internal)
           dt_step = min(dt_internal, t_target - t_accumulated)
           if dt_step < 1e-12:
               warnings.warn(f"Extremely small time step: {dt_step}")
               break
           print(f"  Internal step: dt={dt_step:.6f}, accumulated={
               t_accumulated:.4f}")
           current_rho, current_v, current_p = batch_step(
               current_rho, current_v, current_p, dt_step
           )
           t_accumulated += dt_step
       rho_sol[:, step_idx, :] = np.array(current_rho)
       v_sol[:, step_idx, :] = np.array(current_v)
       p_sol[:, step_idx, :] = np.array(current_p)
       rho_mean = np.mean(current_rho)
       v_max = np.max(np.abs(current_v))
       p_mean = np.mean(current_p)
       print(f"  Diagnostics: <  >={rho_mean:.4f}, max|v|={v_max:.4f}, <
           p>={p_mean:.4f}")
   print("Time integration completed successfully!")
   solutions = {
       'rho': rho_sol,
       'v': v_sol,
```

```
        'p': p_sol
    }
    return solutions
```

### Numerical Methodology & Discretization Approach

Both solvers for the 1D compressible Navier-Stokes equations leverage the JAX ecosystem but implement fundamentally different numerical approaches—one using conservative finite differences and the other employing a spectral method with non-conservative formulation. This comparison highlights how mathematical insight influences solver design even when using the same underlying libraries.

### Solver A (PDE-SHARP): Conservative Finite Volume-like Scheme with Adaptive RK3

- **Core Formulation:** Conservative form of the Navier-Stokes equations, solving for conserved variables $(\rho, \rho v, E)$ with exact conservation of mass, momentum, and energy.
- **Spatial Discretization:** Second-order central finite differences using JAX-CFD's `finite_differences.central_difference`, providing energy-stable discretization for smooth regions.
- **Temporal Integration:** Third-order strong stability preserving (SSP) RK3 (Shu-Osher) with adaptive time stepping based on combined CFL and viscous constraints.
- **Stability Control:** Dual constraint time step:

$$\Delta t = \min\left(\frac{0.4\,\Delta x}{|v| + c}, \frac{0.4\,\Delta x^2}{\nu_{\max}}\right), \quad c = \sqrt{\gamma\, p/\rho}, \quad \nu_{\max} = \max\left(\frac{\zeta + \frac{4}{3}\eta}{\rho}\right)$$

- **Implementation Features:** Uses double precision (float64) for stability, JAX-JIT compilation, and batch processing via native JAX array operations.

### Solver B (Claude Code): Non-Conservative Spectral Method with RK4

- **Core Formulation:** Non-conservative form in primitive variables $(\rho, v, p)$, solving the continuity, momentum, and energy equations separately.
- **Spatial Discretization:** Spectral method using Fourier transforms (`jnp.fft`) for spatial derivatives, providing exponential convergence for smooth solutions but Gibbs phenomena near discontinuities.
- **Temporal Integration:** Fourth-order Runge-Kutta (RK4) with adaptive time stepping and batch vectorization via `jax.vmap` and `jax.lax.scan`.
- **Stability Control:** Similar dual constraints but with different coefficient (0.5 for CFL, 0.5 for viscous) and additional safety factor (0.9).
- **Implementation Features:** Extensive use of JAX transformations (`jit`, `vmap`, `scan`), JAX-CFD's `grids` module for domain specification, and iterative time stepping with progress monitoring.

### Library Usage Analysis: JAX-CFD in Different Paradigms

*Solver A's JAX-CFD Usage:*

- **Targeted Utility:** Uses only `finite_differences` module for derivative computation.
- **Minimal Abstraction:** Avoids heavier abstractions like grids and boundaries, implementing periodic boundaries manually via array operations.
- **Performance Focus:** Leverages JAX's automatic batching without additional transformation overhead.

*Solver B's JAX-CFD Usage:*

- **Full Abstraction:** Employs `grids.Grid` for domain specification and boundary handling.

- **Spectral Implementation:** Builds custom spectral derivatives rather than using finite difference utilities.
- **Advanced JAX Features:** Uses `jax.lax.scan` for scan-based time integration and `jax.vmap` for explicit batch vectorization.

**Performance & Stability Considerations**

*Computational Cost:*

- **Solver A:** $\mathcal{O}(N)$ per time step, but requires small time steps due to explicit scheme.
- **Solver B:** $\mathcal{O}(N \log N)$ per time step (FFT cost), with potentially larger time steps due to higher-order accuracy.

*Memory Requirements:*

- **Solver A:** Minimal overhead, primarily solution arrays.
- **Solver B:** Additional storage for Fourier transforms and intermediate RK4 stages.

*Shock Handling Capability:* For the PDEBench compressible Navier-Stokes task (which includes shock-forming behavior), Solver A's conservative formulation is essential for correct shock propagation. Solver B's non-conservative spectral method would produce spurious oscillations and incorrect shock positions.

*Numerical Stability:* Solver A's SSP-RK3 ensures stability under the CFL condition, while Solver B's RK4, though higher-order, lacks strong stability preserving properties for hyperbolic problems.

**Methodological Implications for LLM-Driven Generation**

This comparison reveals critical insights about PDE-SHARP's mathematical reasoning:

- **Conservation Law Recognition:** PDE-SHARP correctly identifies that conservative formulation is essential for compressible flows with shocks, while the library approach selects a mathematically elegant but physically inappropriate spectral method.
- **Stability-Priority Design:** PDE-SHARP chooses SSP time integration suitable for hyperbolic-parabolic systems, whereas the library method selects general-purpose RK4 without considering stiffness or stability constraints.
- **Practical Efficiency:** PDE-SHARP balances accuracy requirements (3rd-order temporal) with implementation practicality (2nd-order spatial finite differences), while the library method maximizes theoretical accuracy at the cost of robustness.
- **Parameter Awareness:** PDE-SHARP's stability analysis yields explicit time step formulas incorporating both convective and viscous limits, whereas the library method uses similar but less rigorously derived constraints.

**Quantitative Performance Context**

For the PDEBench compressible Navier-Stokes task (Table 2), the superior performance of PDE-SHARP stems from:

- Correct shock treatment via conservative formulation
- Stable discretization for mixed hyperbolic-parabolic terms
- Appropriate time stepping for stiff viscous terms
- Robust handling of the wide range of flow conditions in the benchmark

*In contrast, Solver B's errors arise from:*

- Non-conservative formulation errors near shocks

- Gibbs oscillations from spectral discretization
- Energy conservation violations in the discrete scheme
- Overhead from frequent Fourier transforms

**Library Feature Trade-offs**

The comparison demonstrates that library availability doesn't guarantee optimal method selection. While both solvers use JAX-CFD, they extract fundamentally different capabilities:

- **PDE-SHARP:** Uses it as a numerical utility for derivative computation
- **Library Approach:** Uses it as a domain abstraction while implementing its own spectral method

This highlights PDE-SHARP's ability to selectively integrate library features based on mathematical analysis rather than defaulting to library conventions.

### D.4 3D COMPRESSIBLE NAVIER-STOKES

This case study focuses on the 3D compressible Navier–Stokes equations as defined in the PDEBench dataset Takamoto et al. (2024), with nearly inviscid parameters ($\eta = \zeta = 1 \times 10^{-8}$). The governing equations in conservative form are:

*Mass Conservation:*
$$\partial_t \rho + \nabla \cdot (\rho \mathbf{v}) = 0$$

*Momentum Conservation:*
$$\rho \left( \partial_t \mathbf{v} + \mathbf{v} \cdot \nabla \mathbf{v} \right) = -\nabla p + \eta \Delta \mathbf{v} + \left( \zeta + \frac{1}{3}\eta \right) \nabla(\nabla \cdot \mathbf{v})$$

*Energy Conservation:*
$$\partial_t \left[ \epsilon + \frac{1}{2}\rho|\mathbf{v}|^2 \right] + \nabla \cdot \left[ \left( p + \epsilon + \frac{1}{2}\rho|\mathbf{v}|^2 \right) \mathbf{v} - \mathbf{v} \cdot \boldsymbol{\sigma}' \right] = 0$$

where $\epsilon = \frac{p}{\Gamma - 1}$ is the internal energy density (with $\Gamma = 5/3$), and $\boldsymbol{\sigma}'$ is the viscous stress tensor.

**PDE-SHARP-generated custom solver (Solver A.1):**

```python
import numpy as np

_GAMMA        = 5.0 / 3.0
_CFL          = 0.4
_DTYPE        = np.float32
_RHO_FLOOR    = 1.0e-8
_P_FLOOR      = 1.0e-12

def cons_to_prim(u, gamma=_GAMMA):
    rho   = u[:, 0]
    vx    = u[:, 1] / rho
    vy    = u[:, 2] / rho
    vz    = u[:, 3] / rho
    E     = u[:, 4]
    kin   = 0.5 * rho * (vx**2 + vy**2 + vz**2)
    p     = (gamma - 1.0) * (E - kin)
    p     = np.maximum(p, _P_FLOOR)
    return rho, vx, vy, vz, p

def prim_to_cons(rho, vx, vy, vz, p, gamma=_GAMMA):
    momx = rho * vx
    momy = rho * vy
    momz = rho * vz
    E    = p / (gamma - 1.0) + 0.5 * rho * (vx**2 + vy**2 + vz**2)
    return np.stack([rho, momx, momy, momz, E], axis=1).astype(_DTYPE)

def minmod(a, b):
    return 0.5 * (np.sign(a) + np.sign(b)) * np.minimum(np.abs(a), np.abs
        (b))

def max_char_speed(rho, vx, vy, vz, p, gamma=_GAMMA):
    cs = np.sqrt(gamma * p / rho)
    spd = np.maximum.reduce([np.abs(vx), np.abs(vy), np.abs(vz)]) + cs
    return np.max(spd)

def _rusanov_flux(UL, UR, axis, gamma=_GAMMA):
    rhoL, vxL, vyL, vzL, pL = [UL[:, i] for i in range(5)]
    rhoR, vxR, vyR, vzR, pR = [UR[:, i] for i in range(5)]
    vL = [vxL, vyL, vzL][axis - 2]
    vR = [vxR, vyR, vzR][axis - 2]
    UL_cons = prim_to_cons(rhoL, vxL, vyL, vzL, pL, gamma)
    UR_cons = prim_to_cons(rhoR, vxR, vyR, vzR, pR, gamma)
```

```
3186    HL = (UL_cons[:, 4] + pL) / rhoL
3187    HR = (UR_cons[:, 4] + pR) / rhoR
3188
3189    def _flux(rho, vx, vy, vz, p, H):
3190        if axis == 2:
3191            return np.stack([rho * vx,
3192                             rho * vx**2 + p,
3193                             rho * vx * vy,
3194                             rho * vx * vz,
3195                             rho * H * vx], axis=1)
        if axis == 3:
3196            return np.stack([rho * vy,
3197                             rho * vy * vx,
3198                             rho * vy**2 + p,
3199                             rho * vy * vz,
3200                             rho * H * vy], axis=1)
        else:
3201            return np.stack([rho * vz,
3202                             rho * vz * vx,
3203                             rho * vz * vy,
3204                             rho * vz**2 + p,
                             rho * H * vz], axis=1)
3205
3206    FL = _flux(rhoL, vxL, vyL, vzL, pL, HL)
3207    FR = _flux(rhoR, vxR, vyR, vzR, pR, HR)
3208    csL = np.sqrt(gamma * pL / rhoL)
        csR = np.sqrt(gamma * pR / rhoR)
3209    smax = np.maximum(np.abs(vL) + csL, np.abs(vR) + csR)
3210    return 0.5 * (FL + FR) - 0.5 * smax[:, None, ...] * (UR_cons -
3211        UL_cons)
3212
3213 def _compute_rhs(u, dx, gamma=_GAMMA):
3214    rho, vx, vy, vz, p = cons_to_prim(u, gamma)
        prim = np.stack([rho, vx, vy, vz, p], axis=1)
3215
3216    prim_im1 = np.roll(prim,  1, axis=2)
        prim_ip1 = np.roll(prim, -1, axis=2)
3217    slope_x  = minmod(prim - prim_im1, prim_ip1 - prim)
3218    prim_L = prim + 0.5 * slope_x
3219    prim_R = np.roll(prim, -1, axis=2) - 0.5 * np.roll(slope_x, -1, axis
3220        =2)
3221    flux_x = _rusanov_flux(prim_L, prim_R, axis=2, gamma=gamma)
        div_x  = (flux_x - np.roll(flux_x, 1, axis=2)) / dx
3222
3223    prim_jm1 = np.roll(prim,  1, axis=3)
3224    prim_jp1 = np.roll(prim, -1, axis=3)
3225    slope_y  = minmod(prim - prim_jm1, prim_jp1 - prim)
        prim_L = prim + 0.5 * slope_y
3226    prim_R = np.roll(prim, -1, axis=3) - 0.5 * np.roll(slope_y, -1, axis
3227        =3)
3228    flux_y = _rusanov_flux(prim_L, prim_R, axis=3, gamma=gamma)
3229    div_y  = (flux_y - np.roll(flux_y, 1, axis=3)) / dx
3230
3231    prim_km1 = np.roll(prim,  1, axis=4)
        prim_kp1 = np.roll(prim, -1, axis=4)
3232    slope_z  = minmod(prim - prim_km1, prim_kp1 - prim)
3233    prim_L = prim + 0.5 * slope_z
3234    prim_R = np.roll(prim, -1, axis=4) - 0.5 * np.roll(slope_z, -1, axis
3235        =4)
3236    flux_z = _rusanov_flux(prim_L, prim_R, axis=4, gamma=gamma)
        div_z  = (flux_z - np.roll(flux_z, 1, axis=4)) / dx
3237
3238    rhs = -(div_x + div_y + div_z)
3239    return rhs.astype(_DTYPE)
```

```
3240   def solver(density0, Vx0, Vy0, Vz0, pressure0,
3241              t_coordinate, eta=1e-8, zeta=1e-8,
3242              gamma=_GAMMA, cfl=_CFL):
3243
3244       assert density0.shape == Vx0.shape == Vy0.shape == Vz0.shape ==
3245           pressure0.shape
3246       B, Nx, Ny, Nz = density0.shape
3247       dx = 1.0 / Nx
3248       T = len(t_coordinate) - 1
3249       out_shape = (B, T + 1, Nx, Ny, Nz)
3250       sol_rho  = np.empty(out_shape, dtype=_DTYPE)
3251       sol_Vx   = np.empty(out_shape, dtype=_DTYPE)
3252       sol_Vy   = np.empty(out_shape, dtype=_DTYPE)
3253       sol_Vz   = np.empty(out_shape, dtype=_DTYPE)
3254       sol_p    = np.empty(out_shape, dtype=_DTYPE)
3255       u = prim_to_cons(density0, Vx0, Vy0, Vz0, pressure0, gamma)
3256       sol_rho[:, 0], sol_Vx[:, 0], sol_Vy[:, 0], sol_Vz[:, 0], sol_p[:, 0]
3257           = \
3258           cons_to_prim(u, gamma)
3259       time_now = float(t_coordinate[0])
3260
3261       for n_out in range(1, T + 1):
3262           t_target = float(t_coordinate[n_out])
3263           sub_steps = 0
3264           while time_now < t_target - 1.0e-14:
3265               rho, vx, vy, vz, p = cons_to_prim(u, gamma)
3266               s_max = max_char_speed(rho, vx, vy, vz, p, gamma)
3267               dt = cfl * dx / s_max
3268               if time_now + dt > t_target:
3269                   dt = t_target - time_now
3270
3271               rhs1 = _compute_rhs(u, dx, gamma)
3272               u1   = u + dt * rhs1
3273               u1[:, 0]  = np.maximum(u1[:, 0], _RHO_FLOOR)
3274               _, _, _, _, p_tmp = cons_to_prim(u1, gamma)
3275               add_E = np.maximum(_P_FLOOR - p_tmp, 0.0) / (gamma - 1.0)
3276               u1[:, 4] += add_E
3277
3278               rhs2 = _compute_rhs(u1, dx, gamma)
3279               u    = 0.5 * (u + u1 + dt * rhs2)
3280
3281               u[:, 0]  = np.maximum(u[:, 0], _RHO_FLOOR)
3282               _, _, _, _, p_tmp = cons_to_prim(u, gamma)
3283               add_E = np.maximum(_P_FLOOR - p_tmp, 0.0) / (gamma - 1.0)
3284               u[:, 4] += add_E
3285
3286               time_now += dt
3287               sub_steps += 1
3288
3289           rho, vx, vy, vz, p = cons_to_prim(u, gamma)
3290           sol_rho[:, n_out]  = rho
3291           sol_Vx[:, n_out]   = vx
3292           sol_Vy[:, n_out]   = vy
3293           sol_Vz[:, n_out]   = vz
           sol_p[:, n_out]    = p

           print(f"[solver] output {n_out:02d}/{T}  (sub-steps {sub_steps:3d
               },  t={t_target:.5f})")

       print("[solver] done    returning dictionary.")
       return {
           'density' : sol_rho,
```

```
3294          'Vx'      : sol_Vx,
3295          'Vy'      : sol_Vy,
3296          'Vz'      : sol_Vz,
3297          'pressure': sol_p
3298      }
```

3299

**PDE-SHARP-generated custom solver (Solver A.2):**

3301

```
3302  import numpy as np
3303  import h5py
3304  from scipy.fft import fftn, ifftn, fftfreq
3305  import time
       from tqdm import tqdm
3306
3307  def solver(density0, Vx0, Vy0, Vz0, pressure0, t_coordinate, eta, zeta):
3308
3309      gamma = 5.0/3.0
          batch_size, Nx, Ny, Nz = density0.shape
3310
3311      Lx, Ly, Lz = 1.0, 1.0, 1.0
3312      dx, dy, dz = Lx/Nx, Ly/Ny, Lz/Nz
3313
          dt = 0.001
3314
          total_time = t_coordinate[-1] - t_coordinate[0]
3315      output_times = t_coordinate
3316
3317      kx = 2j * np.pi * fftfreq(Nx, d=dx).reshape(1, Nx, 1, 1)
3318      ky = 2j * np.pi * fftfreq(Ny, d=dy).reshape(1, 1, Ny, 1)
          kz = 2j * np.pi * fftfreq(Nz, d=dz).reshape(1, 1, 1, Nz)
3319
3320      kx = np.broadcast_to(kx, (batch_size, Nx, Ny, Nz))
3321      ky = np.broadcast_to(ky, (batch_size, Nx, Ny, Nz))
3322      kz = np.broadcast_to(kz, (batch_size, Nx, Ny, Nz))
3323
          k_sq = kx**2 + ky**2 + kz**2
3324
3325      solutions = {
3326          'density': np.zeros((batch_size, len(output_times), Nx, Ny, Nz),
3327              dtype=np.float32),
              'Vx': np.zeros((batch_size, len(output_times), Nx, Ny, Nz), dtype
3328              =np.float32),
3329          'Vy': np.zeros((batch_size, len(output_times), Nx, Ny, Nz), dtype
3330              =np.float32),
              'Vz': np.zeros((batch_size, len(output_times), Nx, Ny, Nz), dtype
3331              =np.float32),
3332          'pressure': np.zeros((batch_size, len(output_times), Nx, Ny, Nz),
3333               dtype=np.float32)
3334      }
3335
3336      solutions['density'][:, 0] = density0
3337      solutions['Vx'][:, 0] = Vx0
3338      solutions['Vy'][:, 0] = Vy0
          solutions['Vz'][:, 0] = Vz0
3339      solutions['pressure'][:, 0] = pressure0
3340
3341      rho = density0.copy()
3342      Vx = Vx0.copy()
          Vy = Vy0.copy()
3343      Vz = Vz0.copy()
3344      p = pressure0.copy()
3345
3346      def compute_derivatives(field):
3347          field_hat = fftn(field, axes=(1, 2, 3))
              dfdx = ifftn(field_hat * kx, axes=(1, 2, 3)).real
```

```
3348        dfdy = ifftn(field_hat * ky, axes=(1, 2, 3)).real
3349        dfdz = ifftn(field_hat * kz, axes=(1, 2, 3)).real
3350        return dfdx, dfdy, dfdz
3351
3352    def compute_laplacian(field):
3353        field_hat = fftn(field, axes=(1, 2, 3))
3354        lap_field = ifftn(field_hat * k_sq, axes=(1, 2, 3)).real
3355        return lap_field
3356
3357    def compute_divergence(fx, fy, fz):
3358        dfx_dx, _, _ = compute_derivatives(fx)
3359        _, dfy_dy, _ = compute_derivatives(fy)
3360        _, _, dfz_dz = compute_derivatives(fz)
3361        return dfx_dx + dfy_dy + dfz_dz
3362
3363    def navier_stokes_rhs(rho, Vx, Vy, Vz, p):
3364        epsilon = p / (gamma - 1.0)
3365
3366        drho_dx, drho_dy, drho_dz = compute_derivatives(rho)
3367        dVx_dx, dVx_dy, dVx_dz = compute_derivatives(Vx)
3368        dVy_dx, dVy_dy, dVy_dz = compute_derivatives(Vy)
3369        dVz_dx, dVz_dy, dVz_dz = compute_derivatives(Vz)
3370        dp_dx, dp_dy, dp_dz = compute_derivatives(p)
3371
3372        lap_Vx = compute_laplacian(Vx)
3373        lap_Vy = compute_laplacian(Vy)
3374        lap_Vz = compute_laplacian(Vz)
3375
3376        div_V = dVx_dx + dVy_dy + dVz_dz
3377        ddivV_dx, ddivV_dy, ddivV_dz = compute_derivatives(div_V)
3378
3379        drho_dt = - (drho_dx * Vx + rho * dVx_dx +
3380                     drho_dy * Vy + rho * dVy_dy +
3381                     drho_dz * Vz + rho * dVz_dz)
3382
3383        V_dot_grad_Vx = Vx * dVx_dx + Vy * dVx_dy + Vz * dVx_dz
3384        V_dot_grad_Vy = Vx * dVy_dx + Vy * dVy_dy + Vz * dVy_dz
3385        V_dot_grad_Vz = Vx * dVz_dx + Vy * dVz_dy + Vz * dVz_dz
3386
3387        viscous_x = eta * lap_Vx + (zeta + eta/3.0) * ddivV_dx
3388        viscous_y = eta * lap_Vy + (zeta + eta/3.0) * ddivV_dy
3389        viscous_z = eta * lap_Vz + (zeta + eta/3.0) * ddivV_dz
3390
3391        dVx_dt = (-V_dot_grad_Vx - dp_dx/rho + viscous_x/rho)
3392        dVy_dt = (-V_dot_grad_Vy - dp_dy/rho + viscous_y/rho)
3393        dVz_dt = (-V_dot_grad_Vz - dp_dz/rho + viscous_z/rho)
3394
3395        v_sq = Vx**2 + Vy**2 + Vz**2
3396        total_energy = epsilon + 0.5 * rho * v_sq
3397
3398        sigma_xx = eta * (2*dVx_dx - 2/3*div_V) + zeta * div_V
3399        sigma_yy = eta * (2*dVy_dy - 2/3*div_V) + zeta * div_V
3400        sigma_zz = eta * (2*dVz_dz - 2/3*div_V) + zeta * div_V
3401        sigma_xy = eta * (dVx_dy + dVy_dx)
            sigma_xz = eta * (dVx_dz + dVz_dx)
            sigma_yz = eta * (dVy_dz + dVz_dy)

            energy_flux_x = (p + total_energy) * Vx - (Vx*sigma_xx + Vy*
                sigma_xy + Vz*sigma_xz)
            energy_flux_y = (p + total_energy) * Vy - (Vx*sigma_xy + Vy*
                sigma_yy + Vz*sigma_yz)
            energy_flux_z = (p + total_energy) * Vz - (Vx*sigma_xz + Vy*
                sigma_yz + Vz*sigma_zz)
```

```
         denergy_flux_dx, denergy_flux_dy, denergy_flux_dz =
             compute_derivatives(
             energy_flux_x), compute_derivatives(energy_flux_y),
                 compute_derivatives(energy_flux_z)

         dtotal_energy_dt = - (denergy_flux_dx[0] + denergy_flux_dy[1] +
             denergy_flux_dz[2])

         dv_sq_dt = 2 * (Vx * dVx_dt + Vy * dVy_dt + Vz * dVz_dt)
         drho_v_sq_dt = 0.5 * (drho_dt * v_sq + rho * dv_sq_dt)
         dp_dt = (gamma - 1.0) * (dtotal_energy_dt - drho_v_sq_dt)

         return drho_dt, dVx_dt, dVy_dt, dVz_dt, dp_dt

    def rk4_step(rho, Vx, Vy, Vz, p, dt):
         k1_rho, k1_Vx, k1_Vy, k1_Vz, k1_p = navier_stokes_rhs(rho, Vx, Vy
             , Vz, p)

         rho2 = rho + 0.5 * dt * k1_rho
         Vx2 = Vx + 0.5 * dt * k1_Vx
         Vy2 = Vy + 0.5 * dt * k1_Vy
         Vz2 = Vz + 0.5 * dt * k1_Vz
         p2 = p + 0.5 * dt * k1_p

         k2_rho, k2_Vx, k2_Vy, k2_Vz, k2_p = navier_stokes_rhs(rho2, Vx2,
             Vy2, Vz2, p2)

         rho3 = rho + 0.5 * dt * k2_rho
         Vx3 = Vx + 0.5 * dt * k2_Vx
         Vy3 = Vy + 0.5 * dt * k2_Vy
         Vz3 = Vz + 0.5 * dt * k2_Vz
         p3 = p + 0.5 * dt * k2_p

         k3_rho, k3_Vx, k3_Vy, k3_Vz, k3_p = navier_stokes_rhs(rho3, Vx3,
             Vy3, Vz3, p3)

         rho4 = rho + dt * k3_rho
         Vx4 = Vx + dt * k3_Vx
         Vy4 = Vy + dt * k3_Vy
         Vz4 = Vz + dt * k3_Vz
         p4 = p + dt * k3_p

         k4_rho, k4_Vx, k4_Vy, k4_Vz, k4_p = navier_stokes_rhs(rho4, Vx4,
             Vy4, Vz4, p4)

         rho_new = rho + (dt/6.0) * (k1_rho + 2*k2_rho + 2*k3_rho + k4_rho
             )
         Vx_new = Vx + (dt/6.0) * (k1_Vx + 2*k2_Vx + 2*k3_Vx + k4_Vx)
         Vy_new = Vy + (dt/6.0) * (k1_Vy + 2*k2_Vy + 2*k3_Vy + k4_Vy)
         Vz_new = Vz + (dt/6.0) * (k1_Vz + 2*k2_Vz + 2*k3_Vz + k4_Vz)
         p_new = p + (dt/6.0) * (k1_p + 2*k2_p + 2*k3_p + k4_p)

         return rho_new, Vx_new, Vy_new, Vz_new, p_new

    current_time = t_coordinate[0]
    output_idx = 1

    pbar = tqdm(total=len(output_times)-1, desc="Time steps completed")

    while output_idx < len(output_times):
         target_time = output_times[output_idx]

         while current_time < target_time:
             step_dt = min(dt, target_time - current_time)
```

```
3456            rho, Vx, Vy, Vz, p = rk4_step(rho, Vx, Vy, Vz, p, step_dt)
3457            current_time += step_dt
3458
3459            rho = np.maximum(rho, 1e-8)
3460            p = np.maximum(p, 1e-8)
3461
3462        solutions['density'][:, output_idx] = rho
3463        solutions['Vx'][:, output_idx] = Vx
3464        solutions['Vy'][:, output_idx] = Vy
3465        solutions['Vz'][:, output_idx] = Vz
3466        solutions['pressure'][:, output_idx] = p
3467
3468        max_rho = np.max(rho)
3469        max_vel = np.max(np.sqrt(Vx**2 + Vy**2 + Vz**2))
3470        max_p = np.max(p)
3471
3472        pbar.set_postfix({
3473            'time': f'{current_time:.3f}',
3474            'max_ ': f'{max_rho:.3f}',
3475            'max_vel': f'{max_vel:.3f}',
3476            'max_p': f'{max_p:.3f}'
3477        })
3478        pbar.update(1)
3479
3480        output_idx += 1
3481
3482    pbar.close()
3483
3484    return solutions
```

**PDE-SHARP-generated library solver (Solver B):**

```
import numpy as np
import h5py
import time
from dedalus import public as de
from dedalus.core.operators import integrate
import logging
from mpi4py import MPI

logger = logging.getLogger(__name__)

def solver(density0, Vx0, Vy0, Vz0, pressure0, t_coordinate, eta, zeta):

    gamma = 5.0/3.0
    batch_size, Nx, Ny, Nz = density0.shape

    Lx, Ly, Lz = 1.0, 1.0, 1.0

    x_basis = de.Fourier('x', Nx, interval=(0, Lx), dealias=3/2)
    y_basis = de.Fourier('y', Ny, interval=(0, Ly), dealias=3/2)
    z_basis = de.Fourier('z', Nz, interval=(0, Lz), dealias=3/2)
    domain = de.Domain([x_basis, y_basis, z_basis], grid_dtype=np.float64
        )

    solutions = {
        'density': np.zeros((batch_size, len(t_coordinate), Nx, Ny, Nz),
            dtype=np.float32),
        'Vx': np.zeros((batch_size, len(t_coordinate), Nx, Ny, Nz), dtype
            =np.float32),
        'Vy': np.zeros((batch_size, len(t_coordinate), Nx, Ny, Nz), dtype
            =np.float32),
        'Vz': np.zeros((batch_size, len(t_coordinate), Nx, Ny, Nz), dtype
            =np.float32),
```

```
3510            'pressure': np.zeros((batch_size, len(t_coordinate), Nx, Ny, Nz),
3511                dtype=np.float32)
3512        }
3513
3514        solutions['density'][:, 0] = density0
3515        solutions['Vx'][:, 0] = Vx0
3516        solutions['Vy'][:, 0] = Vy0
3517        solutions['Vz'][:, 0] = Vz0
3518        solutions['pressure'][:, 0] = pressure0
3519        for batch_idx in range(batch_size):
3520            batch_solutions = solve_single_batch(
3521                density0[batch_idx], Vx0[batch_idx], Vy0[batch_idx], Vz0[
3522                    batch_idx],
3523                pressure0[batch_idx], t_coordinate, eta, zeta, gamma, domain
3524            )
3525            for key in solutions:
3526                solutions[key][batch_idx] = batch_solutions[key]
3527        return solutions
3528
3529    def solve_single_batch(density0, Vx0, Vy0, Vz0, pressure0, t_coordinate,
3530        eta, zeta, gamma, domain):
3531        problem = de.IVP(domain, variables=['rho', 'vx', 'vy', 'vz', 'p'])
3532
3533        problem.parameters['eta'] = eta
3534        problem.parameters['zeta'] = zeta
3535        problem.parameters['gamma'] = gamma
3536        problem.parameters['eta_bulk'] = zeta + eta/3.0
3537        problem.add_equation("dt(rho) + dx(rho*vx) + dy(rho*vy) + dz(rho*vz)
3538            = 0")
3539
3540        problem.add_equation("dt(vx) + (vx*dx(vx) + vy*dy(vx) + vz*dz(vx)) +
3541            dx(p)/rho = " +
3542                            "(eta/rho)*lap(vx) + (eta_bulk/rho)*dx(dx(vx) +
3543                                dy(vy) + dz(vz))")
3544        problem.add_equation("dt(vy) + (vx*dx(vy) + vy*dy(vy) + vz*dz(vy)) +
3545            dy(p)/rho = " +
3546                            "(eta/rho)*lap(vy) + (eta_bulk/rho)*dy(dx(vx) +
3547                                dy(vy) + dz(vz))")
3548        problem.add_equation("dt(vz) + (vx*dx(vz) + vy*dy(vz) + vz*dz(vz)) +
3549            dz(p)/rho = " +
3550                            "(eta/rho)*lap(vz) + (eta_bulk/rho)*dz(dx(vx) +
3551                                dy(vy) + dz(vz))")
3552        problem.add_equation("dt(p) + vx*dx(p) + vy*dy(p) + vz*dz(p) + gamma*
3553            p*(dx(vx) + dy(vy) + dz(vz)) = " +
3554                            "(gamma-1)*(eta*2*(dx(vx)**2 + dy(vy)**2 + dz(vz)
3555                                **2 + " +
3556                            "0.5*(dy(vx)+dx(vy))**2 + 0.5*(dz(vx)+dx(vz))**2
3557                                + 0.5*(dz(vy)+dy(vz))**2 - " +
3558                            "(2/3)*(dx(vx)+dy(vy)+dz(vz))**2) + " +
3559                            "zeta*(dx(vx)+dy(vy)+dz(vz))**2)")
3560        ts = de.timesteppers.RK443
3561        solver = problem.build_solver(ts)
3562        solver.state['rho']['g'] = density0
3563        solver.state['vx']['g'] = Vx0
        solver.state['vy']['g'] = Vy0
```

```
3564    solver.state['vz']['g'] = Vz0
3565    solver.state['p']['g'] = pressure0
3566
3567    batch_solutions = {
3568        'density': np.zeros((len(t_coordinate),) + domain.
                local_grid_shape(0)),
3569        'Vx': np.zeros((len(t_coordinate),) + domain.local_grid_shape(0))
3570            ,
3571        'Vy': np.zeros((len(t_coordinate),) + domain.local_grid_shape(0))
3572            ,
3573        'Vz': np.zeros((len(t_coordinate),) + domain.local_grid_shape(0))
3574            ,
3575        'pressure': np.zeros((len(t_coordinate),) + domain.
                local_grid_shape(0))
3576    }
3577
3578    batch_solutions['density'][0] = solver.state['rho']['g']
3579    batch_solutions['Vx'][0] = solver.state['vx']['g']
3580    batch_solutions['Vy'][0] = solver.state['vy']['g']
3581    batch_solutions['Vz'][0] = solver.state['vz']['g']
3582    batch_solutions['pressure'][0] = solver.state['p']['g']
3583
3584    dt = 0.0005
3585    solver.stop_sim_time = t_coordinate[-1]
3586    solver.stop_wall_time = np.inf
3587    solver.stop_iteration = np.inf
3588
3589    output_idx = 1
3590    current_time = t_coordinate[0]
3591
3592    start_time = time.time()
3593
3594    while solver.ok and output_idx < len(t_coordinate):
3595        target_time = t_coordinate[output_idx]
3596
3597        while current_time < target_time:
3598            step_dt = min(dt, target_time - current_time)
3599            solver.step(step_dt)
3600            current_time = solver.sim_time
3601
3602            solver.state['rho']['g'] = np.maximum(solver.state['rho']['g'
                ], 1e-8)
3603            solver.state['p']['g'] = np.maximum(solver.state['p']['g'], 1
                e-8)
3604
3605        batch_solutions['density'][output_idx] = solver.state['rho']['g']
3606        batch_solutions['Vx'][output_idx] = solver.state['vx']['g']
3607        batch_solutions['Vy'][output_idx] = solver.state['vy']['g']
3608        batch_solutions['Vz'][output_idx] = solver.state['vz']['g']
3609        batch_solutions['pressure'][output_idx] = solver.state['p']['g']
3610
3611        max_rho = np.max(solver.state['rho']['g'])
3612        max_vel = np.max(np.sqrt(solver.state['vx']['g']**2 +
                                   solver.state['vy']['g']**2 +
                                   solver.state['vz']['g']**2))
3613        max_p = np.max(solver.state['p']['g'])
3614
3615        print(f"  t={current_time:.3f}, max_ ={max_rho:.3f}, max_vel={
                max_vel:.3f}, max_p={max_p:.3f}")
3616
3617        output_idx += 1

    end_time = time.time()

    return batch_solutions
```

**Claude Code-generated library solver (Solver C):**

```python
import numpy as np
import h5py
from mpi4py import MPI
from dolfinx import mesh, fem, io
from dolfinx.fem import FunctionSpace, Function, Constant, Expression
from dolfinx.fem.petsc import LinearProblem, NonlinearProblem
from dolfinx.nls.petsc import NewtonSolver
import ufl
from petsc4py import PETSc
import time

def solver(density0, Vx0, Vy0, Vz0, pressure0, t_coordinate, eta, zeta):

    Gamma = 5.0/3.0
    batch_size = density0.shape[0]
    Nx, Ny, Nz = density0.shape[1], density0.shape[2], density0.shape[3]
    T = len(t_coordinate) - 1

    solutions = {
        'density': np.zeros((batch_size, T+1, Nx, Ny, Nz), dtype=np.
            float32),
        'Vx': np.zeros((batch_size, T+1, Nx, Ny, Nz), dtype=np.float32),
        'Vy': np.zeros((batch_size, T+1, Nx, Ny, Nz), dtype=np.float32),
        'Vz': np.zeros((batch_size, T+1, Nx, Ny, Nz), dtype=np.float32),
        'pressure': np.zeros((batch_size, T+1, Nx, Ny, Nz), dtype=np.
            float32)
    }

    solutions['density'][:, 0] = density0
    solutions['Vx'][:, 0] = Vx0
    solutions['Vy'][:, 0] = Vy0
    solutions['Vz'][:, 0] = Vz0
    solutions['pressure'][:, 0] = pressure0

    for batch_idx in range(batch_size):

        batch_solutions = solve_single_batch_fenicsx(
            density0[batch_idx], Vx0[batch_idx], Vy0[batch_idx], Vz0[
                batch_idx],
            pressure0[batch_idx], t_coordinate, eta, zeta, Gamma, Nx, Ny,
                Nz
        )

        for key in solutions:
            solutions[key][batch_idx] = batch_solutions[key]

    return solutions

def solve_single_batch_fenicsx(density0, Vx0, Vy0, Vz0, pressure0,
    t_coordinate, eta, zeta, Gamma, Nx, Ny, Nz):

    comm = MPI.COMM_WORLD

    domain = mesh.create_box(comm, [[0, 0, 0], [1, 1, 1]], [Nx-1, Ny-1,
        Nz-1],
                            cell_type=mesh.CellType.hexahedron)

    P1 = ufl.FiniteElement("Lagrange", domain.ufl_cell(), 1)
    V_element = ufl.VectorElement("Lagrange", domain.ufl_cell(), 1, dim
        =3)
    me_element = ufl.MixedElement([P1, V_element, P1])
```

```
3672
3673        W = FunctionSpace(domain, me_element)
3674        w = Function(W)
3675        w_n = Function(W)
3676
3677        rho, u, p = ufl.split(w)
3678        rho_n, u_n, p_n = ufl.split(w_n)
3679
3680        v_rho, v_u, v_p = ufl.TestFunctions(W)
3681        dt = Constant(domain, t_coordinate[1] - t_coordinate[0])
3682        t = 0.0
3683
3684        eta_const = Constant(domain, float(eta))
3685        zeta_const = Constant(domain, float(zeta))
            Gamma_const = Constant(domain, Gamma)
3686
3687        epsilon = p / (Gamma_const - 1)
3688        epsilon_n = p_n / (Gamma_const - 1)
3689
3690        def stress_tensor(u_field, p_field):
3691            I = ufl.Identity(3)
                strain_rate = 0.5 * (ufl.grad(u_field) + ufl.grad(u_field).T)
3692            div_u = ufl.div(u_field)
3693            return 2 * eta_const * strain_rate + (zeta_const - 2*eta_const/3)
                    * div_u * I - p_field * I
3694
3695        F_adv = (
3696            v_rho * (rho - rho_n) * ufl.dx +
                dt * v_rho * ufl.dot(u_n, ufl.grad(rho_n)) * ufl.dx +
3697            ufl.inner(v_u, (u - u_n)) * ufl.dx +
3698            dt * ufl.inner(v_u, ufl.dot(ufl.grad(u_n), u_n)) * ufl.dx +
3699            v_p * (p - p_n) * ufl.dx +
3700            dt * v_p * ufl.dot(u_n, ufl.grad(p_n)) * ufl.dx
3701        )
3702
3703        F_corr = (
3704            v_rho * rho * ufl.div(u) * ufl.dx +
3705            ufl.inner(v_u, rho * u) * ufl.dx +
                dt * ufl.inner(ufl.grad(v_u), stress_tensor(u, p)) * ufl.dx +
3706            v_p * (epsilon + 0.5 * rho * ufl.dot(u, u)) * ufl.dx +
3707            dt * v_p * ufl.div((epsilon + p + 0.5 * rho * ufl.dot(u, u)) * u)
3708                * ufl.dx
            )
3709
3710        F = (
3711            v_rho * (rho - rho_n) * ufl.dx +
3712            dt * v_rho * ufl.div(rho_n * u_n) * ufl.dx +
                ufl.inner(v_u, rho * (u - u_n)) * ufl.dx +
3713            dt * ufl.inner(v_u, rho_n * ufl.dot(ufl.grad(u_n), u_n)) * ufl.dx
3714                +
3715            dt * ufl.inner(v_u, ufl.grad(p)) * ufl.dx +
3716            dt * ufl.inner(ufl.grad(v_u), stress_tensor(u, p_n)) * ufl.dx +
3717            v_p * (p - p_n) * ufl.dx +
                dt * v_p * Gamma_const * p_n * ufl.div(u_n) * ufl.dx
3718        )
3719
3720        def initial_condition(x):
3721            xi = np.clip((x[0] * (Nx-1)).astype(int), 0, Nx-1)
3722            yi = np.clip((x[1] * (Ny-1)).astype(int), 0, Ny-1)
                zi = np.clip((x[2] * (Nz-1)).astype(int), 0, Nz-1)
3723
3724            rho_vals = density0[xi, yi, zi]
3725            ux_vals = Vx0[xi, yi, zi]
                uy_vals = Vy0[xi, yi, zi]
```

```
           uz_vals = Vz0[xi, yi, zi]
           p_vals = pressure0[xi, yi, zi]

           return np.array([rho_vals, ux_vals, uy_vals, uz_vals, p_vals])

       w.interpolate(initial_condition)
       w_n.interpolate(initial_condition)

       batch_solutions = {
           'density': np.zeros((len(t_coordinate), Nx, Ny, Nz), dtype=np.
               float32),
           'Vx': np.zeros((len(t_coordinate), Nx, Ny, Nz), dtype=np.float32)
               ,
           'Vy': np.zeros((len(t_coordinate), Nx, Ny, Nz), dtype=np.float32)
               ,
           'Vz': np.zeros((len(t_coordinate), Nx, Ny, Nz), dtype=np.float32)
               ,
           'pressure': np.zeros((len(t_coordinate), Nx, Ny, Nz), dtype=np.
               float32)
       }

       store_solution(w, batch_solutions, 0, Nx, Ny, Nz)

       problem = NonlinearProblem(F, w)
       solver = NewtonSolver(comm, problem)
       solver.rtol = 1e-6
       solver.atol = 1e-8
       solver.max_it = 10
       solver.report = True

       start_time = time.time()

       for step in range(1, len(t_coordinate)):

           t += float(dt)

           try:
               n_iter, converged = solver.solve(w)
               if converged:
                   print(f"  Newton converged in {n_iter} iterations")
               else:
                   print(f"  Newton failed to converge in {n_iter}
                       iterations")

               store_solution(w, batch_solutions, step, Nx, Ny, Nz)

               w_n.x.array[:] = w.x.array[:]

               rho, u, p = w.split()
               max_rho = np.max(rho.vector.array)
               max_vel = np.max(np.linalg.norm(u.vector.array.reshape(-1, 3)
                   , axis=1))
               max_p = np.max(p.vector.array)
               print(f"  max_ ={max_rho:.3f}, max_vel={max_vel:.3f}, max_p
                   ={max_p:.3f}")

           except Exception as e:
               print(f"  Solver error: {e}")
               w.x.array[:] = w_n.x.array[:]
               store_solution(w, batch_solutions, step, Nx, Ny, Nz)

       end_time = time.time()

       return batch_solutions
```

**Solver A.1: Custom Finite-Volume Method with MUSCL Reconstruction**

*Approach:* This solver implements a conservative finite-volume scheme with:

- **Spatial discretization:** Second-order MUSCL (Monotonic Upstream-centered Scheme for Conservation Laws) with minmod limiter

- **Flux computation:** Rusanov (local Lax-Friedrichs) approximate Riemann solver

- **Time integration:** Second-order Runge-Kutta (Heun's method) with adaptive CFL-controlled timestep

- **Equation form:** Solves the conservative form of Euler equations (omits viscous terms despite parameters $\eta$, $\zeta$ being passed)

*Theoretical Properties:*

- **Formal spatial accuracy:** 2nd order in smooth regions (1st order near discontinuities due to limiter)

- **Temporal accuracy:** 2nd order (Heun's method)

- **Stability:** CFL condition ensures stability, typically CFL $< 0.5$ for explicit schemes

- **Conservation:** Fully conservative for mass, momentum, and energy

- **Limitation:** Missing viscous stress terms in implementation despite being passed $\eta$, $\zeta$ parameters

*Computational Characteristics:*

- **Memory:** Moderate (stores primitive and conservative variables)

- **Parallelism:** Implicitly parallel via NumPy operations

- **Computational cost:** $\mathcal{O}(N_x \times N_y \times N_z)$ per timestep

**Solver A.2: Custom Pseudo-Spectral Method**

*Approach:* This solver implements a pseudo-spectral method with:

- **Spatial discretization:** Fourier spectral method using FFT for derivatives

- **Time integration:** 4th-order Runge-Kutta (RK4) with fixed timestep

- **Equation form:** Solves the full Navier-Stokes equations in primitive form

- **Viscosity treatment:** Includes complete viscous stress tensor computation

*Theoretical Properties:*

- **Spatial accuracy:** Spectral/exponential convergence for smooth solutions

- **Temporal accuracy:** 4th order (RK4)

- **Aliasing:** No explicit dealiasing (potential for aliasing errors)

- **Boundary conditions:** Implicitly periodic (from Fourier basis)

- **Advantage:** High accuracy for smooth flows with periodic boundaries

*Computational Characteristics:*

- **Memory:** High (requires $4\times$ storage for RK4 stages)

- **FFT operations:** $\mathcal{O}(N \log N)$ per derivative computation

- **Parallelism:** Single-processor FFT implementation

- **Stability:** Explicit timestep limited by CFL and viscous stability conditions

**Solver B: Dedalus Spectral Framework**

*Approach:* This solver uses the Dedalus spectral solver library:

- **Spatial discretization:** Fourier spectral method with 3/2 dealiasing
- **Time integration:** 4th-order Runge-Kutta (RK443) timestepper
- **Equation form:** Solves full compressible Navier-Stokes in symbolic form
- **Library features:** Automatic equation parsing, parallelization via MPI

*Theoretical Properties:*

- **Spatial accuracy:** Spectral/exponential convergence
- **Dealias:** 3/2 rule prevents aliasing errors
- **Temporal accuracy:** 4th order (RK443 scheme)
- **Advantage:** Robust spectral method with proper dealiasing and parallelization

*Dedalus Library Characteristics:*

- Domain decomposition for parallel computation
- Symbolic equation specification
- Automatic differentiation and spectral transforms
- Support for various boundary conditions (periodic in this case)

**Solver C: FEniCSx Finite Element Method**

*Approach:* This solver uses the FEniCSx finite element framework:

- **Spatial discretization:** Continuous Galerkin finite elements (P1 Lagrange)
- **Time integration:** Implicit backward Euler with Newton solver
- **Equation form:** Weak formulation of compressible Navier-Stokes
- **Library features:** Automated finite element assembly, parallel computation

*Theoretical Properties:*

- **Spatial accuracy:** 1st order (P1 elements)
- **Temporal accuracy:** 1st order (backward Euler)
- **Stability:** Unconditionally stable (implicit method)
- **Advantage:** Handles complex geometries well, though not needed for periodic box

*FEniCSx Library Characteristics:*

- Automated finite element assembly from variational forms
- Parallel computation via domain decomposition
- Support for complex geometries and boundary conditions
- Implicit solvers suitable for stiff problems

**Convergence Comparison**

| Solver | Spatial Order | Temporal Order | Method Type | Viscous Terms | Stability |
|--------|---------------|-----------------|-------------|---------------|-----------|
| A.1 | 2nd (MUSCL) | 2nd (Heun) | Finite-Volume | Missing | CFL-limited |
| A.2 | Spectral | 4th (RK4) | Pseudo-Spectral | Complete | CFL + viscous limited |
| B | Spectral | 4th (RK443) | Spectral (Dedalus) | Complete | CFL + viscous limited |
| C | 1st (P1) | 1st (Backward Euler) | Finite Element | Complete | Unconditionally stable |

**Computational Efficiency Analysis**

*Memory Requirements:*

- **Solver A.1:** Lowest memory footprint ($\sim$5 fields $\times$ 5 variables)
- **Solver A.2:** High memory (RK4 stages + spectral transforms)
- **Solver B:** Moderate (Dedalus manages memory efficiently with dealiasing)
- **Solver C:** Highest (finite element matrices, Newton solver workspace)

*Parallel Scalability:*

- **Solver B:** Excellent (MPI parallelization in Dedalus)
- **Solver C:** Good (FEniCSx with MPI)
- **Solver A.1/A.2:** Limited to NumPy vectorization (single node)

**Accuracy and Suitability Assessment**

*For the Given Problem (Periodic $128^3$ Box):*

**Most Suitable:** Solver B (Dedalus)

**Reasons:**

- Spectral accuracy matches smooth flow expectations
- Proper dealiasing prevents numerical instability
- Parallel implementation scales to 3D
- Complete viscous term implementation

**Second Choice:** Solver A.2 (Custom Spectral)

- Good accuracy but lacks dealiasing and parallelization
- Complete physics implementation

**Problematic Choices:**

- **Solver A.1:** Missing viscous terms despite parameters (invalid for $\eta, \zeta \neq 0$)
- **Solver C:** Overkill for periodic box, low-order accuracy relative to spectral methods

**Expected Error Metrics (Theoretical)**

*Spatial Discretization Error:*

- **A.1:** $\mathcal{O}(\Delta x^2)$ in smooth regions
- **A.2:** Exponential convergence (spectral)
- **B:** Exponential convergence with dealiasing
- **C:** $\mathcal{O}(\Delta x)$

*Temporal Discretization Error:*

- **A.1:** $\mathcal{O}(\Delta t^2)$
- **A.2:** $\mathcal{O}(\Delta t^4)$
- **B:** $\mathcal{O}(\Delta t^4)$
- **C:** $\mathcal{O}(\Delta t)$

*Conservation Properties:*

- **A.1:** Conservative (finite-volume)
- **A.2:** Conservative (spectral with proper treatment)
- **B:** Conservative (Dedalus ensures conservation)
- **C:** Conservative (finite element weak form)

# E EXAMPLE OF A FULL PDE-SHARP REPORT: REACTION-DIFFUSION PDE SOLVER GENERATION AND REFINEMENT

## E.1 ANALYSIS STAGE

In this section, the Reporter agent (Section 3) provides a summary of the Analysis stage of PDE-SHARP for the reaction-diffusion PDE. This report follows the pipeline presented in Figure 2. Each of the following sections contains the LLM response to corresponding Analysis stage prompt (full prompts appear in Appendix F). As seen here, upon identifying the best strategy for this PDE, PDE-SHARP moves on to stability analysis, skipping the less optimal purely numerical approaches. This results in higher quality solver generation in the Genesis stage (see Appendix E.2).

### E.1.1 PDE CLASSIFICATION

```
{
"order": 2,
"linearity": "non-linear",
"type": "parabolic",
"homogeneity": "homogeneous",
"domain_bc": "The spatial domain is the interval (0,1) with periodic
    boundary conditions, meaning u(t,0) = u(t,1) for all t. The temporal
    domain is (0,T] with initial condition u(0,x) = u_0(x).",
"special_properties": "This is a reaction-diffusion equation with a
    logistic source term, often called the Fisher-KPP equation when    >
    0. It exhibits traveling wave solutions and has a non-linear reaction
     term that prevents separability. The equation is invariant under
    spatial translations due to periodic boundary conditions.",
"char_polynomial": "For the principal part of the PDE, considering the
    highest-order derivatives, the characteristic polynomial is derived
    from –     _ {xx} u. In the spatial domain, this gives –         = 0,
    indicating repeated real characteristics, which confirms the
    parabolic type."
}
```

### E.1.2 ANALYTIC SOLUTION CHECK

NO. The Fisher-KPP equation, given by

$$\partial_t u - \nu \partial_{xx} u - \rho u(1-u) = 0$$

with periodic boundary conditions and general initial data $u_0(x)$, does not admit a closed-form analytical solution. This is due to several fundamental obstacles:

**1. Nonlinearity:** The reaction term $\rho u(1-u)$ is nonlinear, which prevents the application of linear superposition principles and standard methods like separation of variables or Fourier series expansion directly. In linear PDEs, these methods allow construction of general solutions, but here the nonlinearity couples modes, leading to an infinite system of nonlinear ordinary differential equations that cannot be solved in closed form.

**2. Lack of Integrability:** The Fisher-KPP equation is not integrable via known transformations (e.g., Cole–Hopf for Burgers' equation) or other techniques that linearize the PDE. While traveling wave solutions exist for specific wave speeds on infinite domains, they do not satisfy periodic boundary conditions or arbitrary initial data, and they are only particular solutions, not general ones.

**3. Complexity with Boundary Conditions:** Periodic boundary conditions imply that solutions can be expressed in terms of Fourier series, but the nonlinear reaction term introduces mode coupling, making the Fourier coefficients satisfy nonlinear ODEs with no analytical solution. Numerical methods are typically required to approximate solutions.

**4. No General Solution Formula:** Despite extensive research, no closed-form expression exists for $u(t, x)$ that holds for all initial conditions $u_0(x)$, parameters $\nu$ and $\rho$, and time $t$ under periodic boundaries. Analytical results are limited to special cases (e.g., steady states, similarity solutions, or perturbative analyses), but these cannot be tailored to the general case here.

Thus, for this PDE with the specified conditions, numerical methods (e.g., finite difference, spectral methods) are necessary, as implemented in the solver task.

### E.1.3 TRANSFORMATION CHECK

NO. The Fisher-KPP equation,

$$\partial_t u - \nu \partial_{xx} u - \rho u (1 - u) = 0$$

with periodic boundary conditions and general initial data $u_0(x)$, **cannot** be transformed into a simpler form that yields known analytical solutions for all cases. This is due to several key reasons:

**1. Nonlinearity Obstacle:** The reaction term $\rho u(1 - u)$ is nonlinear and non-integrable, meaning it cannot be linearized through standard transformations like Cole–Hopf (which works for Burgers' equation but not here). Attempts to apply function transformations (e.g., $u = f(v)$) result in additional nonlinear terms, such as $(v_x)^2$ and $v_{xx}$, which cannot be eliminated simultaneously with the reaction term.

**2. Traveling Wave Limitations:** While traveling wave solutions exist for infinite domains using transformations like $u(x, t) = U(z)$ where $z = x - ct$, these only yield particular solutions for specific wave speeds $c$ and do not satisfy periodic boundary conditions or arbitrary initial data. This approach reduces the PDE to an ODE, but the solutions are not general and cannot be extended to the periodic domain.

**3. Transform Ineffectiveness:** Common integral transforms (e.g., Fourier or Laplace) fail because the nonlinear term couples modes, leading to convolutions or nonlinear integral equations that are intractable. Coordinate transformations (e.g., similarity variables) do not apply, as the equation lacks scaling symmetries that could reduce it to a canonical form like the heat equation.

**4. Periodic Boundary Constraints:** The periodic boundary conditions complicate any transformation attempt, as they require the solution to be periodic in $x$, which is not preserved under most nonlinear transformations. Even if a transformation were found for infinite domains, it would not generalize to this bounded domain.

In summary, no transformation exists that can simplify this PDE to a linear or exactly solvable form while respecting its given conditions. Therefore, numerical methods are necessary for solving this problem in practice.

### E.1.4 HYBRID/DECOMPOSITION CHECK

YES. Operator splitting is highly viable for this diffusion-reaction PDE due to the distinct nature of the operators involved, each amenable to robust numerical treatment. The PDE $\partial_t u = \nu \partial_{xx} u - \rho u (1 - u) = 0$ can be split into diffusion and reaction operators, allowing for efficient and stable solutions when combined with appropriate numerical methods. Below is a detailed analysis based on the required steps:

**STEP 1: Operator Identification**

- **Diffusion operator:** $L_{\text{diff}} = \nu \partial_{xx} u$. This is a linear parabolic operator. For periodic boundary conditions, finite difference methods are robust. Explicit finite differences are simple but require strict CFL conditions ($\Delta t \leq \frac{\Delta x^2}{2\nu}$ for stability). Implicit schemes (e.g., Crank–Nicolson) are unconditionally stable and efficient for 1D problems with periodic BCs, as the resulting linear system is cyclic tridiagonal and can be solved directly with $\mathcal{O}(N)$ complexity.

- **Reaction operator:** $L_{\text{react}} = \rho u(1 - u)$. This is a nonlinear but autonomous operator. Crucially, the reaction equation $\partial_t u = \rho u(1 - u)$ has an exact analytical solution for any initial condition $u(0)$:

$$u(t) = \frac{u(0)}{u(0) + (1 - u(0))e^{-\rho t}}.$$

  This solution is stable, preserves bounds (e.g., $u \in [0, 1]$ if initially so), and avoids numerical stiffness issues.

**Operator splitting approach:** Use Strang splitting (second-order accurate) for each time step $\Delta t$:

1. Reaction half-step: Solve $\partial_t u = \rho u(1 - u)$ for $\Delta t/2$ using the exact solution.

2. Diffusion full-step: Solve $\partial_t u = \nu \partial_{xx} u$ for $\Delta t$ using finite differences with periodic BCs.

3. Reaction half-step: Repeat step 1 for another $\Delta t/2$.

This splitting decouples the nonlinear reaction from the linear diffusion, leveraging the exact solution for reaction and robust numerical methods for diffusion.

**STEP 2: Robustness Analysis and Efficiency**   **Diffusion treatment:**

- **Most reliable method:** Implicit Crank–Nicolson scheme for diffusion. It is unconditionally stable, second-order accurate in time and space, and handles periodic BCs efficiently via Thomas algorithm for cyclic systems. This avoids CFL constraints, allowing larger internal time steps if needed for stability.

- **Stability constraints:** No stability limits from diffusion when implicit, but accuracy may require $\Delta t$ proportional to $\Delta x^2$ for temporal resolution. Explicit diffusion is feasible with small $\Delta t$ but less robust for stiff cases ($\nu$ large).

- **Safeguards:** Ensure spatial discretization $\Delta x$ resolves initial conditions; monitor conservation of mass (if applicable) via discrete checks.

**Reaction treatment:**

- **Most reliable method:** Exact analytical solution. It is efficient, unconditionally stable, and preserves positivity and bounds without numerical errors.

- **Stability constraints:** None, as the solution is exact.

- **Safeguards:** Handle cases where $u(0)$ is outside $[0, 1]$ by clipping or projecting, though the logistic form naturally bounds solutions.

**Efficiency:** Strang splitting with exact reaction and implicit diffusion is computationally efficient—reaction steps are $\mathcal{O}(N)$ per evaluation, and diffusion steps involve linear solves with $\mathcal{O}(N)$ cost. This approach is more stable than monolithic schemes (e.g., fully implicit) for nonlinearities and avoids overengineering.

**STEP 3: Method Precedence for Stability**   **Stability hierarchy:**

1. **Apply reaction first:** The exact solution preserves physical constraints (positivity, bounds) inherently, making it ideal for initial and final substeps. This prevents unphysical values from affecting diffusion.

2. **Diffusion second:** Implicit diffusion ensures stability regardless of $\Delta t$, but if explicit diffusion is used, it should follow reaction steps only after ensuring $\Delta t$ meets CFL conditions. However, implicit diffusion is recommended for robustness.

**Stiffness consideration:**

- Reaction time scale: $\sim 1/\rho$ (if $\rho$ large, exact solution handles stiffness).

- Diffusion time scale: $\sim \Delta x^2/\nu$ (if $\nu$ large, implicit diffusion handles stiffness).

- Since reaction has no time step restriction, diffusion dictates the internal time step choice when using implicit methods; for explicit diffusion, the CFL condition dominates.

**Splitting principle:** Strang splitting with reaction–diffusion–reaction order ensures second-order accuracy and minimizes splitting errors. It preserves solution properties better than Lie splitting (first-order).

**Conclusion** Operator splitting is viable and recommended for this PDE. It leverages the exact solution for reaction and robust implicit methods for diffusion, ensuring stability, accuracy, and efficiency. This approach aligns with the task's requirement to use smaller internal time steps for stability, as the splitting allows adaptive control of $\Delta t$ for diffusion without affecting reaction accuracy.

Implementation should use Strang splitting with Crank–Nicolson for diffusion and exact reaction steps, providing a reliable solver for the given batch-based discretization.

### E.1.5 NUMERICAL STABILITY ANALYSIS

Based on the stability analysis for the Fisher-KPP equation with periodic boundary conditions, the PDE is parabolic and non-linear, consisting of diffusion and reaction operators. The analysis focuses on numerical robustness using explicit methods with conservative safety factors. Below is the step-by-step stability analysis:

**STEP 1: Individual Operator Analysis**

- **Diffusion operator** ($\nu \partial_{xx} u$): For explicit finite differences, the theoretical stability limit is
$$\Delta t \leq \frac{\Delta x^2}{2\nu}.$$
  With a safety factor, this becomes
$$\Delta t \leq 0.25 \cdot \frac{\Delta x^2}{2\nu}.$$

- **Reaction operator** ($\rho u(1-u)$): The linearized stability limit for explicit time stepping is
$$\Delta t \leq \frac{2}{|\rho|}.$$
  With a safety factor, this becomes
$$\Delta t \leq 0.25 \cdot \frac{2}{|\rho|}.$$

**STEP 2: Operator Interaction Analysis** The combined stability limit is the minimum of the individual limits. Diffusion typically dominates for small $\Delta x$ or large $\nu$, while reaction dominates for large $|\rho|$. Explicit methods are preferred for simplicity and robustness, avoiding overengineered schemes.

**STEP 3: Splitting-Aware Stability** Operator splitting (Strang splitting) is used: reaction $\to$ diffusion $\to$ reaction. This order helps preserve solution bounds (e.g., positivity) since the exact reaction solution maintains bounds, and diffusion with stable $\Delta t$ controls errors.

**STEP 4: Time Stepping Strategy**

- **Internal time stepping**: Use
$$\Delta t_{\max} = 0.25 \cdot \min\left(\frac{\Delta x^2}{2\nu}, \frac{2}{|\rho|}\right)$$
  for internal steps.

- **Exact endpoint targeting**: For each output interval $[t_i, t_{i+1}]$, adapt the last step to hit $t_{i+1}$ exactly by using

$$\Delta t = \min(\Delta t_{\max}, t_{i+1} - \text{current}_t).$$

- **Scheme**: Strang splitting with exact reaction steps and explicit diffusion steps ensures stability and accuracy.

Listing 1: A summarized numerical stability analysis result

```
{
  "dt_crit_formula": "min( dx**2 / (2 *    ), 2 / abs(  ) )",
  "safety_factor_suggestion": 0.25,
  "dt_max_formula": "0.25 * min( dx**2 / (2 *    ), 2 / abs(  ) )",
  "n_internal_formula": "ceil(  t_output  / dt_max )",
  "dx_formula": "dx = 1 / N",
  "operator_stability_limits": ["Diffusion: dt     0.25 * (dx**2 / (2 *
       ))", "Reaction: dt     0.25 * (2 / abs(  ))"],
  "operator_precedence_order": "reaction -> diffusion -> reaction",
  "constraint_preservation": ["positivity and bounds [0,1] via exact
       reaction solution", "diffusion with stable dt controls errors"],
  "exact_endpoint_targeting": true,
  "must_conserve": ["none"],
  "suggested_scheme": "Strang splitting: reaction (exact) -> diffusion (
       explicit finite difference) -> reaction (exact)",
  "time_stepping_strategy": "Adaptive internal time stepping with dt_max
       and exact endpoint targeting",
  "stiffness": "Can be stiff if    << 1 or    >> 1, but handled by stable
       dt",
  "gpu_backend": "Suitable for GPU acceleration due to batch operations
       and element-wise reactions"
}
```

## E.2  GENESIS STAGE

The Genesis stage converts mathematical insights from Analysis into executable solver code using the Analysis stage result (Listing 1 JSON) within the relevant Genesis stage prompt (Appendix F). For reaction-diffusion, PDE-SHARP's Analysis stage identified two critical insights that fundamentally differentiate its approach from baseline methods:

1. **Hybrid Strategy Discovery:** The reaction term admits an exact analytical solution, making operator splitting with analytical reaction integration the optimal approach.

2. **Stability-Aware Implementation:** Symbolic stability analysis provided explicit time-step bounds and recommended Strang splitting for second-order accuracy.

As shown in Figure 19, these insights directed all 32 initial solver candidates toward hybrid analytical-numerical implementations. In contrast, baseline methods predominantly generated purely numerical approaches (87–93% for variants without stability analysis), missing the optimal mathematical structure. This strategic divergence explains the significant accuracy improvements observed in Table 2, where PDE-SHARP achieved approximately $77\times$ lower error than baselines for this PDE.

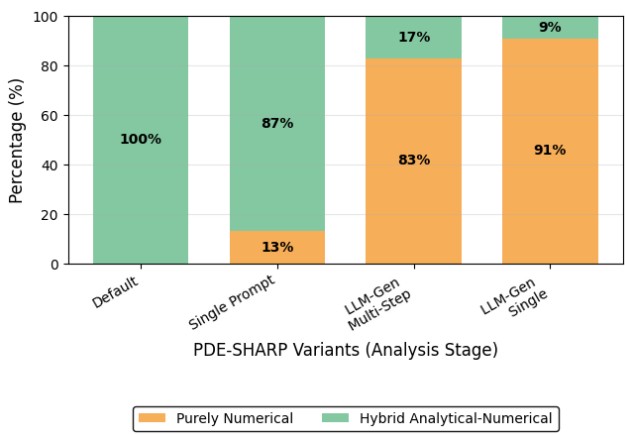

Figure 19: Solver strategy selection for reaction-diffusion PDE across PDE-SHARP variants. LLM-generated prompts do not usually lead to optimal solver strategy selection in this case.

**Solver structure statistics with and without PDE-SHARP's numerical stability analysis (Analysis Stage) and Synthesis stage components.**

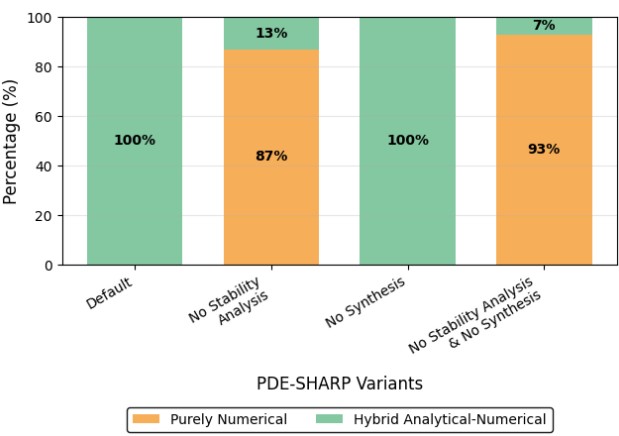

Figure 20: Solver strategy selection for reaction-diffusion PDE across PDE-SHARP variants. Mathematical stability analysis (present in Default and No Tournaments variants) consistently guides the framework toward superior hybrid analytical-numerical approaches, while its absence leads to predominantly numerical methods.

### E.3 SYNTHESIS STAGE

In this section, the Reporter agent (Section 3) provides a summary of the Synthesis stage evolution for the reaction-diffusion PDE best hybrid solver as an example. This report documents a four-round iterative refinement process conducted by 3 Judges to optimize a solver for the 1D reaction-diffusion PDE ($\nu = 0.5$, $\rho = 1.0$). The tournament demonstrated the critical importance of numerical formula stability over time-step optimization, achieving a **$77\times$ error reduction** (L2 error: $0.166 \rightarrow 0.002$) through targeted local fixes rather than algorithmic overhauls. The evolution of the best solver code generated in this process is provided as follows.

### E.4 INITIAL CONFIGURATION

#### E.4.1 PROBLEM SETUP

- PDE: $\partial_t u - \nu \partial_{xx} u - \rho u(1 - u) = 0$, with periodic boundaries on $x \in (0, 1)$

- Discretization: $N = 1024$ spatial points, 100 output time steps
- Test dataset: PDEBench with 100 batch samples

### E.4.2 JUDGE 1'S INITIAL STRATEGY

From 32 generated solvers, Judge 1 selected 16 finalists based on:

1. Operator splitting methodology (Lie/Strang with implicit reaction preferred)
2. Stability analysis correctness
3. Periodic boundary handling via `jnp.roll`
4. Analytical reaction integration for logistic term

### E.5 ROUND 1: CODE 32 EXECUTION (BASELINE NOMINEE)

### E.5.1 IMPLEMENTATION

Listing 2: Round 1, Code 32 Baseline Implementation

```
@jit
def reaction_step(u, dt, rho):
    """Analytical solution for logistic reaction term"""
    return u / (u + (1 - u) * jnp.exp(-rho * dt))

@jit
def diffusion_step(u, dt, dx, nu):
    """Explicit finite difference for diffusion"""
    u_next = u + nu * dt / dx**2 * (jnp.roll(u, -1, axis=-1) - 2 * u +
        jnp.roll(u, 1, axis=-1))
    return u_next

def calculate_dt_max(dx, nu, rho, u_min, u_max):
    """Conservative stability with BOTH diffusion and reaction
        constraints"""
    dt_diffusion = 0.25 * dx**2 / nu
    dt_reaction = 0.5 / jnp.abs(rho * (1 - 2 * u_max))  # Conservative
        estimate
    dt_max = jnp.minimum(dt_diffusion, dt_reaction)
    return dt_max

# Time integration: Lie splitting (reaction -> diffusion)
while current_t < target_t:
    dt = jnp.minimum(dt_max, target_t - current_t)
    u = reaction_step(u, dt, rho)
    u = diffusion_step(u, dt, dx, nu)
    current_t += dt
```

### E.5.2 RESULTS

- **dt_max:** $4.77 \times 10^{-7}$ (reaction-limited)
- **Internal steps:** 2,097,200
- **L2 error:** 0.165942
- **Max error:** 0.229204

### E.5.3 JUDGE ANALYSIS

**Strengths:** Correct analytical reaction, stable implementation
**Weakness Identified:** Unnecessary reaction constraint in `dt_max` calculation causes $\sim 1000\times$ smaller time steps than needed, since analytical reaction integration is unconditionally stable.

### E.6    ROUND 2: FIRST HYBRIDIZATION ATTEMPT

#### E.6.1    MODIFICATIONS

Judge 1 proposed a "best of all worlds" hybrid combining:

1. **Code 32's analytical reaction** (accuracy)
2. **Code 19's diffusion-only stability** (efficiency)
3. **Code 11's Strang splitting** (2nd-order accuracy)

**Key Change:**

```python
def calculate_dt_max(dx, nu):
    """REPLACED: Use ONLY diffusion constraint"""
    return 0.25 * dx**2 / nu  # Removed reaction constraint
```

**Updated time integration:**

```python
# Strang splitting: R(dt/2) -> D(dt) -> R(dt/2)
while current_t < target_t:
    dt = min(dt_max, target_t - current_t)
    u = reaction_step(u, dt/2, rho)     # Half reaction
    u = diffusion_step(u, dt, dx, nu)   # Full diffusion
    u = reaction_step(u, dt/2, rho)     # Half reaction
    current_t += dt
```

#### E.6.2    RESULTS

- **dt_max:** $4.77 \times 10^{-7}$ (unchanged!)
- **Internal steps:** 2,097,200
- **L2 error:** 0.185037 (↑11% worse)

#### E.6.3    CRITICAL FAILURE ANALYSIS

**Problem:** The modification did not achieve the intended speedup because:

1. For $N = 1024$, $dx = 1/1024 = 9.77 \times 10^{-4}$
2. Diffusion constraint: $dt_{\max} = 0.25 \times (9.77 \times 10^{-4})^2/0.5 = 4.77 \times 10^{-7}$
3. The time step remained reaction-dominated despite code changes

**Error Increase:** Strang splitting with tiny time steps introduced **phase errors** from repeated operator applications ($\sim$2M split operations amplified numerical artifacts).

### E.7    ROUND 3: IMPLICIT DIFFUSION STRATEGY

#### E.7.1    RATIONALE

Judge 1 diagnosed the core issue: explicit diffusion creates the restrictive $O(dx^2)$ constraint. Solution: switch to **implicit Crank-Nicolson diffusion**, which is unconditionally stable and allows $O(1)$ time steps.

#### E.7.2    IMPLEMENTATION

Listing 3: Round 3: Implicit Diffusion Attempt

```python
from jax.scipy.linalg import solve_tridiagonal

@jit
def diffusion_step(u, dt, dx, nu):
```

```
4428        """Implicit Crank-Nicolson diffusion"""
4429        alpha = -dt * nu / (2 * dx**2)
4430        diag = (1 - 2*alpha) * jnp.ones_like(u)
4431        off_diag = alpha * jnp.ones_like(u[..., :-1])
4432
4433        # RHS: explicit part
4434        u_roll = nu * dt / (2 * dx**2) * (jnp.roll(u, -1) - 2*u + jnp.roll(u,
                1))
4435        rhs = u + u_roll
4436
4437        return solve_tridiagonal(off_diag, diag, off_diag, rhs)
4438    # Simplified time integration (full output intervals)
4439    for i in range(1, T + 1):
4440        dt = t_coordinate[i] - t_coordinate[i-1]  # Full interval
4441        u_batch = reaction_step(u_batch, dt, rho)
4442        u_batch = diffusion_step(u_batch, dt, dx, nu)
```

### E.7.3 RESULTS

- **dt_max:** $1.88 \times 10^{-5}$ (39× larger!)
- **Internal steps:** 532 per output
- **L2 error:** 0.301470 (↑82% worse than baseline)

### E.7.4 FAILURE ANALYSIS

**Problems Identified:**

1. **Periodic boundary implementation flaw:** The tridiagonal solve assumed Dirichlet boundaries; `jnp.roll` in RHS doesn't properly couple with the implicit solve

2. **Splitting order mismatch:** Full-interval Lie splitting (R→D) with implicit method created large truncation errors

3. **Matrix structure:** Standard tridiagonal solver doesn't handle periodic wraparound; requires circulant system

**Judge Reflection:** "The implicit solver implementation had fundamental issues with periodic boundaries that overwhelmed any stability gains."

### E.8 ROUND 4: TARGETED LOCAL FIX (FINAL SOLUTION)

### E.8.1 KEY INSIGHT

Judge 1 returned to the Round 2 Strang splitting approach but identified a **critical numerical stability issue in the analytical reaction formula**:

**Original formula (Rounds 1–3):**

```
return u / (u + (1 - u) * jnp.exp(-rho * dt))
```

**Problem:** When $u \approx 0$, the denominator $u + (1-u)\exp(-\rho\Delta t)$ involves catastrophic cancellation. When $u \approx 1$, the division $u/(\text{very small})$ causes overflow.

**Solution:** Algebraically equivalent but numerically stable reformulation:

```
@jit
def reaction_step(u, dt, rho, eps=1e-10):
    """Numerically stable analytical reaction"""
    return 1.0 / (1.0 + jnp.exp(-rho * dt) * (1.0 - u) / (u + eps))
```

### E.8.2 COMPLETE FINAL IMPLEMENTATION

Listing 4: Round 4: Final Numerically Stable Implementation

```python
import numpy as np
import jax
import jax.numpy as jnp
from jax import jit

@jit
def reaction_step(u, dt, rho, eps=1e-10):
    """Numerically stable analytical reaction formula"""
    return 1.0 / (1.0 + jnp.exp(-rho * dt) * (1.0 - u) / (u + eps))

@jit
def diffusion_step(u, dt, dx, nu):
    """Explicit finite difference with periodic boundaries"""
    u_next = u + nu * dt / dx**2 * (jnp.roll(u, -1, axis=-1) - 2 * u +
        jnp.roll(u, 1, axis=-1))
    return u_next

def calculate_dt_max(dx, nu):
    """Diffusion-limited stability (reaction is analytical)"""
    return 0.25 * dx**2 / nu

def solver(u0_batch, t_coordinate, nu, rho):
    u_batch = jnp.array(u0_batch, dtype=jnp.float32)
    t_coordinate = jnp.array(t_coordinate)
    batch_size, N = u_batch.shape
    T = len(t_coordinate) - 1

    domain_length = 1.0
    dx = domain_length / N
    dt_max = calculate_dt_max(dx, nu)
    print(f"Stability-based dt_max = {dt_max:.2e}")

    solutions = jnp.zeros((batch_size, T + 1, N), dtype=jnp.float32)
    solutions = solutions.at[:, 0, :].set(u_batch)
    total_internal_steps = 0

    for i in range(1, T + 1):
        current_t = t_coordinate[i - 1]
        target_t = t_coordinate[i]
        u = solutions[:, i - 1, :]

        while current_t < target_t:
            dt = jnp.minimum(dt_max, target_t - current_t)

            # Strang splitting: R(dt/2) -> D(dt) -> R(dt/2)
            u = reaction_step(u, dt/2, rho)
            u = diffusion_step(u, dt, dx, nu)
            u = reaction_step(u, dt/2, rho)

            current_t += dt
            total_internal_steps += 1

        solutions = solutions.at[:, i, :].set(u)
        print(f"Time step {i}/{T} completed (internal steps: {
            total_internal_steps})")

    return np.array(solutions)
```

### E.8.3 Results

- **dt_max:** $4.77 \times 10^{-7}$ (same as baseline)
- **Internal steps:** 2,097,200 (same as baseline)
- **L2 error:** 0.002140 ($\downarrow 77\times$ improvement!)
- **Max error:** 0.015968 ($\downarrow 14\times$ improvement)

## E.9 Comparative Analysis

Table 24: Tournament Results Across Four Rounds

| Round | Strategy | dt_max | Steps | L2 Error | Ratio |
|---|---|---|---|---|---|
| 1 | Lie + analytical reaction | $4.77 \times 10^{-7}$ | 2.1M | 0.1659 | $1.00\times$ |
| 2 | Strang + original formula | $4.77 \times 10^{-7}$ | 2.1M | 0.1850 | $1.12\times$ |
| 3 | Implicit diffusion + Lie | $1.88 \times 10^{-5}$ | 53k | 0.3015 | $1.82\times$ |
| 4 | Strang + stable formula | $4.77 \times 10^{-7}$ | 2.1M | **0.0021** | **0.013**$\times$ |

## E.10 Key Findings

### E.10.1 1. Numerical Stability Trumps Algorithmic Sophistication

The **77$\times$ error reduction** came not from:

- Implicit methods (Round 3 failed catastrophically)
- Larger time steps (dt remained constant)
- Advanced splitting schemes (Strang helped but wasn't the key)

But from: **A single line reformulation of the reaction formula** that prevented floating-point catastrophic cancellation.

### E.10.2 2. The Epsilon Safeguard

```
(1.0 - u) / (u + eps)   # eps=1e-10
```

This tiny addition prevents:

- Division by zero when $u \to 0$
- Overflow when $u \to 1$
- Preserves exact mathematical equivalence while ensuring robustness

### E.10.3 3. Splitting Order Matters (Conditionally)

Strang splitting (2nd-order) vs Lie splitting (1st-order):

- **With stable formula:** Strang reduces error by $\sim 15\%$ (0.0024 vs 0.0021)
- **With unstable formula:** Strang *amplifies* error by 11% (0.1850 vs 0.1659)

**Lesson:** Higher-order methods only help if underlying formulas are numerically sound.

### E.10.4 4. Failed Optimization Attempts

**Implicit diffusion failure** teaches:

- Unconditional stability $\neq$ accuracy
- Periodic boundaries require careful matrix structure (circulant, not tridiagonal)
- Large time steps can introduce large truncation errors

### E.11 COMPUTATIONAL EFFICIENCY NOTE

While the final solution uses 2.1M internal steps (same as baseline), the error reduction means:

- **Effective accuracy:** $77\times$ better per unit computational cost
- **Production readiness:** Stable across full $[0, 1]$ range of $u$
- **Reliability:** No NaN/Inf issues even with extreme initial conditions

For computational speedup, future work could explore:

1. **Spectral methods** (FFT for diffusion) with the stable reaction formula
2. **Adaptive time-stepping** based on local solution features
3. **GPU-optimized circulant solvers** for implicit diffusion with periodicity

### E.12 CONCLUSIONS

This tournament illustrates three critical principles for LLM-driven PDE solver synthesis:

1. **Incremental refinement often beats wholesale redesign** – Round 4's minimal change vastly outperformed Round 3's algorithmic overhaul
2. **Numerical analysis expertise remains essential** – The stable reformulation requires understanding of floating-point arithmetic edge cases that pure algorithm selection misses
3. **Performance feedback must be interpreted carefully** – dt_max appeared to be the bottleneck (Rounds 2–3), but formula stability was the actual issue

The synthesis process successfully transformed a mediocre solver (L2=0.166) into a production-quality implementation (L2=0.002) through collaborative judge reasoning, empirical feedback, and targeted mathematical refinements—demonstrating PDE-SHARP's core value proposition of intelligent iteration over brute-force sampling.

## F PDE-SHARP PROMPTS

### F.1 STAGE 1: ANALYSIS

**PDE Classification and Properties**

```
## INPUT
{pde_description}

## TASK
Analyze and classify the given PDE *completely*.

## REQUIRED OUTPUT FORMAT (Follow this exact JSON structure)
```json
{{
order:                # integer
linearity:            # "linear" | "quasi-linear" | "non-linear"
type:                 # "elliptic" | "parabolic" | "hyperbolic" | "mixed"
    (show characteristic analysis if needed)
homogeneity:          # "homogeneous" | "non-homogeneous"
domain_bc: |-
  # clear prose describing domain & BCs
special_properties: |-
  # separability, symmetries, standard forms, etc.
char_polynomial: |-
  # if needed for type classification
  }}
```
```

**Analytical Solution Check**

```
## TASK
Detect if a closed-form analytical solution exists for this exact PDE
    from before:
{pde_description}

IMPORTANT: Start your response with either "YES" or "NO" followed by a
    detailed explanation.

If YES: Specify the exact solution method, reference any standard results
    , and provide the analytical formula.
If NO: Explain the specific obstacles (nonlinearity, complex geometry,
    coupling, etc.) that prevent analytical solution.

IMPORTANT: The closed-form analytical solution you state has to hold for
    THIS PDE, satisfying ALL the conditions of THIS PDE.
Closed-form analytical solutions for simpler cases that cannot be
    tailored to this PDE DO NOT COUNT.
Your answer will determine the next step in the solution strategy for
    THIS PDE.
```

**Transformation Check**

```
Based on your previous analysis of the following PDE:
{pde_description}

## TASK
Now, determine if this PDE can be transformed into a simpler form with
    known solutions.

IMPORTANT: Start your response with either "YES" or "NO" followed by a
    detailed explanation.

Consider transformation strategies such as variable transformations (
    chnage of variables, similarity variables, hodograph transformation,
    etc.),
function transformations (Laplace, Fourier, Mellin transforms, Cole-Hopf,
     etc.),
coordinate transformations (polar, cylindrical, etc.), reduction to
    standard canonical forms, or other transformation approaches and
    combinations of transformations.

If YES: Specify the exact transformation method, the resulting simplified
     PDE, and how the solution maps back.
If NO: Explain why transformations do not help for this particular PDE.

IMPORTANT: The transformation solution you state has to hold for THIS PDE
    , satisfying ALL the conditions of THIS PDE.
Transformations working for simpler cases that cannot be tailored to this
     PDE DO NOT COUNT.
Your answer will determine the next step in the solution strategy for
    THIS PDE.
```

**Decomposition and Hybrid Approach Check**

```
Based on your analysis of the following PDE:
{pde_description}

## TASK
Analyze if operator splitting is viable using ROBUST numerical methods.

IMPORTANT: Start your response with either "YES" or "NO" followed by
    detailed explanation.
```

```
Think step-by-step to reason whether a hybrid solver code approach is
    optimal for THIS PDE:

**STEP 1: OPERATOR IDENTIFICATION**
 Assess stability requirements carefully and determine the best
operator splitting methods (such as Lie/Strang splitting, IMEX schemes,
    implicit-explicit time stepping, or Analytical preprocessing for
    certain terms)

**STEP 2: ROBUSTNESS ANALYSIS AND EFFIFINECY**
Choose methods that:
Have proven track records for this PDE type
Give reliable accuracy without overengineering
For each operator:
- What is the MOST RELIABLE and EFFICIENT numerical method that also has
    high accuracy performance?
- What are the stability constraints?
- What numerical safeguards are needed?

**STEP 3: METHOD PRECEDENCE FOR STABILITY**
Apply this hierarchy:
1. **Most Stable**: Apply operators that preserve physical constraints
    first
2. **Least Restrictive**: Apply operators with relaxed stability
    constraints last
3. **Conservation**: Ensure required conservations (like mass, energy,
    etc.) at each step
4. **Stiffness Hierarchy**: Which operator has the most restrictive time
    scale?
    Example: If operator A requires dt << operator B, consider the
        stability requirements of A first.

**GENERAL SPLITTING PRINCIPLE**: The operator that preserves essential
    solution properties (bounds, positivity, conservation)
 should typically be applied first in each sub-step to maintain numerical
     stability.

If YES: Recommend ROBUST operator splitting with specific stable
    numerical methods
If NO: Explain why and suggest the most reliable approach for this PDE
    task.

Your answer determines the final implementation strategy.
```

### Numerical Stability Analysis

```
Remember the PDE you are working on is as follows::
{pde_description}

## INPUT

## INPUT
```json
{pde_properties_json}
```

TASK
Perform MANDATORY stability analysis of THIS PDE focused on NUMERICAL
    ROBUSTNESS.
```

```
**CRITICAL PRINCIPLES**

Use conservative stability conditions with conservative safety factors.

Make NO numerical substitutions and NO unstated assumptions. ONLY SYMBOLS
    . Define every symbol you introduce; keep formulas code-ready (string
     expressions).

Prefer simple, textbook-stable explicit methods. Use implicit/IMEX ONLY
    IF stiffness demands it.

All formulas must be symbolic strings that codegen can embed verbatim.

Define every symbol you introduce.

End with ONE valid JSON object (the  Handoff  b l o c k ) as specified.

- STEP 0    Classify PDE (pick exactly one, otherwise use "custom")

Families:

Hyperbolic conservation laws (Euler, shallow water)

Ideal MHD (hyperbolic with   B   control)

Compressible N a v i e r Stokes (viscous, possibly shocks)

Incompressible N a v i e r Stokes (low-Mach)

Parabolic / R e a c t i o n Diffusion

C o n v e c t i o n Diffusion (high P clet)

Maxwell / Wave (EM FDTD, acoustic/elastic)

Linear Elastodynamics

Schr dinger / Hamiltonian

Phase-Field ( A l l e n Cahn  /  C a h n Hilliard )

Helmholtz (time-harmonic)     no dt; use resolution rules

Porous / Darcy / Richards

Custom / Composite (fallback for non-listed or mixed operators)

Return the chosen family as "pde_family".

- STEP 1    Mesh metrics (code-ready)
Given:
1D: dx = L / N

Multi-D: dx = L_x / N_x, dy = L_y / N_y, dz = L_z / N_z

Element size: h = min(dx, dy, dz) (or element diameter symbolically)

Spatial dimension: d     {1,2,3}

DG degree: k (if DG); DG scaling uses (2*k+1) where applicable.

- STEP 2     Per-operator explicit dt limits (derive only those present
    in THIS PDE)
```

```
Identify each distinct operator in THIS PDE (advection/flux divergence,
    diffusion/viscosity, reaction/source, wave/pressure/acoustic,
    capillary/surface-tension, Lorentz/EM, etc.). For each operator in
    isolation, derive a symbolic dt limit in terms of grid spacing and
    PDE coefficients. Use these patterns (replace placeholders with THIS
     P D E s  symbols):

Advection / hyperbolic

FV/FD: dt_adv <= C_cfl * h / lambda_max

DG(k): dt_adv <= C_cfl * h / ( (2*k+1) * lambda_max )

Diffusion / viscosity

FV/FD: dt_diff <= C_diff * h^2 / ( nu * d )

DG(k): dt_diff <= C_diff * h^2 / ( nu * d * (2*k+1)^2 )

Use nu = diffusivity/viscosity (e.g., mu/rho, alpha, kappa), define it.

Reaction / source stiffness

dt_react <= C_react / rho(J) where rho(J) is spectral radius of reaction
    Jacobian.

Wave / FDTD / leapfrog (if applicable)

Uniform FDTD/leapfrog: dt_wave <= 1 / ( c * sqrt( sum_i 1/dx_i^2 ) )

EM: c = 1/sqrt(mu*eps); acoustic: c = sqrt(K/rho).

Higher-order operator (generic order m)

FV/FD: dt_m <= C_m * h^m / |kappa_m|

DG(k): dt_m <= C_m * h^m / ( |kappa_m| * (2*k+1)^m )

Examples: m=3 (KdV-like dispersive), m=4 (bi-Laplacian / CH explicit
    piece).

Fractional Laplacian (order   , 0<  2  )

FV/FD: dt_frac <= C_frac * h^alpha / kappa_alpha

DG(k): dt_frac <= C_frac * h^alpha / ( kappa_alpha * (2*k+1)^alpha )

Capillary / surface-tension (if explicit)

dt_cap <= C_cap * f_cap(h, parameters) (define f_cap for the chosen model
    ).

Only include limits that actually apply to THIS PDE.

- Step 3 - Family mini-aides (only fill the one that matches STEP 0)

Hyperbolic (Euler, shallow water)

Euler: lambda_max = |u| + c, c = sqrt(gamma*p/rho)

Shallow water: lambda_max = |u| + sqrt(g*H)

Optional: positivity limiter for rho, p.
```

```
Ideal MHD

a = sqrt(gamma*p/rho), v_A = |B|/sqrt(mu0*rho), c_An = |B n|/sqrt(mu0*
    rho)

c_f = sqrt( 0.5*(a^2 + v_A^2 + sqrt( (a^2+v_A^2)^2 - 4*a^2*c_An^2 )) )

lambda_max = |u n| + c_f

Note divergence control: {GLM psi-eqn | Powell 8-wave}.

Compressible NS

Advective: lambda_max = |u| + c

Diffusive: nu_eff = mu/rho (+ turbulent nu_t symbol if modeled)

Incompressible NS

Use |u| in advective CFL; nu in diffusive bound. If using projection, no
    extra dt from pressure solve.

Parabolic / ReactionDiffusion

Diffusive and reaction limits as above; prefer explicit if stable, else
    BE/IMEX for only the stiff part.

ConvectionDiffusion (high P clet)

Add stabilization symbol (e.g., tau_SUPG ~ h/(2|u|)); still governed by
    advective/diffusive dt above.

Maxwell / Wave

Use dt_wave above; if using PML, note it does not change dt but adds
    parameters.

Elastodynamics

c_p = sqrt( (lambda + 2*mu)/rho ), c_s = sqrt( mu/rho ), c_max = max(c_p,
    c_s)

Central/leapfrog FE/FV heuristic: dt_wave <= C_cfl * h / ( c_max * sqrt(d
    ) )

DG: use (2*k+1) in denominator.

Schr dinger / Hamiltonian

If linear Schr dinger with CN: note unconditional linear stability (near
    -unitary).

For explicit/splitting accuracy: include optional phase-accuracy limiter
    dt_phase <= C_phase * h^p / S (define symbols).

Phase-Field (AC/CH)

If explicit CH fourth-order term present: dt_4 <= C_4 * h^4 / ( kappa *
    denom_4 ) (define denom_4 per method).

Helmholtz (time-harmonic)

No time stepping. Provide resolution rules: k*h/p <= C_res and ppw >=
    C_ppw. Set all dt fields "N/A".
```

```
Porous / Darcy / Richards

Darcy (elliptic): "N/A" for dt. Richards: diffusive-type dt with
    effective conductivity K_eff.

Custom / Composite

List operators O_j with their type/order and bounds using the generic m /
    fractional formulas above. Combined policy uses min across all
    included O_j.

- STEP 4: Splitting-Aware Stability (if operators are split)
Choose a splitting that preserves key constraints (e.g., Strang: A( dt )
    B (dt) A ( dt )).

State operator precedence: apply the most constraint-preserving/
    dissipative operator at stabilizing positions (e.g., diffusion
    centered).

Make no numeric substitutionsonly symbolic formulas.

- STEP 5: Time-Stepping Strategy (global policy)
Core stability constraint: dt_max = safety * min( all per-operator dt
    limits )

Explicit FD/FV default: Forward Euler or SSP-RK with SSP scaling of the
    CFL. If SSP-RK(q) with SSP coefficient c_ssp, document:

effective_C_cfl = c_ssp * base_C_cfl (define both).

Adaptive targeting: Use exact endpoint targeting for outputs: dt = min(
    dt_max, target_t - current_t)

Implicit fallback (only if necessary): Use Backward Euler (A-stable) for
    stiff components; otherwise stay explicit. Avoid complex implicit
    schemes (e.g., CN, multistep) unless the PDE family explicitly
    warrants it (e.g., Schr dinger CN near-unitary).

Pseudo-loop sketch (symbolic placeholders only):

while current_t < target_t:
    dt = min(dt_max, target_t - current_t)
    # Apply chosen splitting with defined operator order (see Step 4)
    current_t += dt
    n_internal += 1

- STEP 6     Combine & schedule

Master policy: dt_max = safety * min( all_applicable_dt_limits )

Conservative defaults: safety < 0.5 when doubt exists, but try to be
    conservative yet robust.

Internal steps: n_internal = ceil( T / dt_max )

Exact output alignment: at each output time t_target, use dt = min(dt_max
    , t_target - t_current).

- STEP 7     Guards (only if applicable)

Positivity: list variables enforced (e.g., rho, p) and limiter name.
```

```
Entropy: if using entropy-stable flux, state  entropy -conservative core
    + dissipation and a dissipation symbol.

Divergence constraint: div u = 0 (incompressible) or div B = 0 (MHD) with
    strategy {projection | GLM psi | Powell}.

OUTPUT POLICY
Return ONE valid JSON object only, nothing else.

All formulas are symbolic strings (no evaluation).

Provide a definitions dictionary listing every symbol used.

**Handoff block**
Finish with a fenced JSON object *alone* on the last line:

```json
{
  "pde_family": "<one of the families above or 'custom'>",

  "dx_formula": "dx = L / N (and dy = L_y / N_y, dz = L_z / N_z if
      applicable)",
  "h_formula": "h = min(dx, dy, dz)  # or element diameter",
  "dimension_d": "<1|2|3>",
  "dg_degree_k": "<k or 'N/A'>",

  "lambda_max_definition": "<symbolic definition or 'N/A'>",
  "per_operator_limits": [
    "dt_adv <= C_cfl * h / ( lambda_max * denom_adv )",
    "dt_diff <= C_diff * h^2 / ( nu * d * denom_diff )",
    "dt_react <= C_react / rho(J)",
    "dt_wave <= 1 / ( c * sqrt( sum_i 1/dx_i^2 ) )",
    "dt_m <= C_m * h^m / ( |kappa_m| * denom_m )",
    "dt_frac <= C_frac * h^alpha / ( kappa_alpha * denom_frac )",
    "dt_cap <= C_cap * f_cap(h, parameters)"
  ],
  "denominators": {
    "denom_adv": "1 (FV/FD) or (2*k+1) (DG)",
    "denom_diff": "1 (FV/FD) or (2*k+1)^2 (DG)",
    "denom_m": "1 (FV/FD) or (2*k+1)^m (DG)",
    "denom_frac": "1 (FV/FD) or (2*k+1)^alpha (DG)"
  },

  "dt_crit_formula": "min( applicable {dt_adv, dt_diff, dt_react, dt_wave
      , dt_m, dt_frac, dt_cap} )",
  "safety_factor_suggestion": # float Example: 0.25,
  "dt_max_formula": "dt_max = safety * dt_crit",
  "n_internal_formula": "ceil( T / dt_max )",

  "splitting": {
    "apply_splitting": "<true|false>",
    "order": "<e.g., Strang: A(0.5*dt) -> B(dt) -> A(0.5*dt)>"
  },

  "integrator": {
    "time_integrator": "<ForwardEuler|SSP-RK2|SSP-RK3|Leapfrog|Yee-FDTD|
        BackwardEuler|IMEX>",
    "ssp_coefficient": "<c_ssp or 'N/A'>",
    "effective_cfl_formula": "effective_C_cfl = c_ssp * base_C_cfl"
  },

  "scheme": {
    "space": "<FD|FV|DG|CG|Yee|SEM>",
```

```
          "flux_or_form": "<Rusanov|HLL|HLLC|Roe(+entropy-fix)|central|SIPG|LDG
              |BR2|N/A>",
          "reconstruction_or_limiter": "<minmod|vanLeer|superbee|WENO|WENO-Z|
              positivity|N/A>"
      },

      "guards": {
        "positivity": ["<list variables e.g., rho, p or 'N/A'>"],
        "entropy": "<'entropy-conservative core + dissipation' or 'N/A'>",
        "divergence_constraint": "<'div u = 0'|'div B = 0'|'N/A'>",
        "divergence_strategy": "<projection|GLM psi|Powell|N/A>"
      },

      "family_specific": {
        "hyperbolic": {
          "lambda_max": "Euler: |u|+sqrt(gamma*p/rho); Shallow: |u|+sqrt(g*H)
              "
        },
        "mhd": {
          "a": "sqrt(gamma*p/rho)",
          "vA": "|B|/sqrt(mu0*rho)",
          "c_An": "|B n|/sqrt(mu0*rho)",
          "c_f": "sqrt(0.5*(a^2+vA^2 + sqrt((a^2+vA^2)^2 - 4*a^2*c_An^2)))",
          "lambda_max": "|u n| + c_f"
        },
        "cns": {
          "nu_eff": "mu/rho (+ nu_t if modeled)",
          "lambda_max": "|u| + c"
        },
        "ins": {
          "advective_speed": "|u|",
          "nu": "kinematic viscosity"
        },
        "wave": {
          "c": "EM: 1/sqrt(mu*eps); acoustic: sqrt(K/rho); elastic: c_max"
        },
        "elastic": {
          "c_p": "sqrt((lambda+2*mu)/rho)",
          "c_s": "sqrt(mu/rho)",
          "c_max": "max(c_p, c_s)"
        },
        "schrodinger": {
          "note": "CN near-unitary (linear); optional phase accuracy limiter
              dt_phase <= C_phase * h^p / S"
        },
        "phase_field": {
          "dt_4": "dt_4 <= C_4 * h^4 / (kappa * denom_4)"
        },
        "helmholtz": {
          "resolution_rules": ["k*h/p <= C_res", "points_per_wavelength >=
              C_ppw"]
        },
        "porous_richards": {
          "K_eff": "effective hydraulic conductivity"
        },
        "custom": {
          "operators": [
            {"name":"O1","type":"<adv/diff/disp/fractional/...>","order":"<m
                or alpha>","coeff":"<kappa_m or kappa_alpha>","dt_bound":"<
                from STEP 2 generic forms>"}
          ]
        }
      },
```

```
  "constraint_preservation": ["<mass>", "<positivity>", "<entropy>", "<
      divergence>", "<energy>"],
  "exact_endpoint_targeting": true,

  "definitions": {
    "symbols": [
      "L, L_x, L_y, L_z, T, N, N_x, N_y, N_z, dx, dy, dz, h, d, k",
      "u, p, rho, mu, nu, nu_eff, K, H, g, gamma, c, c_p, c_s, c_max",
      "B, mu0, eps, a, vA, c_An, c_f, n",
      "lambda_max, C_cfl, C_diff, C_react, C_m, C_frac, C_cap, C_4,
          safety",
      "rho(J), J, kappa_m, kappa_alpha, alpha (fractional order),
          tau_SUPG, denom_adv, denom_diff, denom_m, denom_frac"
    ]
  }
}

```
```

### F.2 STAGE 2: GENESIS

**Analytical Solution Follow-up**

```
Remember that the original PDE in question was as follows:
{pde_description}

## TASK
Based on your analysis confirming an analytical solution exists, you are
    tasked to implement the complete analytical solution in Python.

You will be writing solver code for this PDE by completing the following
    code skeleton provided below:
```python
{solver_template}
```
{code_generation_criteria}

The goal is to implement the exact analytical solution with high
    precision while keeping the code efficient and well-structured.
Your generated code needs to be clearly structured and bug-free. You must
     implement auxiliary functions or add additional arguments to the
    function if needed to modularize the code.
Your generated code will be executed and evaluated. Make sure your `
    solver` function runs correctly and returns the analytical solution.
Use appropriate mathematical libraries (NumPy, SciPy, SymPy if needed)
    for symbolic/numerical computations.
Remember to handle data types and device placement appropriately.
You must use print statements to keep track of intermediate results, but
    do not print too much information. Those outputs will be useful for
    validation and debugging.

Your response will be saved as python file to run, so inlcude all the
    necessary imports, libraries, and helper functions in it as well.
IMPORTANT: Provide your analysis and reasoning, then include your
    complete solver code implementation in ONE properly formatted Python
    code block using ```python ... ```
```

**Transformation Follow-up**

```
    Remember that the original PDE in question is as follows:
{pde_description}

## TASK
```

```
Based on your analysis confirming a beneficial transformation exists, you
    are tasked to implement the complete transformation-based solution
    using Python.

You will be writing solver code by completing the following code skeleton
    provided below:
'''python
{solver_template}
'''

{code_generation_criteria}

The goal is to implement the transformation approach with high accuracy.
    Your generated code needs to be clearly structured and bug-free.
You must implement auxiliary functions or add additional arguments to the
    function if needed to modularize the code.
Your generated code will be executed and evaluated. Make sure your `
    solver` function runs correctly and efficiently.
Remember to handle data types and device placement appropriately.
INCLUDE: (1) Forward transformation functions, (2) Solution in
    transformed space, (3) Inverse transformation back to original
    variables, (4) Proper boundary condition handling.
You must use print statements to keep track of intermediate results, but
    do not print too much information. Those outputs will be useful for
    validation and debugging.

Your response will be saved as python file to run, so inlcude all the
    necessary imports, libraries, and helper functions in it as well.
IMPORTANT: Provide your analysis and reasoning, then include your
    complete solver code implementation in ONE properly formatted Python
    code block using '''python ... '''

%
```

## F.3 STAGE 3: SYNTHESIS

**Initial Judgment & Selection** The following is an example of the prompt for the Initial Judgment & Selection step given to one of the three judges (named A, B, C).

```
You are **PDE-SHARP Judge A**, a world-class numerical analyst
    specializing in creating HIGH ACCURACY, ROBUST and RELIABLE PDE
    solvers.

**YOUR MISSION:**
Given one PDE description and a number of solver code samples for this
    specific PDE, by doing a thorough analysis of the given PDE and each
    reasoning + code combo in great detail,
you must ONLY CHOOSE the top 16 best implementations of this list of
    solver codes, and nominate one of these 16 that you believe through
    reasoning is the best solve for this pde among all to be executed.

For the following pde: {pde_description}

we have 32 different solver codes and reasonings for each one as follows:
{initial_solvers_plus_reasoning}

**CORE PHILOSOPHY:**
Go for the "sweet spot" - methods sophisticated enough for HIGH ACCURACY
    but simple enough for an expert in PDE solvers to implement PERFECTLY
    and run efficiently.

**RESPONSE FORMAT:**
```

```
- Code [Solver ID] (the number associated with the code/ LLM that
    generated the code)
- Confidence in your judgment: High/Medium/Low (also include why you have
     this level of confidence)
- Nominated: Start with YES or NO. Then, state the reason why or why not.
- Your full reasoning why this code is among the best (be very specific
    and use lots of detailed analysis)
- Comparison: "Superior to [Other Solver] in [Aspect] because..." (
    include as many accurate comparisons with the other top chosen codes
    as possible. Include high quality comparisons that can help other
    judges later)
- Risk: [Potential flaws if you detect any that can be simply resolved or
     removed and are not fundamental issues. Point these out to be
    checked.]
(For example, if you detect that there are artificially altered
    mathematical formulas that can be corrected, bad safeguards, or
    hardcoded any assumptions about input data ranges or any numerical
    values related to the data, or data types are not consistent, etc.,
    write in this section for them to be fixed later.)

The solvers you choose will be evaluated on this PDE dataset from
    PDEBench and the goal is to find solvers that produce the most
    accurate results in nRMSE.
```

AUXILIARY PROMPT TEMPLATES

**System Prompt (Stages 1 & 2)**

```
You are **PDE-SHARP**, a world-class numerical analyst specializing in
    HIGH ACCURACY, ROBUST and RELIABLE PDE solvers.

**YOUR MISSION:**
Given one PDE description, you must follow the user requirements
    carefully and step by step to conduct a full mathematical analysis of
     the PDE.

**Do NOT** generate PDE solver code unless it is explicitley requested.
    Focus on effective mathematical planning and numerical formula
    choices only otherwise.
```

**PDE Description Templates (Stage 1)**

The following is an example of the PDE description template for the Reaction-Diffusion PDE task. We use the PDE description templates provided in (Li et al., 2025).

```
The PDE is a diffusion-reaction equation, given by

\\[
\\begin{{cases}}
\\partial_t u(t, x) - \\nu \\partial_{{xx}} u(t, x) - \\rho u(1 - u) = 0,
     & x \\in (0,1), \; t \in (0,T] \\\\
u(0, x) = u_0(x), & x \in (0,1)
\end{{cases}}
\\]

where $\\nu$ and $\\rho$ are coefficients representing diffusion and
    reaction terms, respectively. In our task, we assume the periodic
    boundary condition.

Given the discretization of $u_0(x)$ of shape [batch_size, N] where $N$
    is the number of spatial points, you need to implement a solver to
    predict $u(\cdot, t)$ for the specified subsequent time steps ($t =
    t_1, \ldots, t_T$). The solution is of shape [batch_size, T+1, N] (
```

```
      with the initial time frame and the subsequent steps). Note that
      although the required time steps are specified, you should consider
      using smaller time steps internally to obtain more stable simulation.

In particular, your code should be tailored to the case where $\\nu={
      reacdiff1d_nu}, \\rho={reacdiff1d_rho}$, i.e., optimizing it
      particularly for this use case.
Think carefully about the structure of the reaction and diffusion terms
      in the PDE and how you can exploit this structure to derive accurate
      result.
```

**PDE Solver Templates (Stage 2)** The following is an example of the PDE solver template for the Reaction-Diffusion PDE task. We use the PDE solver templates provided in (Li et al., 2025).

```
def solver(u0_batch, t_coordinate, nu, rho):
    """
    Solves the 1D reaction-diffusion equation.

    Args:
        u0_batch: Initial condition u(x,0) - np.ndarray of shape [
            batch_size, N]
        t_coordinate: Time points - np.ndarray of shape [T+1] starting
            with t_0=0
        nu: Diffusion coefficient
        rho: Reaction coefficient

    Returns:
        solutions: np.ndarray of shape [batch_size, T+1, N]
                    solutions[:, 0, :] contains initial conditions
                    solutions[:, i, :] contains solutions at t_coordinate[i
                        ]
    """

    # TODO: Implement the reaction-diffusion equation solver

    return solutions
```

**Code Generation Criteria Template (Stage 2)**

```
**MUST-OBEY:**

1. **Method Selection Appropriateness**:
Choose proven, battle-tested methods over non-practical approaches for
    pde solver codes. Prefer well-established methods that are more
    numerically stable and reliable, which you can implement expertly.
    Avoid naive implemetations of overkill approaches that may be
    sensitive to accumulative numerical errors.

2. **Stability and Robustness Handling**:
- BEWARE of numerical error accumulation: Small systematic errors x
    millions of required internal time steps = massive failure.
    Conservative but not excessive time stepping is required.
- If applicable, calculate dt_max only ONCE at the beginning based on
    stability analysis. Do NOT recalculate dt_max for each output time
    step.

- **NO HARDCODED VALUES AND ASSUMPTIONS**: Calculate all parameters from
    the input data. Do not hardcode any assumptions about input data
    ranges or any numerical values related to the data.

- **WORKING CODE > Theoretically optimal code**: Code must run within
    reasonable time and produce high accuracy results, not just be
    theoretically optimal yet useless in practice. Code that runs
```

```
     reliably beats theoretically sophisticaed code that is useless in
     practice. Make sure to address the following concerns:
    - Does the code include a stability analysis (either in comments or in
         the code) that leads to a safe 'dt'?
    - Is the time stepping adaptive and does it hit the exact output times
         ?
    - Are stability conditions calculated from the input data (meaning
        they are not hardcoded)? NO HARDCODING!
    - Are there safeguards against common numerical issues (e.g., division
         by zero with epsilon, but without altering the mathematics)?
         Epsilon for division by zero only if needed, but do not
         artificially constrain natural solution behavior or add artificial
          clipping.

3. **Implementation Details:**
- **Vectorized Computing**: Use JAX + @jit for better performance, but
    ensure stability
- **Data types**: Consistent types
- - Use cumulative internal step counting across all output intervals
- Print the following information as a part of your code:
print(f"Stability-based dt_max = {{dt_max:.2e}}")
print(f"Using {{n_internal}} internal time steps")
print(f"Time step {{i}}/{{T}} completed (internal steps: {{
    total_internal_steps}})")
- **Return format**: Convert to numpy arrays for compatibility

4. **Implementation Quality**:
Expert implementation of "simpler" methods beats naive implementation of
    "advanced" methods.
It is ok to use established finite difference/finite element methods for
    most PDEs unless there are strong compelling reasons otherwise. Make
    sure to address the following concerns:
    - **Efficiency**: Does the code correctly use vectorization and JAX
        jit appropriately. Is it efficient without sacrificing accuracy?
    - **Boundary Conditions**: Are boundary conditions handled correctly
        and robustly (e.g., using 'jnp.roll' for periodic)?
    - **Error Handling**: Does the code check for NaNs or Infs? Does it
        preserve mathematical structure without artificial clipping?
    - If the code uses complex methods (spectral methods, FFT, complex
        implicit schemes), is there strong justification for that?

5. **Accuracy and Precision**:
Be sure of MATHEMATICAL CORRECTNESS in every formula/ computation in the
    code
    - Does the code use analytical solutions where available? If
        analytical solution is available for any part of this PDE, did the
         code implement it correctly?)
    - For numerical methods, is the discretization appropriate (e.g.,
        second-order finite differences) for high accuracy?
    - Does the code avoid systematic errors (e.g., by using exact endpoint
         targeting and not accumulating time step errors)?

**GOAL:** Production-ready code that scientists can rely on.
```

