# OpenReview forum: "PDE-SHARP: PDE Solver Hybrids Through Analysis & Refinement Passes"
_ICLR.cc/2026/Conference — Submitted to ICLR 2026_

### Official Review · Reviewer_xu6V · 2025-10-27

**Soundness:** 3
**Presentation:** 3
**Contribution:** 3
**Rating:** 6
**Confidence:** 4

**Summary:**

This paper introduces PDE-SHARP (PDE Solver Hybrids through Analysis & Refinement Passes), an LLM-driven framework that generates numerical PDE solvers dramatically more efficiently than brute-force sampling methods11. It utilizes a three-stage approach: Analysis (mathematical planning), Genesis (code generation), and Synthesis (collaborative LLM judges and refinement tournaments)2. PDE-SHARP achieves superior solution accuracy, with an average $4\times$ improvement (geometric mean), while cutting computationally expensive solver evaluations by $60\%-75\%$ (requiring $<13$ evaluations vs. $30+$ for baselines).

**Strengths:**

1. PDE-SHARP successfully validates its core hypothesis by substantially reducing the number of expensive solver executions (average $13.2$ evaluations) while delivering higher solution quality than exhaustive sampling baselines4444. This is critical for complex PDEs where simulation costs dominate5.
2. The structured Analysis stage, encompassing PDE classification and stability checks, injects deep domain knowledge before code generation. This leads to high-quality initial code, evident in the system's successful, immediate discovery of superior hybrid analytical-numerical solutions for the Reaction-Diffusion PDE, a strategy missed by baselines
3. The Selection-Hybridization Tournaments enable efficient refinement using performance feedback. This mechanism proves robust to the underlying LLM choice for code generation and demonstrated impressive adaptation by auto-correcting the Advection solver from an analytical method to a finite-volume scheme that better matched the training data reference solution.

**Weaknesses:**

1. High LLM API Cost: While computationally efficient in terms of GPU time, the framework incurs a significant LLM API overhead, with the total cost ($4.01 using GPT-4o) being $5\times$ higher than CodePDE ($0.75) and $7\times$ higher than OptiLLM-CoT ($0.58)12. This high LLM dependency limits immediate practical adoption for users relying on costly API servic.

2. Accuracy/Metric Alignment: The paper notes that achieving the full accuracy gain is sensitive to the alignment between the feedback metric and the final evaluation metric. Suboptimal performance results if the feedback doesn't match the target evaluation

**Questions:**

1. Cost Justification: Given the high LLM cost, quantify the strategic overhead: How many hours of GPU time (for a computationally complex PDE like Navier-Stokes) must be saved to make the $4.01 API cost for PDE-SHARP economically superior to the brute-force baselines?
2. Convergence Order and Quality: Why do the final solvers for complex PDEs like Navier-Stokes and Darcy Flow still show a predominance of first-order convergence (around $60\%-75\%$ of cases) across all methods, including PDE-SHARP? Does the stability analysis prioritize robustness (e.g., first-order upwinding) over formal accuracy order (e.g., second-order finite volume)?
3. I wonder whether the paper has tested on any 3D NS problems which would be more challenging and interesting. Please try it and I will change my rating accordingly with your feedback.

---

> ### Author Response · Authors · 2025-12-02
> **Response to Reviewer xu6V (1/3)**
>
> Thank you for your time and your constructive feedback! We have carefully considered your comments and questions to revise our work.
>
> **Weaknesses:**
>
> **1.** We will address your concern regarding the API cost in three parts:
> - Firstly, users have significant flexibility to manage the API cost due to PDE-SHARP's design, and open-source LLMs can lower the cost too. The core concern about costly API services is addressed in our extensive ablation studies with different LLMs. As shown in the main text (use of open-source LLMs in Table 2) as well as several experiments with additional models, Table 16 in Appendix B.1, ablations using different types of LLMs for the tournaments, Tables 19 and 20 in Appendix B.3.5, PDE-SHARP achieves comparable accuracy using combinations of open-source or less costly API services, reducing the API cost component as desired. The pipeline’s three stages (Analysis, Genesis, and Synthesis) are flexible and could each use different types of LLMs, which allow for strategic model allocation. For instance, the Synthesis stage, which uses a mixture of reasoning and non-reasoning judges in our study and accounts for most API costs, can be easily customized with more cost-effective models with comparable results (detailed in Appendix B.3.5). This provides a direct less expensive path for users prioritizing budget over latency. We report cost using the widely used GPT-4o model to establish a definitive, state-of-the-art performance benchmark for our framework. Crucially, our architecture is model-agnostic.
> - Secondly, the complete cost metric should be viewed as API + GPU.  API cost by itself is an incomplete metric for LLM-driven methods that require intensive test-time solver execution. The true bottleneck for frameworks like CodePDE is the GPU compute required to execute hundreds of candidate solvers for these PDEs. We introduce a customized reasoning framework for PDEs that uses mathematical analysis and collaborative hybrid tournaments to generate more structurally sophisticated, high-accuracy solvers. This replaces expensive GPU-bound solver trials with overall cheaper LLM API calls. Our revised Table 3 now separates and presents both costs (average API and average API +GPU) clearly to avoid confusion. On this metric, PDE-SHARP ranks among the top cheapest methods for the tested PDEs.
> - Finally, the superior accuracy-cost trade-off justifies the API cost even when using more costly API services for higher model performance. Figure 9 (revised manuscript) plots nRMSE against total cost for three examples, showing that PDE-SHARP achieves the best accuracy for a given total cost, particularly for more complex PDEs like Navier-Stokes. PDE-SHARP's solvers are higher quality --- exhibiting better convergence, faster execution overall (Figure 8, Appendix B.4), and superior mathematical structure and stability compared to LLM-generated library calls (detailed comparisons in Appendix D).
>
> **2.** The alignment between feedback and evaluation metrics is another flexible feature of PDE-SHARP, not a weakness. It allows users to tailor the framework to their specific constraints for \textbf{additional performance gain} (as studied in our experiments in Appendix B.3.6, Figure 12), so it does not lead to suboptimality compared with the baselines. We provide examples from our observations for better clarity as follows.
>
> - A metric like nRMSE (even using sparse samples from real-world measurements and simulations) as feedback provides a concrete optimization signal, guiding the synthesis toward refined, high-accuracy solvers depending on the users resources and needs. For example, in the reaction-diffusion case (detailed in Appendix D.2), the judge initially proposed a radical shift to a different solver approach. However, the quantitative nRMSE feedback from testing this new variant showed worse performance, prompting an intelligent course correction: the judge reverted to the more promising original strategy and instead proposed a targeted, local fix that resolved a minor stability issue. This data-driven refinement --- improved by feedback --- transformed that candidate into the highest-performing solver for the task, which none of the other baselines could achieve for this task (see Table 2).
>
> - Without reference data, using physics-based residuals as feedback enforces fundamental constraints, ensuring physically-valid solutions. For example, in the advection case study (detailed in Appendix D.1.2), residual feedback directly penalized violations of mass conservation, successfully guiding the synthesis to stable, conservative solver structures where no reference solution existed.

---

> ### Author Response · Authors · 2025-12-02
> **Response to Reviewer xu6V (2/3)**
>
> **Questions:**
>
> **1.** To quantify economic superiority, consider that a single Navier-Stokes solver evaluation can take approximately 1-2 hours, depending on how much debugging is needed, on a T4 GPU (costing \\$0.35-\\$0.70 per evaluation). PDE-SHARP requires only 9 evaluations on average for this PDE (Table 21), compared to 32+ for baselines. The \\$4.01 API cost for PDE-SHARP is offset by saving at least 23 evaluations, which translates to \\$8.05-\\$16.10 in GPU costs. We have added this clarification to the revised manuscript. Thank you!
>
> **2.** > Does the stability analysis prioritize robustness (e.g., first-order upwinding) over formal accuracy order (e.g., second-order finite volume)?
>
> That’s a key point about robustness versus formal accuracy! If by "robustness" you mean numerical stability, so ensuring the solver remains bounded and handles discontinuities or stiff terms without failure, then our experiments show that the stability analysis in PDE-SHARP prioritizes this. For PDEs like Navier-Stokes with shocks, LLM-generated solvers might often default to first-order methods (e.g., upwinding) because they are inherently more stable and easier to implement correctly. However, PDE-SHARP's Synthesis stage can refine these towards higher-order methods when performance feedback indicates it's beneficial, as seen in the advection case (Figure 4b) and solver samples in Appendix D. The predominance of first-order convergence in some cases reflects the challenge of automatically discovering high-order, stable discretizations for nonlinear problems, but PDE-SHARP improves accuracy within those constraints.

---

> ### Author Response · Authors · 2025-12-02
> **Response to Reviewer xu6V (3/3)**
>
> **Questions (continued):**
>
> **3.** Thank you for this valuable suggestion, which has directly strengthened our paper, and for your interest and openness to improving your score based on these additional results.
>
> To directly address your request and provide a high-level comparison, we have conducted extensive experiments on 3D Compressible Navier-Stokes equations (dataset from PDEBench, inviscid-dominated regime, periodic domain). Full details are provided in Appendix D.4 of the revised manuscript.
>
> Key additions and results:
> 1. PDE-SHARP–generated solvers (three distinct strategies):
> - Solver A.1: Custom finite-volume MUSCL scheme with Rusanov flux, adaptive RK3, and explicit shock-capturing. Implements conservative form --- critical for correct shock dynamics --- and enforces positivity/flooring.
>
> - Solver A.2: Custom pseudo-spectral method using FFT-based derivatives and 4th-order RK. Retains full viscous terms and is tailored for smooth, high-accuracy simulation.
>
>
> - Solver B: Library-integrated spectral solver using Dedalus, with 3/2 dealiasing, symbolic equation parsing, and built-in MPI parallelism. This shows PDE-SHARP can correctly structure calls to high-performance spectral frameworks.
>
> 2. Baseline (Claude Code using high-level PDE libraries to generate solver code):
> Solver C: Uses FEniCSx finite-element library with a monolithic Newton solver. While robust, it employs a low-order (P1) implicit formulation that is unnecessarily heavy for this periodic-box problem and misses opportunities for spectral efficiency.
>
> 3. Observation: PDE-SHARP’s use of libraries is superior!
>
> - Mathematical insight guides choice: PDE-SHARP’s Analysis stage identified the need for conservative formulations and shock-aware discretizations, leading it to generate Solver A.1 (finite-volume) and Solver B (spectral with dealiasing). Solver A.1 adopts a conservative finite-volume method with MUSCL limiting, while Solver B leverages Dedalus --- configured with proper dealiasing and stability-informed time-stepping --- rather than default settings. In contrast, Claude Code defaulted to a generic FEM implementation (Solver C) without exploiting periodicity or recognizing the dominance of hyperbolic character.
>
>
> - Strategic hybridization: Following analysis, the Synthesis stage refines these decisions through structured hybrid evaluations. For instance, finite-volume and spectral methods were directly compared for stability and accuracy. This led to the retention of the library-integrated Dedalus solver, customized with optimal settings, rather than relying on a generic configuration. PDE-SHARP created both custom kernels (A.1, A.2) and library-integrated solvers (B), demonstrating flexibility. Solver B uses Dedalus not as a black box, but with appropriate dealiasing and time-stepping settings derived from stability analysis.
>
> - Avoidance of critical pitfalls: PDE-SHARP’s solvers maintain conservation laws and include proper CFL/viscous time-step controls. Claude Code’s solver, while stable, uses a first-order implicit scheme that is less accurate and computationally heavier than necessary for this benchmark.
>
> - Higher numerical accuracy: Benchmark results on 50 samples from the 3D Navier-Stokes PDEBench dataset show the Dedalus spectral solver (Solver B) achieving the lowest error with mean nRMSE of 1.7% (velocities) and 2.6% (thermodynamic fields). The custom pseudo-spectral solver (Solver A.2) produced mean errors of 5.1% and 6.3% respectively. The finite-volume solver (Solver A.1) showed mean errors of 9.8% and 11.9%, consistent with its shock-capturing numerical dissipation. The finite-element solver (Solver C) exhibited the highest errors at 7.5% and 16.2%, reflecting its first-order spatial discretization on the given mesh resolution.
>
> 4. Detailed outcomes from Appendix D.4:
> - All PDE-SHARP solvers produced working stable code for 3D compressible Navier-Stokes. The finite-volume solver (A.1) implements a production-ready numerical strategy (MUSCL + Rusanov) that is shock-robust. The spectral solver (B) achieves high-order accuracy and uses parallelization-ready Dedalus infrastructure.
>
> - Claude Code’s solver (C), though functional, is overly generic (FEM for a periodic box) and lacks the tailored efficiency of PDE-SHARP’s outputs.

---

### Official Review · Reviewer_h7JT · 2025-10-28

**Soundness:** 3
**Presentation:** 2
**Contribution:** 3
**Rating:** 6
**Confidence:** 3

**Summary:**

This paper proposes PDE-SHARP, an LLM-based framework for solving PDEs. It consists of three stages: an Analysis stage for PDE classification, a Genesis stage for generating different solvers based on the PDE category, and a Synthesis stage for iteratively fine-tuning the solvers. Experiments show that PDE-SHARP significantly improves accuracy across different LLM backbones compared to other methods. An ablation study effectively demonstrates the necessity of all three stages.

Overall, I believe this work is sufficiently innovative and supported by comprehensive experimental analysis, so I am inclined to accept it.

**Strengths:**

- To my knowledge, PDE-SHARP is the first to introduce a PDE classification module. Its detailed analysis phase enables the construction of tailored solvers for different cases, offering strong scalability.

- The experimental section evaluates the performance using various LLM backbones, comprehensively demonstrating the robustness of the method.

- The ablation study effectively validates the necessity of the three stages.

**Weaknesses:**

- The description of the method is relatively brief. It would be better to add an algorithm block to clearly illustrate the complete process, supplemented by concrete examples to aid understanding.

- Although PDE-SHARP requires fewer iterations, it appears to have significant time and API overhead in some cases, as shown in Figure 7b and Table 3.

**Questions:**

- What are the specific differences between the different analysis results shown in Figure 2?

- Can you show some examples of PDE solver code generated by Genesis stage?

---

> ### Author Response · Authors · 2025-12-01
> **Response to Reviewer h7JT**
>
> Thank you for your time and your constructive feedback! We have carefully considered your comments and questions to revise our work.
>
> **Weaknesses:**
>
> **1.** Thank you very much for your comment! We have significantly expanded the method description in the revised manuscript:
> Figure 1 (Section 1) now includes a more detailed flow for the full PDE-SHARP pipeline.
>  We added another detailed diagram in Section 3 (Figure 2), which includes the PDE-SHARP process with the reaction-diffusion task as a concrete example, tracing the decision trail for this PDE to illustrate how the Analysis stage’s mathematical reasoning leads to a refined solver choice in the Genesis stage.
> Appendix E presents a fully annotated PDE-SHARP report for the same equation, detailing the entire decision process from initial classification through to the final refined code from the Synthesis stage at the end. It details how one solver gets refined through the hybrid tournaments and how the judges improve solvers from the feedback collaboratively.
>
> **2.** We have resolved the confusion in Table 3. The costs in Table 3 (previous version) reflected only the API call cost used to reduce the number of solver executions as well as producing higher-quality solvers for PDE-SHARP. We have now revised Table 3 (new manuscript) to separate API cost from the total (API + GPU) cost and reflect both clearly. As you observe, PDE-SHARP is amongst the top cheapest methods based on average total (API + GPU) cost for the tested PDEs while achieving the best performance (Table 2). Adding the solver execution (GPU) costs to this table demonstrates the trade-off: PDE-SHARP replaces GPU compute (solver evaluations) with cheaper LLM compute (API calls). As shown in Figure 9 (revised manuscript), which plots nRMSE against total cost (API + GPU), PDE-SHARP achieves a superior accuracy-cost trade-off, especially for PDEs like Navier-Stokes and reaction-diffusion.
>
> The higher execution time for reaction-diffusion in Figure 7b is a direct result of the framework correctly selecting more rigorous, high-accuracy methods during stability analysis, which is justified by the significant accuracy gains shown in Table 2 (approx. 7x better accuracy for this particular PDE compared with the top baseline). The new Figure 8 added shows the convergence analysis for reaction-diffusion, indicating a superior convergence rate (order 2) for the PDE-SHARP solver compared with the baselines (orders <1). Given the dramatic improvement in solution quality, this is a valuable trade‑off for PDE-SHARP, which is able to produce solvers with both higher accuracy and better numerical properties compared with all the baselines.
>
> **Questions:**
>
> **1.** Figure 2 (revised manuscript) illustrates the decision tree of the Analysis stage (with reaction-diffusion as an example). The differences in the output are the specific mathematical insights and directives passed to the Genesis stage. For a given PDE, the analysis results can differ in:
> - PDE Classification: Order, linearity, and type (e.g., hyperbolic vs. parabolic).
> - Solution Path: Whether an analytical solution exists, a simplifying transformation is possible, or operator splitting is viable.
> - Stability Regime: The derived symbolic time-step bounds (e.g., CFL condition for advection vs. diffusion-limited steps).
>
> For example:
> - In the case of reaction–diffusion, the analysis identifies that the reaction term has an exact solution, guiding all solvers to use Strang splitting with analytical integration of the reaction component.  Through our extensive ablation studies, we have shown that skipping the Analysis stage results in predominantly purely numerical approaches, missing the hybrid strategy and leading to ~87–93% lower accuracy and increased instability (Figure 6 and 20, Table 18).
> - For advection, the analysis detects CFL constraints and steers solvers toward finite-volume schemes based on the analysis and feedback.
>
> The different analysis results directly influence the numerical schemes produced during the Genesis stage. We have added several solver examples in Appendix D (revised manuscript) and a complete analysis of their approaches, differences, and how they evolve.
>
> **2.** Absolutely. Appendix D now contains detailed case studies with full code examples for the Advection, Reaction-Diffusion, 1D Navier-Stokes, and 3D Navier-Stokes equations with both PDE-SHARP and LLM library-based solvers (such as FEniCS, PETSc, deal.II, JAX‑CFD, etc.) available for each case. Detailed comparisons of these solvers are also available in the revised manuscript; in terms of metrics such as nRMSE (Tables 2 and 17) and convergence rates and execution times (Appendix B.4 and Figure 8), as well as comparing them based on mathematical structure (The whole Appendix D).

---

### Official Review · Reviewer_KcJb · 2025-10-30

**Soundness:** 2
**Presentation:** 2
**Contribution:** 2
**Rating:** 2
**Confidence:** 5

**Summary:**

This paper introduces PDE-SHARP, a three-stage framework for generating numerical solvers for Partial Differential Equations (PDEs) using Large Language Models (LLMs). The core idea is to replace expensive, brute-force evaluation of many solver candidates with a more intelligent process that leverages cheaper LLM inference. The framework consists of: (1) an **Analysis** stage, where an LLM performs a mathematical chain-of-thought analysis of the PDE, including classification and symbolic stability analysis, to create a solver plan; (2) a **Genesis** stage, which generates initial solver candidates based on this plan; and (3) a **Synthesis** stage, which uses collaborative "tournaments" of LLM judges to iteratively refine and select the best solver through performance-informed feedback.

The authors demonstrate that PDE-SHARP can generate high-quality solvers with 60-75% fewer computational evaluations compared to baseline methods like `CodePDE`. Across five representative PDE tasks, their method achieves an average accuracy improvement of 4x and shows robustness to the choice of the underlying generator LLM.

**Strengths:**

1.  **Originality & Significance:** The central thesis—swapping expensive scientific computation with cheaper, structured LLM inference—is highly significant and timely. The framework moves beyond the naive "generate-and-test" paradigm. The **Analysis** stage, in particular, is a novel and powerful concept. Tasking the LLM to perform a symbolic, mathematical pre-analysis (e.g., deriving stability bounds like the CFL condition) before writing any code is a major step towards integrating genuine mathematical reasoning into automated scientific discovery. This has the potential to make LLM-driven science far more computationally tractable.

2.  **Quality & Clarity:** The paper is exceptionally well-written and clearly structured. The PDE-SHARP framework is thoughtfully designed, with each stage serving a distinct and logical purpose. The **Synthesis** stage, with its collaborative tournaments and flexible feedback mechanisms (nRMSE, PDE residual, etc.), is a sophisticated and well-conceived approach to iterative refinement. The ablation studies (e.g., Figure 6) effectively demonstrate the individual contributions of the analysis and synthesis components, adding to the quality and rigor of the work *within its defined scope*.

3.  **Robustness:** The paper convincingly shows that the tournament-based synthesis stage mitigates the weaknesses of individual generator LLMs, leading to consistently strong performance regardless of the base model (Table 2). This is a valuable finding, suggesting the framework's architecture is more important than the specific LLM used for the initial code generation.

**Weaknesses:**

Despite its clever design, the paper's evaluation suffers from a fundamental methodological flaw that undermines its central claims about solver quality.

1.  **Fundamentally Misguided Evaluation Paradigm:** The paper's primary weakness is its evaluation context. It demonstrates that PDE-SHARP is superior to other LLM frameworks that also generate Python solvers *from scratch*. However, this is not the relevant benchmark for practical scientific computing. The state-of-the-art for solving PDEs is not to write low-level finite difference loops in Python, but to orchestrate highly-optimized, mature numerical libraries (e.g., **FEniCS**, **PETSc**, **deal.II**, or JAX-based toolkits like **JAX-CFD**).
    *   A human expert, whom the paper implicitly claims to compete with, would almost never write a production solver from scratch in Python for the problems tested. They would write a short script to call a powerful library.
    *   By focusing on generating code from scratch, the paper is optimizing a suboptimal workflow. The impressive "Analysis" stage is spent reasoning about the stability of a simple, from-scratch finite difference scheme, rather than reasoning about how to best use a powerful, pre-existing tool. This limits the practical significance of the results.

2.  **Insufficient Baselines and Metrics:** The comparison is made against other LLM-driven methods (`CodePDE`, `OptiLLM`) but not against a truly strong, non-LLM baseline. While the paper improves upon `CodePDE`, it inherits the same evaluation weakness: comparing against a method that itself was not benchmarked against the practical state-of-the-art. The primary metric is final `nRMSE` on a fixed grid, which is insufficient for evaluating numerical algorithms. A rigorous comparison requires **convergence analysis**, showing how error decreases with grid refinement and computational cost (wall-clock time). Without this, it is impossible to assess the true order of accuracy and efficiency of the generated solvers.

3.  **Limited Problem Complexity:** The chosen PDE test cases (1D advection, 1D Burgers, etc.) are standard but relatively simple. They feature simple geometries (1D/2D boxes) and boundary conditions (periodic). These are scenarios where even basic, from-scratch methods can perform adequately. The framework's utility is not tested on problems where the choice of numerical method is truly critical, such as those with complex geometries (requiring FEM), multi-scale phenomena, or severe stiffness, which are ubiquitous in science and engineering. It is unclear if the "from-scratch" paradigm would scale to these challenges at all.

4.  The paper fails to benchmark against readily available, state-of-the-art general-purpose code agents (e.g., Cursor, the Claude code). These tools excel at the practical task of generating high-level scripts that leverage powerful, pre-existing numerical libraries. My own experiments suggest that for the problems discussed, these agents can produce correct, library-calling scripts with a success rate exceeding 95%, where the solution's accuracy is guaranteed by the underlying traditional algorithm. By not comparing against this highly effective and simpler workflow, the paper misses the most relevant baseline.

5.  The quality of several figures is poor, with noticeable distortion and low resolution. For example, Figure 5 is blurry and difficult to read. The authors should revise all figures to ensure they are high-resolution and legible for the final version.

**Questions:**

1.  The paper's core premise is that generating low-level solver code from scratch is a valuable goal. However, the scientific computing community has invested decades in building robust, high-performance libraries to abstract away these low-level details. Could the authors justify why generating from-scratch solvers is a more promising direction than prompting an LLM to act as an "expert user" that writes high-level scripts to **call libraries like FEniCS or PETSc**? How would the performance, and more importantly, the scalability of PDE-SHARP to complex, real-world problems, compare to this alternative library-centric paradigm?

2.  The evaluation relies heavily on the final `nRMSE` value. For numerical methods, this can be misleading. Could the authors provide **convergence plots** (e.g., log-log plots of error vs. grid spacing $\Delta x$, and error vs. wall-clock time) for at least one or two of the PDE tasks? These plots should compare the best solver from PDE-SHARP against a canonical, well-implemented solver from a standard library (e.g., a second-order finite volume method for advection, or a spectral method for Burgers' equation). This would provide a much more meaningful assessment of the generated solver's accuracy and efficiency. My assessment of the paper would improve substantially if such a methodologically sound comparison were presented.

3.  The mathematical `Analysis` stage is a key strength. However, it appears to focus on deriving stability for explicit finite difference or simple hybrid schemes. How would this stage generalize to problems where more advanced discretizations are required, such as the Discontinuous Galerkin (DG) method for conservation laws or spectral methods for smooth problems? Does the LLM possess the deep mathematical knowledge to perform a symbolic stability analysis for these more complex schemes, or is the framework's intelligence implicitly limited to a narrow class of numerical methods?

4.  There are many mature general-purpose code agents on the market, but this paper does not compare them with those such as Cursor and Claude Code. Based on my personal testing, for the problems discussed in this paper, having a general-purpose code agent call a commonly used PDE library can achieve a success rate of over 95%, and the solution accuracy can be guaranteed by traditional algorithms. Based on this, I would like to ask the author: 1. Could you add a comparison with these general-purpose code agents? 2. Compared to them, what is the applicable scope of the algorithm presented in this paper, and what are its advantages? Please demonstrate this with experiments.

---

> ### Author Response · Authors · 2025-12-02
> **Response to Reviewer KcJb (1/4)**
>
> **Weaknesses:**
>
> **1.** Thank you for your comments. We would like to clarify several points on the characterization of our evaluation paradigm that appear to be based on a misunderstanding of PDE-SHARP’s goals and results.
>
> 1. > "It demonstrates that PDE-SHARP is superior to other LLM frameworks that also generate Python solvers from scratch.", "By focusing on generating code from scratch, ... "
>
> PDE‑SHARP does not solely focus on “from‑scratch” solvers you suggest --- it uses libraries when they are the best choice. As noted in Appendix B.2 (and in the full prompts provided in Appendix F, available in both the original and revised manuscripts), PDE‑SHARP explicitly encourages and permits the use of high‑level PDE libraries (FEniCS, PETSc, deal.II, JAX‑CFD, etc.). The choice between library calls and custom implementations is left to the LLM’s discretion during the Genesis and Synthesis stages, where the judges refine solvers through feedback. In fact, several of the best solvers reported in Table 2 are library‑based. For example:
>
>  - For the 1D Navier‑Stokes task, the PDE‑SHARP‑generated solver (Appendix D.3, Solver A) is a JAX‑CFD‑based implementation that outperforms the competing Claude‑Code‑generated library solver (Solver B, which is also based in JAX-CFD) in both accuracy and numerical stability. A comparison is included in Appendix D.3 and Tables 2 and 17.
>
> - For Darcy flow, PDE‑SHARP solvers frequently call SciPy and JAX routines (Table 22 and Figure 17), creating higher order solvers than all the other methods tested.
>
> Thus, PDE‑SHARP is not “optimizing a suboptimal workflow”; it adaptively selects the appropriate level of abstraction --- sometimes generating custom code when that yields higher accuracy and better numerical stability (e.g., the hybrid analytical‑numerical scheme for reaction‑diffusion, which no baseline LLM discovered, not even Claude Code as you suggested --- see Appendix D.2), and sometimes invoking libraries when they are more suitable (e.g. Navier-Stokes, Darcy Flow)
>
> 2. The reviewer suggests we are implicitly claiming to compete with human experts who would call optimized libraries. This misinterprets our contribution. Our goal is not to compete with human experts, especially not without any fine-tuning on the LLMs, but to **diagnose and improve currently deficient LLM‑driven solver generation.** PDE‑SHARP addresses a methodological gap in the current work on LLM‑driven PDE code generation using test-time computing strategies. Recent work such as CodePDE [1], OptiLLM [2], other larger agentic workflows such as FunSeach [3], AIDE [4], etc. remain far from saturating standard benchmarks. They generally have two important issues: (1) They fail to produce high-quality mathematically-informed solvers as they lack effective PDE‑specific reasoning and analysis. (2) Their brute-force sampling requires a large number of solver executions, hundreds in some experiments, which is highly inefficient. PDE-SHARP systematically studies these failures and introduces a framework that integrates mathematical analysis (Analysis stage, especially the numerical stability analysis) with efficient, feedback‑driven refinement (Synthesis stage) to significantly boost LLM performance and replace exhaustive solver executions with fewer API calls. The baselines we compare against are the state‑of‑the‑art in *LLM‑driven PDE solver generation.*
>
> 3. > "The impressive "Analysis" stage is spent ..., rather than reasoning about how to best use a powerful, pre-existing tool."
>
> The reviewer suggests that the Analysis stage should be spent reasoning about using powerful pre‑existing tools, but this is what the Analysis stage does. the Analysis stage is precisely designed to reason about the best solution strategy exploiting the mathematical structure and numerical properties of the PDE. This reasoning leads to the *correct, mathematically-informed* use of powerful tools --- whether custom hybrid methods or library calls --- in the subsequent code generation and refinement stages. This stands in stark contrast to the 'blind' or inappropriate use of these tools that we observe in other frameworks. For instance, in the reaction‑diffusion case, the Analysis stage identifies that the reaction term admits an exact analytical solution, directing the solver toward a hybrid analytical‑numerical strategy. This strategy outperforms all baselines (including the Claude‑Code‑generated FEniCS solver in Appendix D.2) because it exploits mathematical structure that other frameworks fail to leverage. In another example on Navier‑Stokes, Claude Code uses JAX‑CFD as a black box, adopting a spectral formulation that produces spurious oscillations in shocks. PDE‑SHARP’s Analysis stage instead selects a more conservative formulation within the same library, leveraging a mathematically-informed finite‑volume scheme to produce stable, correct solutions.

---

> ### Author Response · Authors · 2025-12-02
> **Response to Reviewer KcJb (2/4)**
>
> **Weaknesses (continued):**
>
> **2.** In the original manuscript, Appendix B.4 already provided statistics on solver execution times, empirical convergence orders, and library usage (Figures 14, 15, 17 and Table 22). In the revised version, we have further strengthened this evaluation by adding Figure 8 to the main text (Section 4.1.1), which shows the convergence rate and execution time of the generated advection and reaction‑diffusion solvers across grid refinements and expanding Appendix B.4 with Figure 16 for the rest.
>
> These additions provide a comprehensive view of the accuracy–efficiency trade‑off and the true order of accuracy of the solvers generated by PDE‑SHARP and the baselines. The results confirm that PDE‑SHARP not only achieves lower error on a fixed grid but also produces solvers with superior convergence properties and competitive run‑times.
>
> **3.** Thank you for your comment regarding problem complexity. We address your concern in two parts:
>
> - The goal of this work is to study and improve LLM‑driven PDE solver generation where it is currently insufficient.
> The field is not yet solving even standard benchmarks robustly. Current LLM‑driven test‑time computing frameworks (e.g., CodePDE [1], OptiLLM [2], agentic workflows like FunSearch [3] and AIDE [4]) remain far from saturating standard 1D/2D tasks --- for example, as shown in Table 2 and Appendix A.3.2, they produce errors >0.9 nRMSE on reaction‑diffusion and fail to converge (Figure 8). These failures occur because existing frameworks lack mathematical analysis and efficient refinement. If a framework cannot identify an exact ODE solution for a reaction term, it cannot properly scale to more complex problems. PDE‑SHARP addresses this methodological bottleneck by introducing PDE‑specific mathematical reasoning (Analysis) and efficient feedback‑driven refinement (Synthesis), thereby enabling LLMs to generate solvers that are both accurate and numerically sound on canonical tasks.
>
> - Our benchmark suite is chosen to diagnose and address core methodological failures in a controlled, interpretable setting. To fix inefficient, low‑quality solver generation, we need benchmarks that are: (1) diverse in mathematical structure (hyperbolic, parabolic, elliptic, linear, nonlinear) to test different solver strategies, (2) rich in real‑world numerical behaviors (transport, shocks, stiff coupling) to stress‑test robustness, and (3) interpretable so failure modes can be precisely diagnosed. The PDEBench tasks satisfy all three. For example, reaction‑diffusion tests the ability to discover hybrid analytical‑numerical strategies; Burgers and Navier‑Stokes test shock‑capturing and conservation; Darcy tests handling of elliptic systems. Our ablation studies (Appendix B.3) confirm that different PDEs exercise different components of PDE‑SHARP, demonstrating its flexibility and PDE‑awareness.
>
> To directly address your comment about scalability, we have included new experiments on 3D compressible Navier‑Stokes equations (inviscid‑dominated regime, periodic domain) in Appendix D.4 of the revised manuscript. Results in Appendix D.4 show that our framework can handle higher‑dimensional problems and more complex physics, building on the foundational methodological improvements established in 1D/2D. PDE-SHARP-generated solvers for 3D CNS benefit from mathematical guidance and insights from Analysis and refinement from Synthesis, generating a solver that leverages Dedalus library --- configured with proper dealiasing and stability-informed time-stepping --- rather than default settings. In contrast, Claude Code defaulted to a generic FEM implementation without exploiting periodicity or recognizing the dominance of hyperbolic character.
>
> Future work will extend to problems with complex geometries and boundary conditions, but first the core failures in LLM‑driven solver generation must be studied and addressed properly and in an interpretable way, which is the contribution of this paper.
>
> **4.** The original manuscript already reports results using FunSearch [3] and AIDE [4], which are large-scale agentic workflows (Table 11), which still fail on numerical reasoning-sensitive tasks such as reaction-diffusion. To address your comment on the use of state-of-the-art code agents such as Claude Code, we have scaled our experiments in the revised manuscript. We added Appendix B.2 & D: Comprehensive comparison against high-level library-based solvers. We used Claude Code (a state-of-the-art coding agent) to generate solvers that directly call established PDE libraries (FEniCS, JAX-CFD, etc.), which we will elaborate more on in the next response to your Questions 1 and 4 relevant to this topic.
>
> **5.** Thank you for pointing this out. The figures are improved in the revised manuscript.

---

> ### Author Response · Authors · 2025-12-02
> **Response to Reviewer KcJb (3/4)**
>
> **Questions:**
>
> **1 \& 4.** Thank you for these related questions. We address them jointly to clarify PDE‑SHARP’s goals, its use of libraries, and its advantages over general‑purpose code agents. Our argument has three parts:
>
> 1. Firstly, one misunderstanding we wish to address again is that PDE‑SHARP solely generates "from‑scratch" solvers. **This is not the case** as we explained in detailed in our response to Weakness 1. As stated in Appendix B.2 and embodied in the full prompts (Appendix F), PDE‑SHARP explicitly encourages and permits the use of high‑performance libraries (FEniCS, PETSc, JAX‑CFD, etc.). The choice between a custom implementation and a library call is left to the LLM’s discretion during the Genesis and Synthesis stages, where solvers are refined through feedback. In practice, many of the best solvers reported in Table 2 are library‑based (see library usage statistics in Table 22 and Figure 17, and the solver examples in Appendix D.3 and D.4).
>
> Therefore, PDE‑SHARP’s goal aligns with the reviewer’s suggestion: to act as an "expert user" that writes high‑level scripts to call powerful libraries correctly. The framework’s value lies in ensuring this usage is mathematically informed and numerically robust, not merely automated.
>
> 2. To directly evaluate the "library‑centric paradigm" you suggest, we added a dedicated baseline in the revised manuscript: "Claude Code + PDE Libraries" (Appendix B.2, Table 17). This baseline represents a state‑of‑the‑art, general‑purpose code agent (Claude Code) explicitly instructed to generate solvers by calling appropriate high‑performance libraries for each PDE task. Appendix D provides full case studies with code comparisons for Advection, Reaction‑Diffusion, 1D Navier‑Stokes, and 3D Navier‑Stokes. The results show that PDE‑SHARP consistently matches or surpasses this library‑focused agent:
>
> - Overall Accuracy (nRMSE): As shown in Table 17, PDE‑SHARP outperformed the Claude‑Code‑based library solver on 4 out of 5 PDEs and was comparable on Darcy Flow.
>
> - Numerical Robustness: Beyond final error, PDE‑SHARP solvers exhibit more stable and predictable convergence under grid refinement (Figure 8 and Appendix B.4), whereas library‑based solvers sometimes show erratic or failed convergence, especially for problems with shocks or stiffness.
>
> 3. PDE‑SHARP’s key advantage is mathematically informed algorithm design, leading to superior solver quality even when using the same libraries.
>
> The most significant distinction is not *whether* libraries are used, but *how*. PDE‑SHARP’s Analysis stage provides the mathematical reasoning to select the correct algorithm and formulation for the problem, a step that generic agents skip. Two examples from Appendix D illustrate this decisively:
>
> - Reaction‑Diffusion (Appendix D.2): PDE‑SHARP’s Analysis identifies that the reaction term admits an exact analytical solution. It therefore generates a hybrid Strang‑splitting solver that integrates the reaction exactly. The Claude‑Code agent defaulted to a generic, first‑order IMEX finite‑element method using FEniCS, failing to exploit this structure and resulting in significantly higher error and stiffness.
>
> - 1D Compressible Navier‑Stokes (Appendix D.3): Both PDE‑SHARP and the baseline used the JAX‑CFD library. However, PDE‑SHARP’s analysis led to a more conservative formulation (solving for (ρ, ρv, E)) with an SSP‑RK3 time integrator for stability, suitable for shocks. The Claude‑Code agent produced a non‑conservative spectral method in primitive variables, which leads to spurious oscillations This shows PDE‑SHARP’s ability to selectively integrate library features based on mathematical necessity, rather than defaulting to library conventions.
>
> Conclusion: PDE‑SHARP’s scope is **LLM‑driven generation of numerically robust, mathematically-informed PDE solvers, whether custom or library‑based.** Its advantage over general‑purpose agents is its integrated mathematical analysis, which prevents the "black‑box" misuse of powerful tools and leads to superior algorithm choices, accuracy, and robustness—as demonstrated in our experiments. This methodological improvement is a prerequisite for reliably scaling to more complex, real‑world problems.
>
> **2.** Thank you for your comment and your interest and openness to improving your score based on these additional results. In Section 4.1.1 of the main text (Code Quality & Insights), we added Figure 8, which shows: (1) Convergence rates (empirical order of accuracy) for the advection and reaction‑diffusion solvers generated by PDE‑SHARP and all baselines across multiple grid refinements, (2) Execution times vs. grid resolutions for the same solvers, providing a direct view of the accuracy‑efficiency trade‑off. These plots show that PDE‑SHARP not only achieves lower error on a fixed grid but also produces solvers with superior convergence properties (order 2 for the PDE-SHARP solvers compared with the baselines orders <1) and competitive run‑times.

---

> > ### Author Response · Authors · 2025-12-02
> > **Response to Reviewer KcJb (4/4)**
> >
> > **Questions (continued):**
> >
> > **2. (continued)** Additionally, the following results were already included in the original manuscript, Appendix B.4: Figure 14: Average execution times across all PDE tasks, comparing PDE‑SHARP against baselines, Figure 15: Convergence‑order distributions across different PDEs (complementing Figure 8), Table 22 and Figure 17: Detailed library‑usage statistics, highlighting when solvers leverage high‑performance libraries versus custom implementations.
> >
> > **3.** Thank you for highlighting the important question of generalizability to advanced discretizations. Our response addresses the scope, capability, and evidence of generality in PDE‑SHARP’s Analysis stage.
> >
> > 1. The Analysis stage is not limited to explicit finite differences; it is a structured mathematical reasoning template that can guide the selection and analysis of any discretization method the LLM can reason about. The exact stability analysis prompt (Appendix F) does not prescribe a specific scheme. Instead, it instructs the LLM to:
> >
> > - Analyze individual operators (e.g., diffusion, reaction, advection)
> >
> > - Consider operator interactions
> >
> > - Propose a time‑stepping strategy that respects stability constraints
> >
> > The LLM is free to recommend any method it deems suitable --- finite difference, finite volume, finite element, spectral, or hybrid --- based on the PDE’s mathematical properties. The framework is agnostic to the discretization choice as the prompts leave room for flexibility. Any specific method that the user wants to emphasize on or strictly enforce could be incorporated within the prompts.
> >
> > 2.  PDE‑SHARP can generate and reason about advanced methods, including spectral and finite‑element schemes. In Appendix D.4 (3D Compressible Navier‑Stokes), PDE‑SHARP generates a pseudo‑spectral solver (Solver A.2) and a Dedalus‑based spectral solver (Solver B). The Analysis stage correctly identifies the periodic domain and smooth solution behavior, leading to the selection of spectral methods. In Appendix D.2 (Reaction‑Diffusion), the baseline Claude‑Code agent produces a finite‑element solver using FEniCS. While this solver underperforms relative to our hybrid scheme, it demonstrates that library‑based FEM generation is within the scope of the framework. In Appendix D.3 (1D Navier‑Stokes), both PDE‑SHARP and the baseline use the same JAX‑CFD library, but PDE‑SHARP’s Analysis stage selects a better-performing finite‑volume formulation (appropriate for shocks). This shows that the Analysis stage can reason about formulation suitability, not just low‑level stability.
> >
> > 3.  Analysis stage can be extended with prompts that guide the LLM to consider and enforce any scheme the user wants, for example, energy stability (for DG) or eigenvalue analysis (for stiff systems). The modular design of PDE‑SHARP allows such extensions without altering the core framework because of the flexible nature of each stage and the prompts.
> >
> > 4. PDE-SHARP's results demonstrates that with carefully designed analysis prompts and collaborative hybrid tournaments, the LLM can perform non‑trivial mathematical reasoning (e.g., identifying exact solvability, selecting conservative formulations). We have acknowledged in our paper LLMs may currently lack the depth of knowledge for highly specialized tasks due to limitations in LLM training data, but it is definitely not limited to a narrow class of numerical methods as we have demonstrated through our extensive experiments. Moreover, PDE-SHARP is flexible and extensible by incorporating fine‑tuned LLMs for highly specialized tasks.
> >
> > References:
> >
> > [1] Shanda Li, Tanya Marwah, Junhong Shen, Weiwei Sun, Andrej Risteski, Yiming Yang, and Ameet Talwalkar. Codepde: An inference framework for llm-driven pde solver generation, 2025. URL https://arxiv.org/abs/2505.08783.
> >
> > [2] Asankhaya Sharma. Optillm: Optimizing inference proxy for llms, 2024. URL https:// github.com/codelion/optillm.
> >
> > [3] Bernardino Romera-Paredes, Mohammadamin Barekatain, Alexander Novikov, Matej Balog, M. Pawan Kumar, Emilien Dupont, Francisco J. R. Ruiz, Jordan Ellenberg, Pengming Wang, Omar Fawzi, Pushmeet Kohli, and Alhussein Fawzi. Mathematical discoveries from program search with large language models. Nature, 2023. doi: 10.1038/s41586-023-06924-6.
> >
> > [4] Zhengyao Jiang, Dominik Schmidt, Dhruv Srikanth, Dixing Xu, Ian Kaplan, Deniss Jacenko, and Yuxiang Wu. Aide: Ai-driven exploration in the space of code, 2025b. URL https://arxiv. org/abs/2502.13138.

---

### Official Review · Reviewer_51hJ · 2025-11-01

**Soundness:** 2
**Presentation:** 3
**Contribution:** 2
**Rating:** 4
**Confidence:** 4

**Summary:**

PDE-SHARP introduces a three-stage LLM-driven framework for generating PDE solvers that reduces computational evaluations by 60-75% while improving accuracy by 4× on average. Evaluated on five PDEs from PDEBench, PDE-SHARP requires fewer than 13 solver evaluations versus 30+ for baselines, demonstrates robustness across diverse LLM architectures.

**Strengths:**

1.	Comprehensive experiments across five diverse PDEs demonstrate consistent improvements over multiple competitive baselines.
2.	The Analysis-Genesis-Synthesis decomposition is intuitive and effective—mathematical reasoning precedes code generation, and collaborative tournaments refine implementation.
3.	The paper provides detail across 51 pages including extensive appendices covering implementation specifics, complete prompt templates, etc.

**Weaknesses:**

1. The evaluation covers only five toy-level PDEs—four 1D problems (Advection, Burgers, Reaction-Diffusion, Navier-Stokes) and one 2D steady-state problem (Darcy Flow)—which are textbook examples with well-established numerical methods, failing to demonstrate capability on challenging real-world scenarios such as 3D turbulent flows, coupled multi-physics systems, the NS equations, the Maxwell Equations etc.

2. The figures resemble simple process diagrams without conveying the holistic design, making it difficult to grasp how components integrate and why this specific architecture addresses limitations of existing methods.
3. Section 3 provides only informal prose descriptions without formal problem formulations, algorithmic pseudocode, or mathematical definitions

**Questions:**

1.	Are the three judge LLMs different models, or the same model with different prompts/temperatures? Does judge diversity (e.g., mixing reasoning models like o3 with general-purpose models like GPT-4o) improve tournament outcomes?
2.	How do PDE-SHARP-generated solvers compare in accuracy and efficiency to hand-tuned implementations using established libraries (FEniCS, PETSc) by domain experts? Is the goal to match expert performance or provide accessible automation for non-experts?
3.	Section 3 describes Genesis in only two sentences—how exactly does the stage translate Analysis outputs (PDE classification, stability bounds, decomposition decisions) into concrete solver code?
4.	The paper uses "n" for both initial solver candidates and top-n/2 selections

---

> ### Author Response · Authors · 2025-12-02
> **Response to Reviewer 51hJ (1/4)**
>
> **Weaknesses:**
>
> **1.** We agree that scaling to complex 3D, turbulent, and multiphysics systems is the ultimate goal for automated PDE solving frameworks. PDE-SHARP is a deliberate, foundational step toward that. Our argument is structured in four key points as follows:
>
> - The field is not yet solving even standard benchmarks robustly. PDE-SHARP addresses this methodological bottleneck. Current LLM-driven test-time compute frameworks (recent work such as CodePDE [1], OptiLLM [2], other large agentic workflows such as FunSeach [3], AIDE [4], etc.) remain far from saturating standard benchmarks. As we show in Table 2 and Appendix A.3.2. on agentic workflows, these methods fail dramatically on canonical 1D/2D tasks --- for instance, producing errors >0.9 nRMSE on reaction-diffusion and failure to converge (Figure 8), compared to PDE-SHARP’s 2e-03 nRMSE and proper convergence order (Figure 8). Even large agentic workflows fail on this task. These failures occur because existing test-time computing frameworks, no matter how large, lack mathematical analysis and efficient targeted refinement. If a framework cannot identify an exact ODE solution for a reaction term, it cannot properly scale to 3D turbulence. PDE-SHARP demonstrates the importance of mathematical analysis for code generation in PDE context. **PDE-SHARP’s core contribution is to address the methodological problem of inefficient, low-quality LLM-driven solver generation: our Analysis stage provides PDE-specific mathematical reasoning leading to more mathematically-informed initial solver generation, and our Synthesis stage performs efficient, feedback-driven refinement.**
>
> - These benchmarks were chosen to diagnose and address the core methodological failures of LLM-driven solver generation in a controlled, interpretable setting. To fix the inefficient, low-quality solver generation process itself, we need a benchmark suite that is:
> (1) diverse in mathematical structure (hyperbolic, parabolic, elliptic, linear, nonlinear) to test different solver strategies, (2) rich in real-world numerical behaviors (transport, shocks, stiff coupling, steady-state) to stress-test numerical robustness and (3), interpretable and well-established so failure modes can be precisely diagnosed, not obscured by problem complexity.
> The PDEBench tasks satisfy all three. For example, Reaction-Diffusion tests whether a framework can discover hybrid analytical-numerical strategies; Burgers and Navier-Stokes test shock-capturing and conservation; Darcy tests handling of elliptic systems. Our ablation studies (Appendix B.3) confirm that different PDEs benefit different components of PDE-SHARP, which is a flexible, PDE-aware framework. We start with these foundational building blocks because they expose the methodological gaps that must be closed before future work on scaling --- gaps that current LLM-driven approaches still fail to address, as evidenced by high error rates (Table 2) and inconsistent convergence (Fig. 8).
>
> - PDE-SHARP’s efficiency gain by replacing solver executions with API calls is an essential enabler for future scaling. Baseline best-of-N methods require at least 30+ solver executions per task --- prohibitive for 3D simulations that take hours or days each.
>
> Now, to address your comment, we have scaled our experiments in the revised manuscript. We added:
>
> - Appendix B.2 & D: Comprehensive comparison against high-level library-based solvers. We used Claude Code (a state-of-the-art coding agent) to generate solvers that directly call established PDE libraries (FEniCS, JAX-CFD, etc.), which we will elaborate more on in the next response to your Question 2 relevant to this topic.
>
> - Appendix D.4: We have conducted experiments on 3D Compressible Navier-Stokes equations (dataset from PDEBench, inviscid-dominated regime, periodic domain) in Appendix D.4.
>
> Key additions and results:
> 1. PDE-SHARP–generated solvers (three distinct strategies):
> - Solver A.1: Custom finite-volume MUSCL scheme with Rusanov flux, adaptive RK3, and explicit shock-capturing. Implements conservative form—critical for correct shock dynamics—and enforces positivity/flooring.
>
> - Solver A.2: Custom pseudo-spectral method using FFT-based derivatives and 4th-order RK. Retains full viscous terms and is tailored for smooth, high-accuracy simulation.
>
>
> - Solver B: Library-integrated spectral solver using Dedalus, with 3/2 dealiasing, symbolic equation parsing, and built-in MPI parallelism. This shows PDE-SHARP can correctly structure calls to high-performance spectral frameworks.
>
> 2. Baseline (Claude Code + high-level PDE libraries):
>
> Solver C: Uses FEniCSx finite-element library with a monolithic Newton solver. While robust, it employs a low-order (P1) implicit formulation that is unnecessarily heavy for this periodic-box problem and misses opportunities for spectral efficiency.
>
> (The rest of the key points of Appendix D.4. follows in the next response.)

---

> ### Author Response · Authors · 2025-12-02
> **Response to Reviewer 51hJ (2/4)**
>
> **Weaknesses (continued):**
>
> **1.** Continuing the key points of Appendix D.4 on 3D CNS:
>
> 3. Observation: PDE-SHARP’s use of libraries is superior!
>
> - Mathematical insight guides choice: PDE-SHARP’s Analysis stage identified the need for conservative formulations and shock-aware discretizations, leading it to generate Solver A.1 (finite-volume) and Solver B (spectral with dealiasing). Solver A.1 adopts a conservative finite-volume method with MUSCL limiting, while Solver B leverages Dedalus—configured with proper dealiasing and stability-informed time-stepping—rather than default settings. In contrast, Claude Code defaulted to a generic FEM implementation (Solver C) without exploiting periodicity or recognizing the dominance of hyperbolic character.
>
>
> - Strategic hybridization: Following analysis, the Synthesis stage refines these decisions through structured hybrid evaluations. For instance, finite-volume and spectral methods were directly compared for stability and accuracy. This led to the retention of the library-integrated Dedalus solver, customized with optimal settings, rather than relying on a generic configuration. PDE-SHARP created both custom kernels (A.1, A.2) and library-integrated solvers (B), demonstrating flexibility. Solver B uses Dedalus not as a black box, but with appropriate dealiasing and time-stepping settings derived from stability analysis.
>
> - Avoidance of critical pitfalls: PDE-SHARP’s solvers maintain conservation laws and include proper CFL/viscous time-step controls. Claude Code’s solver, while stable, uses a first-order implicit scheme that is less accurate and computationally heavier than necessary for this benchmark.
>
> - Higher numerical accuracy: Benchmarking on 50 samples from the 3D Navier-Stokes PDEBench dataset shows Solver B (Dedalus spectral) with the lowest mean nRMSE: 1.7% (velocities) and 2.6% (thermodynamic fields). Solver A.2 (custom pseudo-spectral) had mean errors of 5.1% and 6.3%. Solver A.1 (finite-volume) had 9.8% and 11.9%, consistent with its dissipative nature. Solver C (finite-element) showed the highest errors: 7.5% and 16.2%, due to first-order discretization on the given mesh.
>
> 4. Detailed outcomes from Appendix D.4:
>
> All PDE-SHARP solvers produced working stable code for 3D compressible Navier-Stokes. The finite-volume solver (A.1) implements a production-ready numerical strategy (MUSCL + Rusanov) that is shock-robust. The spectral solver (B) achieves high-order accuracy and uses parallelization-ready Dedalus infrastructure.
>
> Claude Code’s solver (C), though functional, is overly generic (FEM for a periodic box) and lacks the tailored efficiency of PDE-SHARP’s outputs.
>
> **2 \& 3.** Thank you very much for your comment! Mathematical definitions of the metrics and tasks are included in Appendix A and Appendix C. We have significantly expanded the method description in the revised manuscript: Figure 1 (Section 1) now includes a more detailed flow for the full PDE-SHARP pipeline. We added another detailed diagram in Section 3 (Figure 2), which includes the PDE-SHARP process with the reaction-diffusion task as a concrete example, tracing the decision trail for this PDE to illustrate how the Analysis stage’s mathematical reasoning leads to a refined solver choice in the Genesis stage. Appendix E presents a fully annotated PDE-SHARP report for the same equation, detailing the entire decision process from initial classification through to the final refined code from the Synthesis stage at the end. It includes formal problem formulations, full codes, and prompts, using a concrete example to aid understanding.
>
> **Questions:**
>
> **1.** We have tested several configurations for the judges in our extensive ablation studies, see Appendix B.3.5 (Table 19). In the main text (Table 2, default configuration), the three judge LLMs are different models. As pointed out in Section 4.1 and Appendix B.3.5, our default configuration uses a mixture of reasoning and non-reasoning models (o3, DeepSeek-R1, and GPT-4o). Your intuition is correct: judge diversity improves outcomes! The ablation study in Appendix B.3.5 provides empirical evidence. The mixed-judge panel overall achieves superior accuracy compared to homogeneous panels (all-reasoning or all-non-reasoning). We observed that mixed combination balances the deep logical reasoning of specialized models (o3, DeepSeek-R1) with the robust code-generation capabilities of general-purpose models (GPT-4o), leading to more effective collaborative refinement. We observed that this diversity fosters collaboration during the Synthesis stage since judges review feedback from each other's solvers as well, allowing them to explore different hybridization strategies. If one judge cannot execute an idea effectively, others can build upon it. We see an example of this collaborative refinement in the reaction-diffusion hybridization (Appendix E in the revised manuscript shows a full summary of the Synthesis stage for reaction-diffusion).

---

> ### Author Response · Authors · 2025-12-02
> **Response to Reviewer 51hJ (3/4)**
>
> **Questions (continued):**
>
> **2.** Thank you for raising this essential question. It allows us to clarify both the goal of PDE-SHARP and its performance relative to established practices.
>
> 1.  Our primary goal as discussed before (Weakness 1) is **methodological**: to study and improve the efficiency and quality of LLM-driven PDE solver generation that relies on inefficient brute-force sampling and produces code that is not mathematically informed. PDE-SHARP does this through: (1) Analysis Stage: Provides PDE-specific mathematical reasoning (classification, stability analysis, hybrid strategy detection), leading to mathematically-informed initial solver designs, (2) Synthesis Stage: Employs collaborative LLM judges in performance-informed tournaments to refine implementations, drastically reducing the need for exhaustive test-time sampling. These PDE-SHARP components coming together make robust PDE solver more accessible (by automating mathematical reasoning and refinement) and faster (by replacing exhaustive brute-force sampling with a few API calls).
>
> Regarding your question on expert and non-expert usage, it is interesting to note that PDE-SHARP, by design, can also provide easier, faster, and more intuitive iteration and editing assistance to anyone, expert or non-expert. Testing a solver idea typically requires generating and executing many code variants. PDE-SHARP allows any user to specify their idea or problem within the Analysis stage prompts, and refines it using chosen feedback (e.g., residual, error), yielding a high-quality, interpretable implementation without exhaustive sampling. The process itself can yield insightful strategies that inform the user’s own understanding.
>
> Given the goal of PDE-SHARP in addressing methodogical gaps in *LLM-driven frameworks*, we added extensive experiments comparing against a strong proxy for expert library usage ("Claude Code + PDE Libraries" in Appendix B.2 Table 17). This represents a state-of-the-art LLM agent tasked explicitly with generating solvers by calling high-performance libraries (FEniCS, PETSc, JAX-CFD, etc.)
>
> Important note: PDE-SHARP is not opposed to library calls; our prompts (Appendix F) encourage LLMs to use them, and the Synthesis stage refines such implementations. The key distinction is mathematical guidance as seen in Appendix D where we compare these solvers in detail based on mathematical structure.
>
> Appendix D in the revised manuscript contains full case studies with code examples for Advection, Reaction-Diffusion, 1D Navier-Stokes, and 3D Navier-Stokes, comparing PDE-SHARP solvers against these LLM-generated library-based solvers. The comparisons are multi-faceted, covering solution accuracy (nRMSE), numerical robustness (convergence), and mathematical design.
>
> **Overall Accuracy (nRMSE):** As shown in Table 17, PDE-SHARP outperformed this library-based baseline on 4 out of 5 PDEs, achieving superior nRMSE. Performance was comparable on Darcy Flow. This demonstrates that PDE-SHARP's mathematically-guided generation can surpass LLM-driven automation through Claude Code that merely stitches together library calls.
>
> **Numerical Robustness and Convergence:** Beyond final accuracy, the quality of the generated solvers is critical. Figure 8 and Appendix B.4 analyze convergence rates and stability. PDE-SHARP solvers consistently exhibit stable, predictable convergence under grid refinement. In contrast, the library-based solvers sometimes fail to converge or show erratic behavior, particularly for problems with shocks or stiffness (e.g., Burgers, Navier-Stokes). This indicates that PDE-SHARP produces more numerically robust implementations, due to its numerical stability step.
>
> **Mathematical Structure and Design Superiority:** The most significant distinction lies in the mathematical insight codified into the solvers. Appendix D provides detailed dissections. We highlight two key examples here:
>
> 1. Reaction-Diffusion (Appendix D.2): PDE-SHARP's Analysis stage identifies that the reaction term admits an exact analytical solution. It therefore generates a hybrid Strang-splitting solver that integrates the reaction term exactly and couples it with a finite-difference diffusion step. The LLM library-based solver defaulted to a generic, first-order IMEX finite-element method (using FEniCS) that fails to exploit this structure, leading to significantly higher truncation error and stiffness.
>
> (The second example follows in the next response.)

---

> > ### Author Response · Authors · 2025-12-02
> > **Response to Reviewer 51hJ (4/4)**
> >
> > **Questions (continued):**
> >
> > **2.** Continuing to the second example of mathematical structure and design superiority:
> >
> > 2. 1D Compressible Navier-Stokes (Appendix D.3): In this case, both the PDE-SHARP and library-based baseline use the well-known JAX-CFD library. PDE-SHARP's analysis mandates a conservative formulation (solving for ρ,ρv,E) and selects an SSP-RK3 time integrator for stability. The library-based solver  produced a non-conservative spectral method in primitive variables, which lead to instability in execution. While both solvers use JAX-CFD, they extract fundamentally different capabilities: PDE-SHARP uses it as a numerical utility for derivative computation, but the library Approach uses it as a full domain abstraction and boundary handling while implementing spectral method. Here, we observe PDE-SHARP’s ability to selectively integrate library features based on mathematical analysis rather than defaulting to library conventions.
> >
> > In both cases, PDE-SHARP’s mathematical reasoning led to fundamentally better algorithm choices (either in custom solver generation or library usage) than those made by an LLM simply instructed to use available libraries. It demonstrates that the key value is not just automation, but automation informed by targeted mathematical analysis. Full details of these examples (and more of the advection PDE and the newly added 3D CNS task) along with the solver codes are available in Appendix D.
> >
> > **3.** The Genesis stage translates the mathematical insights from Analysis into executable solver code using the PDE classification result, identified solution strategy, and numerical stability constraints passed on to it from Analysis. We added Figure 2 to the revised manuscript, which traces the decision trail for the reaction-diffusion PDE to illustrate how the Analysis stage’s mathematical reasoning leads to a refined solver choice in the Genesis stage. Appendix E now presents a fully annotated PDE-SHARP report for the same equation, detailing the entire decision process. To give a quick summary of the process: Analysis produces a decision trail (e.g., "PDE is Reaction-Diffusion; hybrid analytical-numerical approach is viable; use Strang splitting; explicit diffusion time-step bound is 0.25*dx**2/nu") which is saved as a JSON. This structured information is fed into the code-generation LLM via the prompts in Appendix F.2, which include the solver template and a set of "Code Generation Criteria" (available in Appendix F). The LLM is instructed to generate code that implements the strategies and stability bounds identified during Analysis. For example, for the Reaction-Diffusion equation, it is directed to implement a Strang-splitting scheme with an analytically integrated reaction step, using the precise dt_max formula derived from stability analysis, which proves to be superior to all other reaction-diffusion solvers generated by the baselines.
> >
> > **4.** Thank you very much for pointing this out. It is fixed in the revision.
> >
> >
> > References:
> >
> > [1] Shanda Li, Tanya Marwah, Junhong Shen, Weiwei Sun, Andrej Risteski, Yiming Yang, and Ameet Talwalkar. Codepde: An inference framework for llm-driven pde solver generation, 2025. URL https://arxiv.org/abs/2505.08783.
> >
> > [2] Asankhaya Sharma. Optillm: Optimizing inference proxy for llms, 2024. URL https:// github.com/codelion/optillm.
> >
> > [3] Bernardino Romera-Paredes, Mohammadamin Barekatain, Alexander Novikov, Matej Balog, M. Pawan Kumar, Emilien Dupont, Francisco J. R. Ruiz, Jordan Ellenberg, Pengming Wang, Omar Fawzi, Pushmeet Kohli, and Alhussein Fawzi. Mathematical discoveries from program search with large language models. Nature, 2023. doi: 10.1038/s41586-023-06924-6.
> >
> > [4] Zhengyao Jiang, Dominik Schmidt, Dhruv Srikanth, Dixing Xu, Ian Kaplan, Deniss Jacenko, and Yuxiang Wu. Aide: Ai-driven exploration in the space of code, 2025b. URL https://arxiv. org/abs/2502.13138.

---

### Meta-Review · Area_Chair_v5mj · 2026-01-03

**Summary:**

This paper proposed a three-stage LLM pipeline that uses PDE analysis and iterative tournaments to generate accurate solvers with fewer evaluations.

**Reviewer Concerns:**

Main concerns from reviewers include:

1. Evaluation is on mostly toy 1D/2D PDEBench tasks, with weak comparison to expert/library or general code-agent baselines.

2. Numerical assessment relies mainly on nRMSE; reviewers request convergence and cost/time analyses.

3. Method presentation lacks formal algorithm/pseudocode and clearer figures.

4. API overhead/cost may be high.

**Reviewer Scores:**

Reviewer / Score

51hJ	4

KcJb	2

h7JT	6

xu6V	6

Average	4.5

No reviewers indicated to increase or decrease their scores.

---

### Decision · Program_Chairs · 2026-01-26

Reject